# PP6 regulation of Aurora A–TPX2 limits NDC80 phosphorylation and mitotic spindle size

Tomoaki Sobajima[1]*, Katarzyna M. Kowalczyk[1]*, Stefanos Skylakakis[1]*, Daniel Hayward[2,3], Luke J. Fulcher[1], Colette Neary[1], Caleb Batley[1], Samvid Kurlekar[1], Emile Roberts[2], Ulrike Gruneberg[2], and Francis A. Barr[1]

**Amplification of the mitotic kinase Aurora A or loss of its regulator protein phosphatase 6 (PP6) have emerged as drivers of genome instability. Cells lacking PPP6C, the catalytic subunit of PP6, have amplified Aurora A activity, and as we show here, enlarged mitotic spindles which fail to hold chromosomes tightly together in anaphase, causing defective nuclear structure. Using functional genomics to shed light on the processes underpinning these changes, we discover synthetic lethality between PPP6C and the kinetochore protein NDC80. We find that NDC80 is phosphorylated on multiple N-terminal sites during spindle formation by Aurora A–TPX2, exclusively at checkpoint-silenced, microtubule-attached kinetochores. NDC80 phosphorylation persists until spindle disassembly in telophase, is increased in PPP6C knockout cells, and is Aurora B-independent. An Aurora-phosphorylation-deficient NDC80-9A mutant reduces spindle size and suppresses defective nuclear structure in PPP6C knockout cells. In regulating NDC80 phosphorylation by Aurora A–TPX2, PP6 plays an important role in mitotic spindle formation and size control and thus the fidelity of cell division.**

## Introduction

Micronucleation and related structural defects of the nucleus are emerging as important drivers for genome instability in cancers and metastasis (Bakhoum et al., 2018; Crasta et al., 2012; Zhang et al., 2015). As a consequence, there is great interest in understanding how these changes arise and whether they expose vulnerabilities that can be exploited by new therapies. Defective chromosome segregation in mitosis has long been understood to be a cause of changes to nuclear structure (Levine and Holland, 2018). However, few of the structural or regulatory components required for chromosome segregation are mutated in cancers, and the specific molecular mechanisms going awry are often unclear and thus require further investigation. Here, we focus on two regulators of mitotic spindle formation, Aurora A and PPP6C, the catalytic subunit of protein phosphatase 6 (PP6), which are either amplified or undergo loss of function mutations in cancers, respectively (Anand et al., 2003; Bischoff et al., 1998; Hammond et al., 2013; Hodis et al., 2012; Krauthammer et al., 2012; Sen et al., 1997).

Chromosome segregation is mediated by the mitotic spindle (Goshima and Scholey, 2010; Petry, 2016), a structure formed when microtubules emanating from the separated centrosomes at the spindle poles attach to chromosomes through a conserved protein complex at kinetochores formed by NDC80, NUF2, and SPC24/25 (Cheeseman et al., 2006; Ciferri et al., 2008; DeLuca et al., 2006). The number and length of these microtubules and hence the formation of the mitotic spindle is tightly regulated in proportion to the number of chromosomes or amount of chromatin (Goshima and Scholey, 2010; Petry, 2016). Chromatin promotes spindle formation by the RCC1–Ran pathway, which regulates the availability of a cohort of microtubule-binding spindle assembly factors (Carazo-Salas et al., 1999; Clarke and Zhang, 2008; Heald et al., 1996; Kalab et al., 1999). Key among these factors is TPX2, a microtubule-binding protein and the targeting and activating subunit for the mitotic kinase Aurora A, which phosphorylates TPX2 and other proteins important for spindle formation, positioning, and orientation (Bayliss et al., 2003; Bird and Hyman, 2008; Fu et al., 2015; Gruss et al., 2001; Gruss et al., 2002; Helmke and Heald, 2014; Kotak et al., 2016; Kufer et al., 2002; Polverino et al., 2021). Although Aurora A and TPX2 carrying microtubules contact the kinetochores during mitotic spindle formation, interestingly, TPX2 mutants unable to bind Aurora A still form bipolar but much shorter spindles and are able to support chromosome segregation albeit with some defects (Bird and Hyman, 2008). Aurora A and TPX2 are therefore important for the formation of correctly scaled mitotic spindles; however, the key Aurora A targets explaining this function have not been identified yet.

This spindle assembly process is monitored by two additional kinases, Aurora B, localized to centromeres, and MPS1, which is

....................................................................................................................................................................................

[1]Department of Biochemistry, University of Oxford, Oxford, UK;   [2]Sir William Dunn School of Pathology, University of Oxford, Oxford, UK;   [3]Randall Centre for Cell and Molecular Biophysics, King's College London, London, UK.

*T. Sobajima, K.M. Kowalczyk, and S. Skylakakis contributed equally to this paper.   Correspondence to Francis A. Barr: francis.barr@bioch.ox.ac.uk.

dynamically recruited to kinetochores (Musacchio, 2015). Together Aurora B and MPS1 detect and initiate the checkpoint signal preventing exit from mitosis until spindle assembly is complete. MPS1 recruitment occurs at kinetochores with incorrect microtubule attachment geometries or without attached microtubules. Microtubule binding and the generation of tension are thought to physically pull the microtubule-binding outer kinetochore proteins away from Aurora B at the centromere, thus altering their phosphorylation state (Lampson and Cheeseman, 2011). NDC80, the principal microtubule binding factor at the kinetochore, has been reported to be phosphorylated within its N-terminal region by Aurora B in the absence of tension, reducing microtubule-binding affinity and promoting MPS1 recruitment, spindle checkpoint signaling, and chromosome biorientation (Ciferri et al., 2008; DeLuca et al., 2006; DeLuca et al., 2011; Ji et al., 2015; Zhu et al., 2013). A similar function has been proposed for Aurora A phosphorylation of NDC80 during chromosome alignment and pole-based error correction processes (Ye et al., 2015). In that case, NDC80 is phosphorylated at incorrectly positioned chromosomes overlapping with Aurora A activity at spindle poles, allowing movement of the chromosome toward the metaphase plate by other Aurora-controlled kinesin motors such as KIF4A and CENP-E (Kim et al., 2010; Poser et al., 2019).

Although Aurora A and Aurora B are related kinases with highly similar phosphorylation consensus motifs ([RK]x[TS] [ILV]) and their activities appear to converge on some common targets such as NDC80, other evidence shows they must also have distinct functions. Inhibition of Aurora A results in spindle formation defects and a spindle assembly checkpoint–dependent arrest in mitosis, whereas inhibition of Aurora B has the opposite effects and results in loss of the spindle assembly checkpoint signal and failure to arrest in mitosis when spindle defects are present (Ditchfield et al., 2003; Hauf et al., 2003; Hoar et al., 2007). However, the specific targets of Aurora A and Aurora B that explain these differences and the mechanisms that avoid crosstalk between the two kinases are not clearly understood. In addition to their distinct localizations, another point of difference between Aurora A and B is in their regulation. Aurora B activity on chromosomes is controlled by the PP1–Repoman complex (Qian et al., 2015), whereas the activity of Aurora A–TPX2 complexes is limited by PP6 (Zeng et al., 2010). This regulation of Aurora A is necessary for proper spindle assembly, and cancer-associated loss of function driver mutations in PPP6C, the catalytic subunit of PP6, leads to micronucleation due to defective chromosome segregation (Hammond et al., 2013; Hodis et al., 2012; Krauthammer et al., 2012; Zeng et al., 2010). These micronuclei are positive for the DNA damage marker γ-H2AX, consistent with other findings on the consequences of micronucleation and direct evidence that the loss of PPP6C is a driver for genome instability (Crasta et al., 2012; Hammond et al., 2013; Hodis et al., 2012; Krauthammer et al., 2012; Zeng et al., 2010; Zhang et al., 2015).

Despite understanding the importance of Aurora A and PP6 for spindle assembly and chromosome segregation, the crucial Aurora A targets that explain genomic instability and micronucleation when its activity is amplified remain unclear. Here,

we show that PP6 regulation of Aurora A–TPX2 complexes plays a crucial role in regulation of mitotic spindle size, by controlling phosphorylation of the kinetochore protein NDC80. We find that during spindle formation, NDC80 is phosphorylated exclusively at checkpoint-silenced, microtubule-attached kinetochores by K-fiber associated Aurora A–TPX2 complexes, calling into question the previously reported role for Aurora B in phosphorylating NDC80 at checkpoint-active kinetochores.

## Results

### Mitotic spindle size is increased in PPP6C knockout (KO) cells
To address the mechanistic consequences of Aurora A amplification, we constructed PPP6C KO HeLa cell lines and confirmed they showed abnormal nuclear structure and an increase in the activating pT288 phosphorylation on Aurora A (Fig. S1, A–C). Compared with the control, PPP6C KO cells showed a significant and reproducible increase of metaphase spindle size from 9 to 12 μm, P < 0.001 (Fig. 1 A). To test if the increase of spindle size in PPP6C KO cells was dependent on the amplified Aurora A activity, Aurora A kinase inhibitors were used, with staining for active Aurora A pT288 to confirm Aurora A inhibition. Inhibition of Aurora A resulted in loss of Aurora A pT288 and significantly smaller spindles in the parental cells (Fig. 1 B). In PPP6C KO cells where spindle size is increased relative to the parental control, inhibition of Aurora A also resulted in significantly smaller spindles (Fig. 1 B). As PPP6C KO cells cannot efficiently dephosphorylate Aurora A pT288, Aurora A is retained on the spindle (Fig. 1 B), but the inhibitor is still expected to prevent kinase activity toward downstream targets.

To differentiate between effects on spindle size caused by altered centrosome or spindle pole separation and the length of microtubules attached to the kinetochores, we used monopolar spindle assays. For these assays, cells were treated with the KIF11 inhibitor S-trityl L-cysteine (STLC) to create monopolar spindles (Skoufias et al., 2006). In this situation, chromosomes are captured by spindle microtubules and form a rosette clustered around a single combined spindle pole. Monopolar spindle diameter was increased from 7 μm in parental cells to 9 μm in PPP6C KO cells, P < 0.0001 (Fig. 1 C). This effect was prevented by the addition of an Aurora A inhibitor with monopolar spindles tending to the same minimal size in both cases (Fig. 1 C). By contrast, Aurora B inhibition resulted in the spread of chromosomes in both parental and PPP6C KO cells and appeared additive to the effects of amplified Aurora A in PPP6C KO cells (Fig. 1 C). Thus, the larger spindle size observed in PPP6C KO cells is Aurora A activity dependent but does not appear to be explained by altered regulation of spindle pole separation since monoastral spindle size was also increased.

Previously, we demonstrated that cells with perturbed PPP6C or cancer-associated loss of function mutants in PPP6C show chromosome instability, leading to the formation of aberrant nuclei and micronuclei with damaged DNA (Hammond et al., 2013; Zeng et al., 2010). We, therefore, explored the functional consequences of PPP6C KO for chromosome segregation in more detail. The enlarged spindles in PPP6C KO cells showed less compact metaphase plates and failed to tightly hold segregating

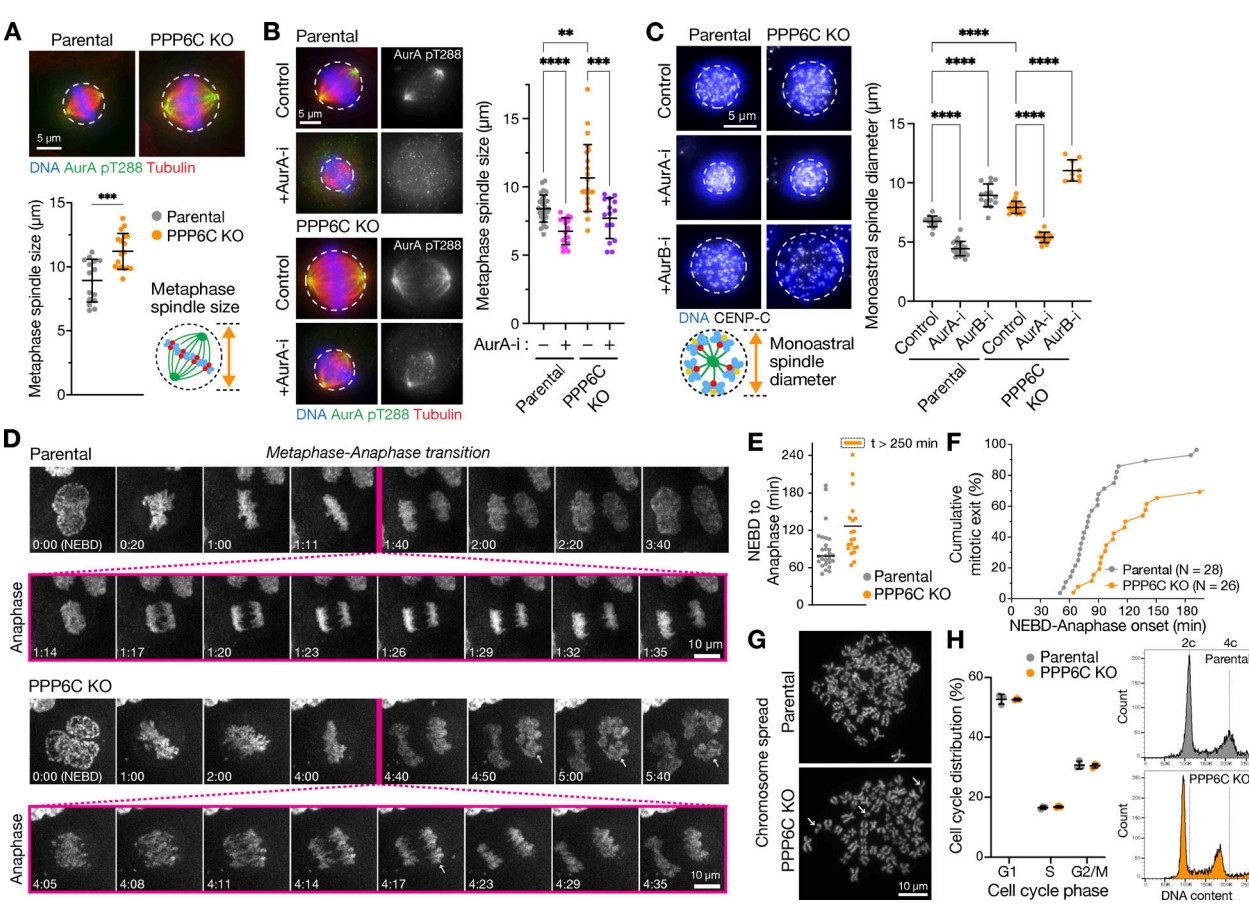

Figure 1. **PP6 and Aurora A regulate the size of the mitotic spindle. (A)** Metaphase spindle size (mean ± SD; *n* = 12–13) in parental and PPP6C KO HeLa cell lines stained for active Aurora A pT288, tubulin, and DNA. Statistical significance was analyzed using an unpaired two-tailed *t* test with Welch's correction (***, P < 0.001). **(B)** Metaphase spindle size (mean ± SD; *n* = 15–29) in parental and PPP6C KO HeLa cell lines after 30 min treatment in the presence (+) or absence (–) of Aurora A inhibitor (AurA-i). Statistical significance was analyzed using a Brown-Forsythe ANOVA (**, P < 0.01; ***, P < 0.001; ****, P < 0.0001). **(C)** Parental and PPP6C KO cell lines were treated with STLC for 3 h to arrest cells in mitosis with monopolar spindles and then treated for 30 min in the absence (Control) and presence of Aurora A (AurA-i) or Aurora B (AurB-i) inhibitors. The cells were then stained for DNA and CENP-C. Monoastral spindle diameter (mean ± SD; *n* = 9–21) is shown for the different conditions. Statistical significance was analyzed using a Brown-Forsythe ANOVA (****, P < 0.0001). **(D)** Time-lapse imaging of DNA segregation in parental and PPP6C KO cells. NEBD was taken as the start of mitosis. Anaphase is shown with higher time resolution with arrows to mark anaphase spindle defects in PPP6C KO cells. Arrows indicate chromosomes escaping the anaphase spindle. **(E and F)** Mitotic progression from NEBD to anaphase onset (E; the line marks the median value) and cumulative mitotic index in parental and PPP6C KO cells (F; *n* = 26–28). PPP6C KO cells show extended mitosis and delayed mitotic exit. **(G)** Mitotic chromosome spreads from parental and PPP6C KO HeLa cell lines. Arrows indicate broken or unpaired chromosomes. **(H)** Flow cytometry was used to measure cell cycle distribution (mean ± SD; *n* = 3) and ploidy of parental and PPP6C KO HeLa cells. Plots show counts of DNA content with dotted lines to mark 2c and 4c in the parental control cell line.

chromosomes into compact units in anaphase, resulting in the formation of aberrant nuclei and micronuclei (Fig. 1 D). This was accompanied by an increase in the time cells took to form a congressed metaphase plate after nuclear envelope breakdown (NEBD) and enter anaphase (Fig. 1, E and F). Consistent with our previous observations, chromosome spreads and flow cytometry provided evidence for chromosome instability in PPP6C KO cells, with an increase in the number of broken chromosomal fragments (Fig. 1 G) and a gradual loss of chromosomes, but importantly no major change to ploidy (Fig. 1 H). These changes and the overall reduction in average chromosome number are hence unlikely to explain the increased spindle size in PPP6C KO cells. We, therefore, conclude that PP6 and Aurora A have opposing effects on mitotic spindle size and that the amplified Aurora A activity in PPP6C KO cells acts on proteins important for spindle size control, chromosome alignment, and segregation (Fig. 2, A and B).

## Functional genomics reveals synthetic lethality between PPP6C and NDC80

To identify pathways and specific targets that could explain the effects of amplified Aurora A activity in cells lacking PP6, we performed haploid genomic screens for genes that showed synthetic growth defects with, or that suppressed, PPP6C KO (Fig. S2 A). First, we established that human haploid eHAP cells were a suitable model for this approach. PPP6C KO in human haploid eHAP cells resulted in an increase in the activating pT288 phosphorylation on Aurora A, enlarged mitotic spindles, spread of the active kinase on metaphase and anaphase spindles, and formation of micronuclei (Fig. S2, B–E). This recapitulated the phenotype of HeLa cells transiently depleted of PPP6C and melanoma cell lines with inactivating mutations in PPP6C, confirming that they were a suitable model to study loss of PP6 function (Hammond et al., 2013; Zeng et al., 2010). Replicate

Figure 2. **Genome-wide CRISPR screening reveals synthetic growth defects between the catalytic subunit of PP6 and the kinetochore protein NDC80. (A)** A schematic depicting the role of TPX2 in stabilization of the active pool of Aurora A at the mitotic spindle. Aurora A switches between inactive unphosphorylated (T-form) and active phosphorylated (P-form) conformations. PP6 dephosphorylates the Aurora A–TPX2 complex and promotes Aurora A inactivation. Removal of PP6 thus results in amplified Aurora A activity. **(B)** A cartoon outlining how the enlarged spindles in PPP6C KO cells fail to maintain the compact array of chromosomes seen in parental cells during metaphase and anaphase. Escaped chromosomes in PPP6C KO cells go on to form micronuclei and cause other nuclear shape defects. **(C)** Pooled genome-wide CRISPR screens were performed in parental and PPP6C KO eHAP cells. Data from two independent screens were combined and analyzed using Fisher's method to calculate Fisher's combined P value (FCP). Significance ($-\log_{10}$FCP) is plotted against the LFC in PPP6C KO compared with the parental cells. Significantly positively (orange) and negatively (blue) selected genes, P < 0.01 in both screens with mean LFC < 0.25 or >0.25 are highlighted on the plot.

genome-wide CRISPR screens were performed as described in the methods to allow for the difference in growth rate between parental and PPP6C KO cells. In both screens, a list of core essential genes required for viability across a panel of cell lines showed negative selection as expected (Fig. S2 F; Hart et al., 2017; Wang et al., 2019). These core essential genes were similarly selected against in both the parental and PPP6C KO cells, consistent with the idea that removal of PPP6C only affects a few specific pathways within the cell. To identify dependencies created by loss of PP6, we used these combined datasets to perform a genome-wide comparison of relative log fold-change on a per-gene basis in PPP6C KO compared with parental cells (Fig. 2 C and Table S1).

The Aurora A activator TPX2 was among a subset of genes showing significant positive selection in PPP6C KO cells, Fisher's combined P value FCP <0.01 (Fig. 2 C). We interpreted this as relative suppression of the growth defect in PPP6C KO compared with the parental eHAP cells. This agrees with the previously established positive relationship between Aurora A and TPX2 (Bayliss et al., 2003; Fig. 2 A) and showed that this approach could identify cellular pathways relevant to understanding Aurora A function. A slightly larger cohort of genes was found to undergo negative selection consistent with synthetic growth defects with PPP6C. Among these, the kinetochore protein NDC80 and all subunits of the HUSH chromatin silencing complex underwent significant negative selection, Fisher's combined P value FCP <0.01 (Fig. 2 C).

**TPX2-dependent Aurora A activity drives increased spindle size in PPP6C KO cells**

Previous work has suggested that the Aurora A–TPX2 complex regulates spindle length in human cells and may thus play a role in spindle size control (Bird and Hyman, 2008). The identification of TPX2 as a positively selected gene, potentially suppressing PPP6C KO, therefore fitted with the idea that PP6 is a specific regulator of Aurora A–TPX2 complexes. To test this idea,

the relation between Aurora A, TPX2, and PP6 in spindle size control was explored in parental and PPP6C KO HeLa cells. Overexpression of Aurora A or TPX2 in parental cells resulted in increased spindle size (Fig. 3 A), supporting the proposal that Aurora A activity is normally limiting for spindle formation. In agreement with that idea, replacement of TPX2 with a YYD/AAA mutant unable to bind Aurora A resulted in smaller spindles (Fig. 3 B). Western blotting confirmed that depletion of TPX2 reduced the level of active Aurora A pT288 and that WT TPX2 but not the YYD/AAA mutant increased pT288 levels (Fig. 3 C). TPX2 is therefore an important factor limiting Aurora A activation and formation of correctly sized mitotic spindles, as proposed previously (Bird and Hyman, 2008).

When TPX2 was depleted in PPP6C KO HeLa cells, metaphase spindle and monopolar spindle size tended to the same lower value seen in parental cells (Fig. 3, D and E). In both parental and PPP6C KO cells, active Aurora A pT288 was lost from the spindle microtubules but not the centrosomes (Fig. 3, D and E, AurA pT288). This agrees with previous reports that there are two separate populations of Aurora A dependent on TPX2 (Kufer et al., 2002) and the centrosomal protein CEP192 (Joukov et al., 2014), respectively. Thus, removal of TPX2 had similar effects on spindle size to Aurora A inhibition (Fig. 1, A–C). To confirm this effect was due to a failure to recruit and activate Aurora A on the microtubules, RNA interference rescue experiments using WT TPX2 or the YYD/AAA mutant were performed (Fig. 3 F). In parental cells expressing TPX2 YYD/AAA, there was a failure to recruit active Aurora A to the mitotic spindle, and spindle size was reduced (Fig 3, F–H). In PPP6C KO cells, although pT288 was elevated and spindle size was increased relative to the parental control, replacement of TPX2 with the YYD/AAA mutant resulted in a reduction of spindle size to the same extent as parental cells (Fig. 3, F–H). Taken together, these results show that the increase in spindle size in PPP6C KO cells is driven by TPX2-dependent Aurora A activity localized to the spindle microtubules.

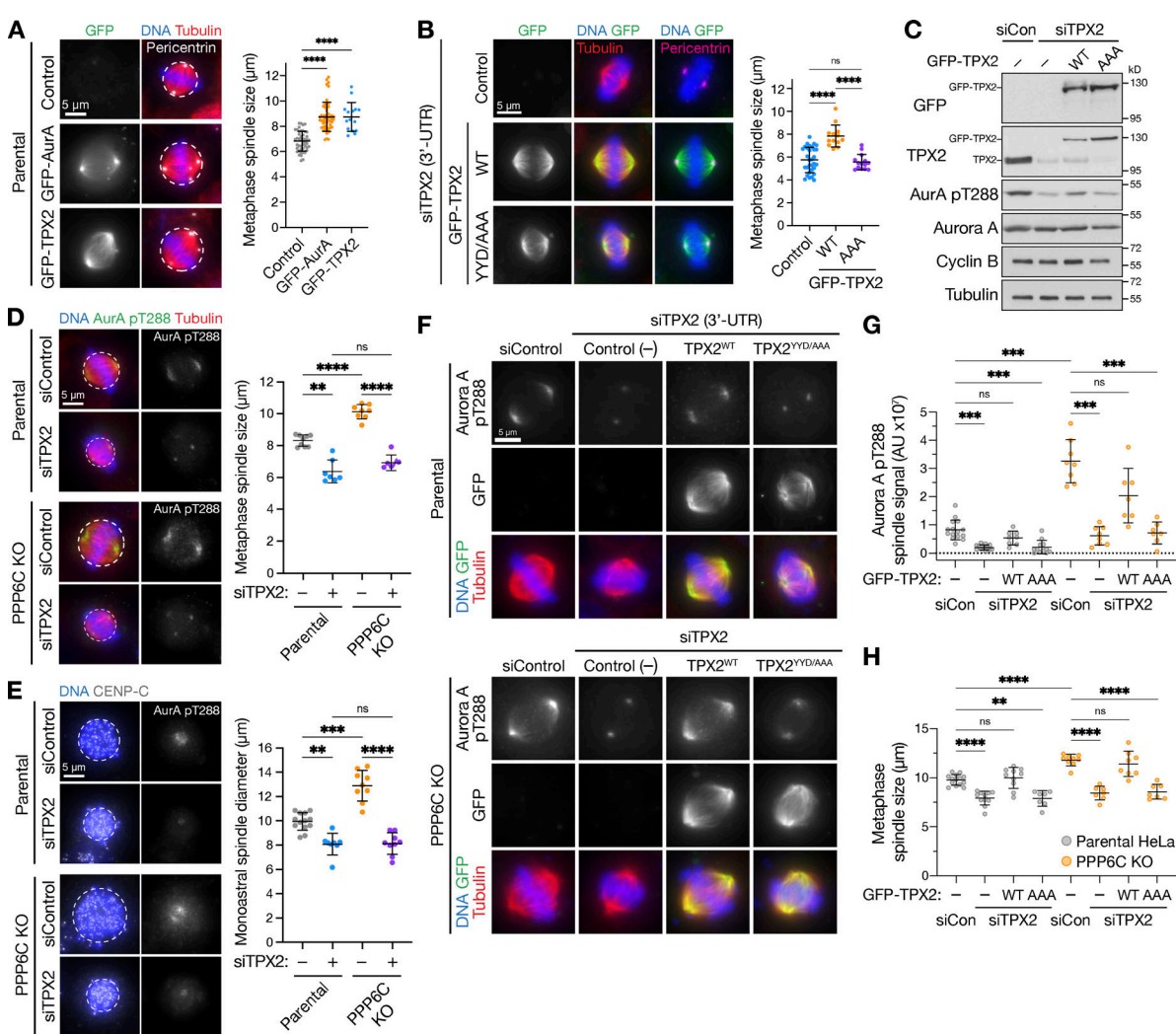

**Figure 3. Aurora A–TPX2 drives enlarged spindle size in PPP6C KO cells. (A)** Metaphase spindle size (mean ± SD; *n* = 15–52) was measured in HeLa cells overexpressing GFP-Aurora A, GFP-TPX2 compared with the untransfected control (Control). Pericentrin staining marks centrosomes at the spindle poles. Statistical significance was analyzed using a Brown-Forsythe ANOVA (****, P < 0.0001). **(B)** Metaphase spindle size (mean ± SD; *n* = 13–31) in HeLa cells depleted of endogenous TPX2 using a 3'-UTR siRNA and then transfected with GFP-TPX2 (WT) or a mutant unable to bind Aurora A (YYD/AAA) or left untransfected (control). Statistical significance was analyzed using a Brown-Forsythe ANOVA (****, P < 0.0001). **(C)** Western blot of cells in B showing depletion of endogenous TPX2 and expression of GFP-TPX2 constructs. **(D)** Metaphase spindle size (mean ± SD; *n* = 6–9) was measured in parental and PPP6C KO HeLa cells treated with control or TPX2 siRNA and stained for activated Aurora A pT288, tubulin, and DNA. Statistical significance was analyzed using a Brown-Forsythe ANOVA (**, P < 0.01; ****, P < 0.0001). **(E)** Parental and PPP6C KO cell lines treated with control or TPX2 siRNA were arrested in mitosis with monopolar spindles by STLC for 3 h, then stained for DNA and CENP-C. Monoastral spindle diameter (mean ± SD; *n* = 8–13) is shown for the different conditions. Statistical significance was analyzed using a Brown-Forsythe ANOVA (**, P < 0.01; ***, P < 0.001; ****, P < 0.0001). **(F–H)** Parental and PPP6C KO HeLa cells were treated with control or TPX2 3'-UTR siRNA and either mock transfected (Control [−]) or transfected with either GFP-TPX2 (WT) or the YYD/AAA mutant, and then stained for activated Aurora A pT288, tubulin, and DNA. The intensity of Aurora A pT288 signal (G) and metaphase spindle size (H) are shown for the different conditions in parental and PPP6C KO cells (mean ± SD; *n* = 7–15). Statistical significance was analyzed using a Brown-Forsythe ANOVA (**, P < 0.01; ***, P < 0.001; ****, P < 0.0001). Source data are available for this figure: SourceData F3.

## NDC80 phosphorylation is increased in PPP6C KO cells

We then turned to the negatively selected genes, candidates that show synthetic lethality with PPP6C KO, and carried out further validation experiments. NDC80 was prioritized in subsequent work for two reasons. First, the synthetic growth defect between NDC80 and PPP6C was validated in eHAP cells using specific CRISPR gRNAs (Fig. S2, G and H). Western blots showed that NDC80 was reduced, and in the case of gRNA2, truncated fragments were present (Fig. S2 I). This suggested that PPP6C KO cells are more sensitive to the levels of NDC80 than the parental cells,

indicative of a functional link between NDC80 and PP6. Second, when candidates from both screens were rescreened in aneuploid HeLa cells, only NDC80 depletion resulted in pronounced nuclear morphology and growth defects after 48 h in PPP6C KO compared with the parental HeLa cells (Fig. 4 A and Fig. S2 J). For the other candidate genes, there were less obvious growth defects and only modest increases in fragmented nuclei even at 72 h, suggesting they are less important than NDC80 for normal mitosis and chromosome segregation in PPP6C KO HeLa cells (Fig. S2 J). This may be due to differences between the haploid eHAP cells used

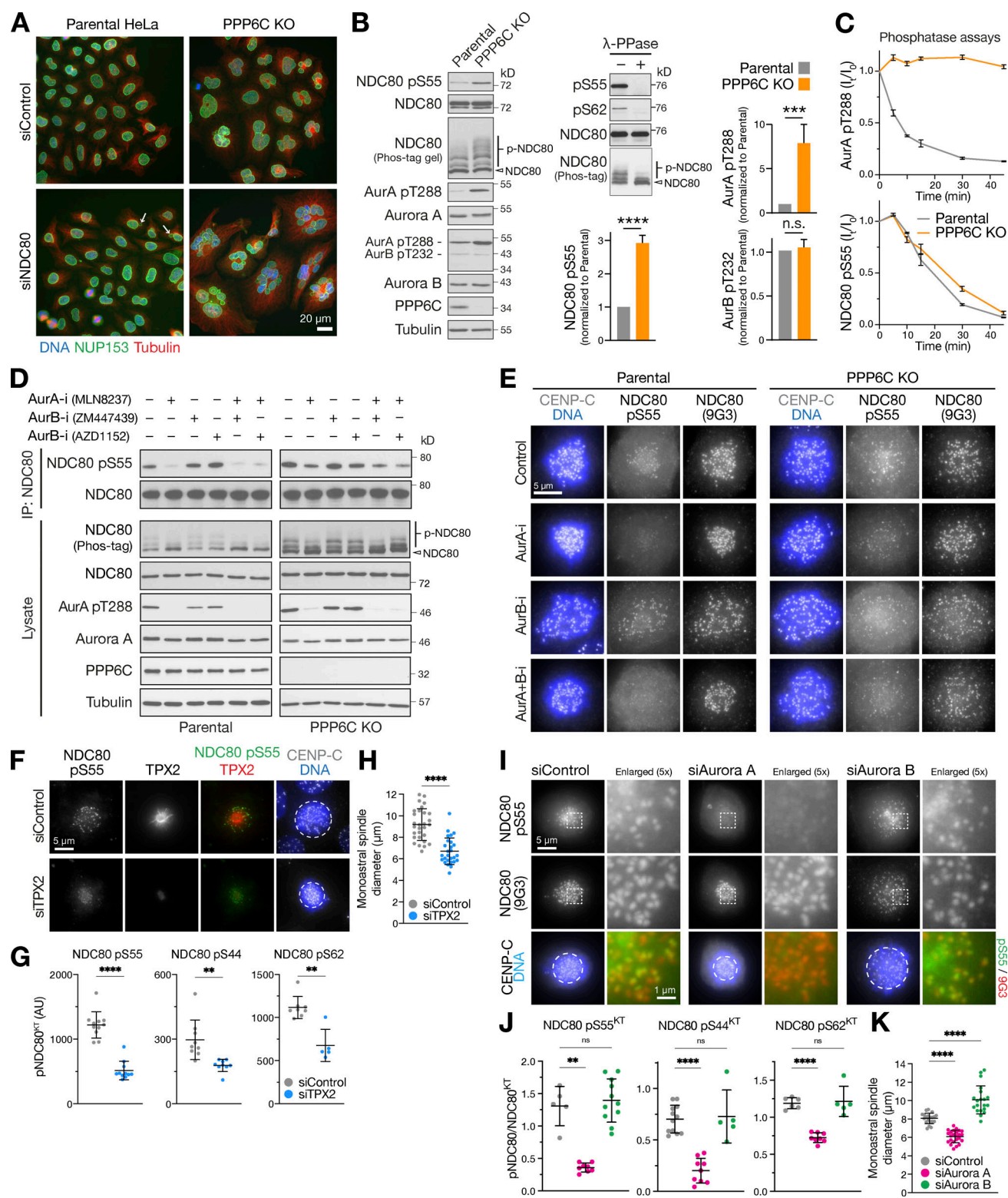

Figure 4. **PPP6C KO cells show elevated Aurora A dependent phosphorylation of the kinetochore protein NDC80. (A)** Parental and PPP6C KO cells depleted of NDC80 were stained for DNA, tubulin, and the nuclear pore marker NUP153, and imaged at 40× magnification to visualize nuclear morphology. Arrows mark micronuclei in NDC80-depleted parental cells. **(B)** Mitotic lysates of parental and PPP6C KO HeLa cells were blotted for the proteins listed in the figure. Overall NDC80 phosphorylation was monitored using a Phos-tag gel. The relative levels of NDC80 S55 phosphorylation, active Aurora A (pT288), and active Aurora B (pT232) were measured (mean ± SEM; $n$ = 4–8). Statistical significance was analyzed using an unpaired two-tailed $t$ test with Welch's correction (***, P < 0.001; ****, P < 0.0001). To control for antibody phospho-selectivity, HeLa cell lysates were mock- (−) or λ-phosphatase–treated (+λ-PPase) and blotted for NDC80, NDC80 pS55, and pS62. **(C)** Dephosphorylation kinetics of Aurora A pT288 and NDC80 pS55 were followed in extracts of parental and PPP6C KO cells. Graphs show dephosphorylation kinetics (mean ± SD; $n$ = 3). **(D)** Mitotic lysates of parental and PPP6C KO HeLa cells treated for 10 min with

Aurora A (AurA-i) and two different Aurora B (AurB-i) kinase inhibitors in the combinations shown were blotted for the proteins listed in the figure. To increase sensitivity, NDC80 was isolated by immunoprecipitation for NDC80 pS55 blots. An extended analysis of this experiment, including blots for additional NDC80 phosphorylation sites, is shown in Fig. S3 D. **(E)** Parental and PPP6C KO HeLa cells in different kinase inhibited and control conditions from D stained for NDC80 pS55, NDC80, and DNA. **(F)** HeLa cells were depleted of endogenous TPX2 for 72 h or treated with a non-targeting control siRNA (siControl). Cells were fixed and then stained for TPX2, CENP-C, DNA, or antibodies to specific NDC80 pS55. **(G and H)** NDC80 pS55, pS44, and pS62 signal at kinetochores (G) and monoastral spindle size in siControl and siTPX2 (H; mean ± SD; $n$ = 5–29). Statistical significance was analyzed using an unpaired two-tailed $t$ test with Welch's correction (**, $P < 0.01$; ****, $P < 0.0001$). **(I)** HeLa cells were depleted of Aurora A or Aurora B for 72 h using siRNA or treated with a non-targeting control siRNA (siControl). Cells were fixed and then stained with antibodies to specific NDC80 phosphorylation sites. Images for NDC80 pS55 are shown in the figure, pS44 and pS62 are shown in Fig. S4 D. **(J and K)** NDC80 pS55, pS44, and pS62 signal at kinetochores expressed relative to NDC80 (J) and monoastral spindle size in siControl, siAurora A, and siAurora B treated cells (K; mean ± SD; $n$ = 5–24). Statistical significance was analyzed using a Brown-Forsythe ANOVA (**, $P < 0.01$; ****, $P < 0.0001$). Source data are available for this figure: SourceData F4.

for the functional genomics and highly aneuploid HeLa cells used for validation, but was not explored further here.

NDC80 has been reported to be phosphorylated within its N-terminal region either by Aurora B to promote spindle checkpoint signaling and chromosome biorientation or by Aurora A during chromosome alignment and pole-based error correction processes (DeLuca et al., 2006; DeLuca et al., 2011; Ji et al., 2015; Lampson and Cheeseman, 2011; Ye et al., 2015; Zhu et al., 2013). Using mass spectrometry, we could confirm mitotic phosphorylation at four of the nine reported Aurora sites (S5, S15, S55, and S69), a potential Aurora consensus site (S7), and a CDK-site (T31) in NDC80 (Fig. S3 A; Guimaraes et al., 2008; Kucharski et al., 2022). When compared with the parental cells, Western blots revealed more NDC80 phosphorylation at S55 and an upshift of NDC80 in Phos-tag gels indicative of increased overall phosphorylation of NDC80 in PPP6C KO (Fig. 4 B). The NDC80 phospho-antibodies used here reacted exclusively with phospho-specific epitopes on NDC80 and reactivity was abolished by λ-phosphatase treatment (Fig. 4 B, λ-PPase). This treatment also removed the phosphorylated NDC80 species detected on Phos-tag gels. In PPP6C KO cells, the pT288 activating phosphorylation on Aurora A, but not the equivalent pT232 for Aurora B, was elevated compared with the parental cells, suggesting the increase in NDC80 pS55 was due to Aurora A (Fig. 4 B). In support of this interpretation, phosphatase assays showed that NDC80 was dephosphorylated with the same kinetics in extracts from parental and PPP6C KO cells (Fig 4 C). Therefore, PP6 did not appear to be the NDC80 phosphatase, and we concluded that increased NDC80 phosphorylation in PPP6C KO cells was due to the elevated level of Aurora A activity rather than decreased dephosphorylation.

### Aurora A is the PP6-regulated kinase phosphorylating the N-terminus of NDC80

To determine if elevated NDC80 phosphorylation in PPP6C KO HeLa cells was due to Aurora A, combined Western blotting and mass spectrometry analysis of mitotic cells treated with either Aurora A or Aurora B inhibitors was performed. Together, these approaches revealed reduced NDC80 phosphorylation following Aurora A inhibition at pS55 (Fig. 4 D and Fig. S3, B–D), pS44, pT61-pS62, and pS69 in parental cells (Fig. S3, C and D). In PPP6C KO cells, similar results were obtained; however, due to the increased steady-state level of phosphorylation caused by elevated Aurora A activity, NDC80 pS55 was reduced but not fully dephosphorylated (Fig. 4 D). Aurora B inhibition with two

different compounds did not result in reduction of NDC80 phosphorylation at any of these sites (Fig. 4 D and Fig. S3, B–D). When Aurora A and Aurora B inhibitors were combined, NDC80 phosphorylation was reduced to the same level as Aurora A inhibitor alone (Fig. 4 D and Fig. S3 D), suggesting they do not play additive roles. Examination of the respective T-loop phosphorylation sites on Aurora A and Aurora B confirmed the selectivity and efficacy of the different Aurora inhibitors under the conditions used here (Fig. 4 D and Fig. S3 D). Phosphorylation of the CDK consensus site at T31 was not altered by either Aurora A or Aurora B inhibitors (Fig. S3 C), further supporting the view that the inhibitors are specifically targeting the relevant kinases. Importantly, all the phospho-antibodies used here for Western blotting are selective for the specific sites on NDC80 (Fig. S3 E).

These biochemical findings were confirmed using microscopy analysis, which showed NDC80 pS55, pS44, and pS62 staining at kinetochores of prometaphase cells with monoastral spindles (Fig. 4 E and Fig. S4 A). These phospho-antibody signals were lost in cells depleted of NDC80, confirming that they depend on the presence of NDC80 (Fig. S4, A and B). In agreement with the Western blot and mass spectrometry analysis, the NDC80 pS55 signal was also strongly reduced in cells treated with Aurora A inhibitors but not Aurora B inhibitors (Fig. 4 E). Importantly, the total level of NDC80 at kinetochores and levels of the centromere protein CENP-C were not altered (Fig. 4 E). In support of the idea that NDC80 phosphorylation is being carried out by Aurora A, depletion of either the Aurora A activator TPX2 or Aurora A itself, but importantly not Aurora B, resulted in a reduction of NDC80 pS55, pS44, and pS62 staining at kinetochores (Fig. 4, F and G; Fig. S4 C; Fig. 4, I and J; and Fig. S4 D) and smaller monoastral spindles (Fig. 4, H and K).

These data show that under our experimental conditions, NDC80 is phosphorylated at multiple sites in the N-terminus by Aurora A but not Aurora B. This provides an explanation for the increased level of NDC80 phosphorylation in PPP6C KO cells that have increased Aurora A activity. This effect appears to be a specific function carried out by Aurora A, discrete from Aurora B, and is consistent with global phospho-proteomic mapping of substrates for these two kinases, which also identified NDC80 as an Aurora A–specific substrate (Kettenbach et al., 2011).

### Aurora dependence of spindle assembly checkpoint signaling in PPP6C KO cells

Spindle checkpoint signaling is normally dependent on Aurora B rather than Aurora A (Ditchfield et al., 2003; Santaguida et al.,

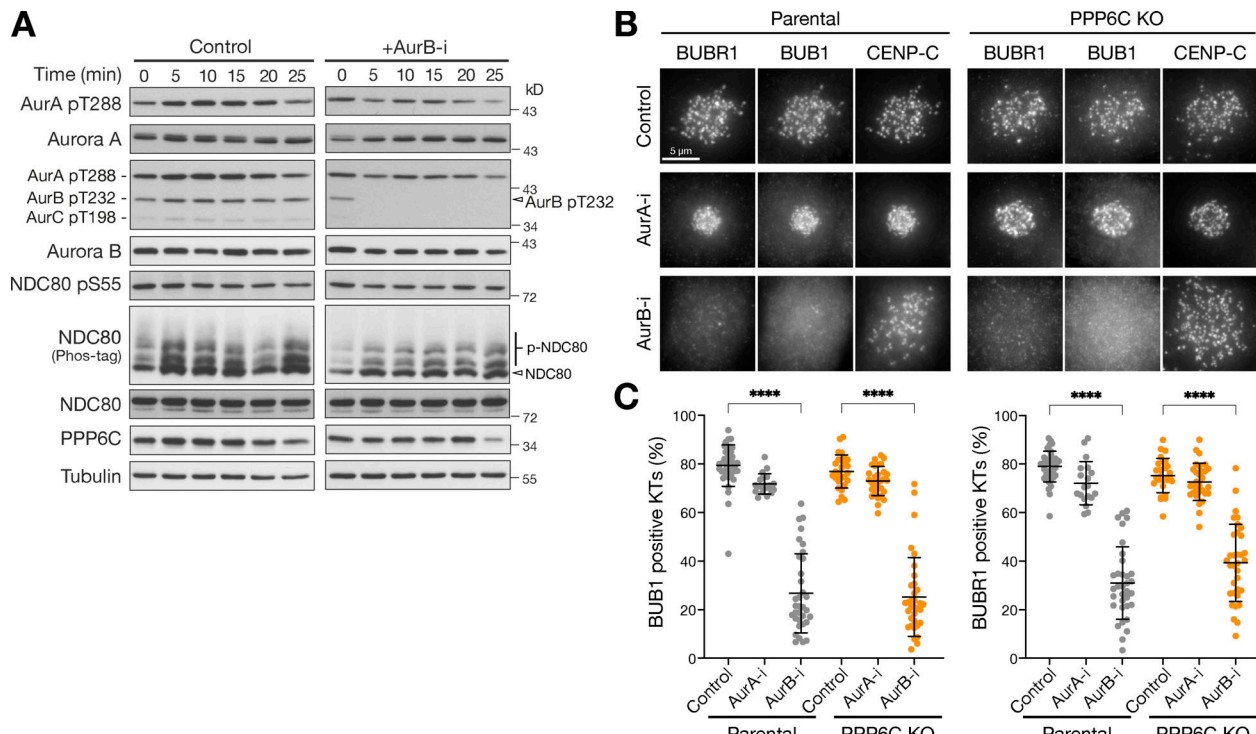

Figure 5. **Spindle checkpoint signaling remains Aurora B dependent in PPP6C KO cells. (A)** HeLa cells were treated with Aurora B inhibitor (AurB-i) and lysed at the times indicated from 0 to 25 min. Samples were blotted with the pan-Aurora T-loop antibody that detects active Aurora A/B/C. **(B)** Parental and PPP6C KO HeLa cells treated with Aurora A (AurA-i) or Aurora B inhibitors (AurB-i) for 30 min were stained for the spindle checkpoint proteins BUB1 and BUBR1. **(C)** Graphs that show the number of BUB1- or BUBR1-positive checkpoint active kinetochores are significantly reduced after Aurora A but not Aurora B inhibition (mean ± SD; n = 19–40). Statistical significance was analyzed using Dunn's multiple comparison test (****, P < 0.0001). Source data are available for this figure: SourceData F5.

2011), and we next sought to test if the relationship seen in normal cells remains in place in PPP6C KO cells. First, we confirmed that Aurora B was fully inhibited under the conditions we have used here. To do this, we carried out a biochemical time course analysis, which showed rapid loss of the Aurora B pT232 activating phosphorylation within 5 min of Aurora B inhibition, with little effect on the equivalent pT288 site on Aurora A (Fig. 5 A). Microscopy analysis revealed this was matched by loss of Aurora B–dependent checkpoint signaling components BUB1 and BUBR1 from kinetochores in both parental and PPP6C KO cells after 30 min Aurora B inhibition (Fig. 5, B and C). Crucially, Aurora A inhibition had no effect on the recruitment of checkpoint proteins to kinetochores (Fig. 5, B and C). Similar results were obtained using siRNA depletion of Aurora A and Aurora B. Localization of the checkpoint proteins at kinetochores was lost exclusively in cells depleted of Aurora B (Fig. S4, E and F); conversely kinetochore staining for NDC80 pS44, pS55, and pS62 was lost only in cells depleted of Aurora A (Fig. 4, I and J and Fig. S4 D).

Therefore, spindle assembly checkpoint signaling remained Aurora B sensitive and was insensitive to Aurora A inhibition in both parental and PPP6C KO HeLa cells. This appears to eliminate the possibility that amplified Aurora A in PPP6C KO cells was driving processes at the kinetochore and centromeres normally regulated by Aurora B, and is in agreement with the idea that Aurora A and Aurora B must have distinct targets in

line with their distinct inhibition phenotypes (Ditchfield et al., 2003; Hauf et al., 2003; Hoar et al., 2007).

## NDC80 is phosphorylated at microtubule-attached kinetochores from prometaphase to anaphase

To better understand the role of NDC80 phosphorylation by Aurora A and its dysregulation in PPP6C KO cells, we asked when the phosphorylation takes place and how long it persists in mitosis. We first looked at the localization of Aurora A at different stages of mitosis. Confirming previous findings (Zeng et al., 2010), the active, pT288-positive, TPX2-dependent pool of Aurora A localized to spindle fibers during metaphase and anaphase cells in parental cells (Fig. 6 A, Parental). PPP6C KO resulted in the spread of both TPX2 and Aurora A pT288 along the spindle in metaphase and increased levels of Aurora A pT288 in anaphase (Fig. 6 A, PPP6C KO arrowheads). Biochemical analysis of NDC80 phosphorylation using specific antibodies to pS55 as well as Phos-tag gels was then performed. This approach showed that NDC80 phosphorylation was present from prometaphase and metaphase into anaphase in parental cells (Fig. 6 B, Parental). Both NDC80 pS55 blots and Phos-tag gels showed NDC80 phosphorylation was increased in PPP6C KO cells and extended later into anaphase and telophase (Fig. 6 B, PPP6C KO). Cyclin B destruction and removal of the inhibitory pT320 CDK-phosphorylation on PP1, both hallmarks of mitotic exit, showed similar kinetics in both parental and PPP6C KO cells (Fig. 6 B).

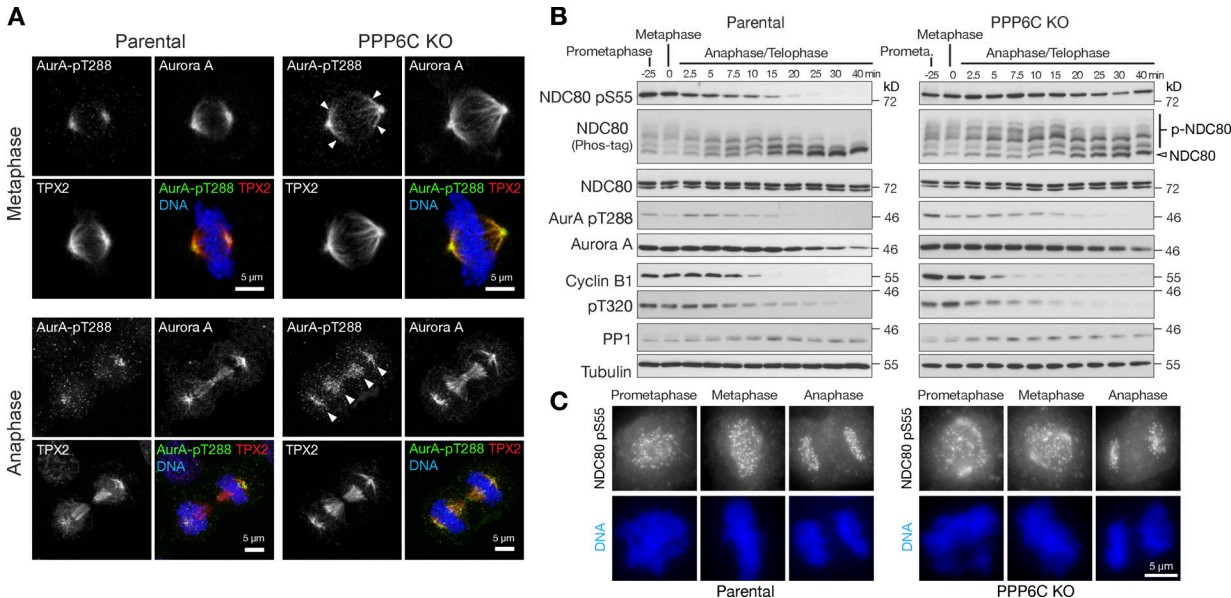

Figure 6. **Timing of NDC80 phosphorylation during mitosis and mitotic exit. (A)** Parental and PPP6C KO HeLa cells stained for Aurora A, the Aurora A activating protein TPX2, active Aurora A pT288, and DNA. Examples of metaphase and anaphase cells are shown. Arrowheads indicate the spread of active Aurora A on metaphase and anaphase spindles in PPP6C KO cells. Note the enlarged metaphase plate in PPP6C KO cells. **(B)** Synchronized parental and PPP6C KO HeLa cells in mitosis were treated with CDK inhibitor to promote entry into anaphase and mitotic exit. NDC80 phosphorylation was followed using NDC80 pS55 and Phos-tag gels. Mitotic exit was confirmed by blotting for cyclin B, the inhibitory pT320 modification of PP1, and active Aurora A pT288. **(C)** Parental and PPP6C KO HeLa cells stained for NDC80 pS55 and DNA. Examples of prometaphase, metaphase, and anaphase cells are shown corresponding to the conditions in B. Source data are available for this figure: SourceData F6.

Microscopy confirmed the presence of NDC80 pS55 at kinetochores in prometaphase, metaphase, and anaphase cells in both parental and PPP6C KO cells (Fig. 6 C).

To more precisely determine the relationship between NDC80 phosphorylation and microtubule-attachment state, cells were treated with STLC to create monopolar spindles with a mixture of astrin-positive microtubule-attached and astrin-negative checkpoint-positive microtubule-free kinetochores were analyzed. When stained with antibodies against the astrin–kinastrin complex, a marker of microtubule-attached kinetochores (Schmidt et al., 2010), NDC80 pS55 and pS62 were present at the subset of kinetochores positive for astrin in these mono-astral spindles (Fig. 7 A). This suggests that NDC80 phosphorylation occurred during or after microtubule attachment to the kinetochore. To explore this idea, we examined the localization of Aurora A and Aurora B relative to kinetochores during mitosis. Aurora A localized to mitotic spindle fibers that terminate at the kinetochores in both metaphase and anaphase cells (Fig. 7, B and C; Bird and Hyman, 2008). By contrast, yet as expected, Aurora B was present on the centromeres in metaphase and central spindle in anaphase, clearly resolved from Aurora A and spatially separated from kinetochores (Fig. 7, D and E; Carmena et al., 2012). Based on its localization to spindle fibers contacting the kinetochore, Aurora A is thus more likely to be the kinase phosphorylating NDC80 at microtubule-attached kinetochores in prometaphase, metaphase, and anaphase. This idea is also consistent with our biochemical data. Together, these observations show that NDC80 phosphorylation is dynamically modulated at microtubule-attached kinetochores rather than at unaligned chromosomes (Fig. 7 F). This proposal implies that

NDC80 phosphorylation should not occur at checkpoint-active kinetochores which lack stable microtubule attachments and requires the presence of spindle microtubules. To test these ideas, we explored the relationship between NDC80 phosphorylation, spindle microtubules, and Aurora B–dependent spindle checkpoint signaling.

## NDC80 phosphorylation occurs at checkpoint-silenced microtubule-attached kinetochores

Cells were either arrested in mitosis with nocodazole in a prometaphase-like state where microtubules are depolymerized and all the kinetochores are checkpoint active or with MG132 at a metaphase plate stage where microtubules and attachment to kinetochores remain intact and all kinetochores are checkpoint silenced. As expected, the spindle checkpoint kinase MPS1 localized to kinetochores in nocodazole (Fig. 8 A, +Noc) but not MG132-treated cells, (Fig. 8 A, +MG132). The kinetochore signal for MPS1 was lost when Aurora B was inhibited in nocodazole-treated cells (Fig. 8 A, +Noc +AurB-i). There was no kinetochore signal for NDC80 pS55 in the nocodazole-arrested conditions (Fig. 8 A, +Noc), whereas all kinetochores were positive for NDC80 pS55 and negative for MPS1 in checkpoint-silenced MG132-arrested cells (Fig. 8 A, +MG132). In both nocodazole and MG132-arrested cells, Aurora B inhibition resulted in loss of the pT232 epitope marking active Aurora B but had no effect on NDC80 pS55 (Fig. 8 A, +AurB-i).

We then examined the relationship between spindle assembly checkpoint signaling and NDC80 phosphorylation in cells using monopolar spindle assays. Under these conditions depicted in the cartoon shown in the figure, a clear inverse

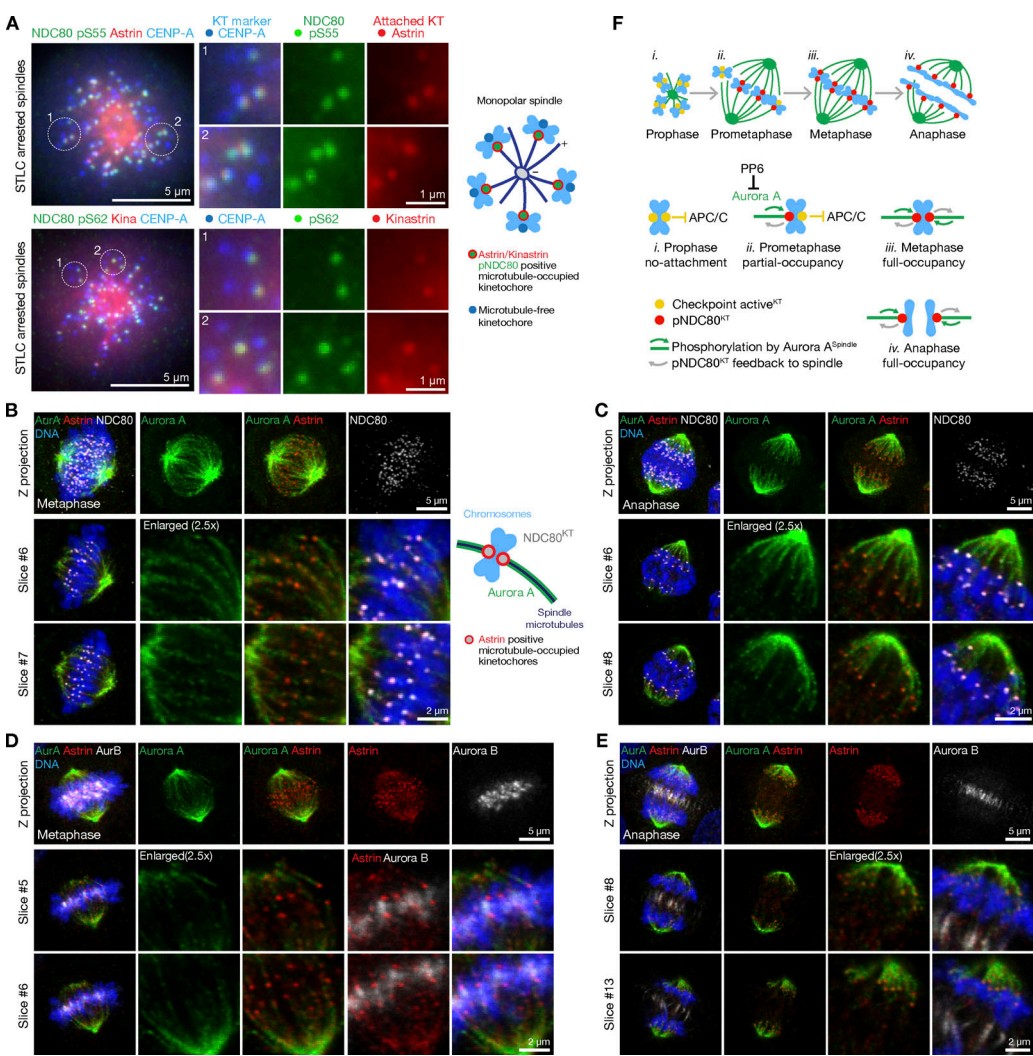

**Figure 7. NDC80 is phosphorylated by Aurora A at microtubule-attached kinetochores. (A)** Hela cells arrested with STLC were stained for NDC80 pS55, astrin, CENP-A, and DNA or NDC80 pS62, kinastrin, CENP-A, and DNA. Enlarged region and schematics show the relationship of NDC80 pS55 or pS62 and astrin or kinastrin staining relative to the centromere and kinetochore marker CENP-A. **(B and C)** Confocal images of metaphase (B) or anaphase (C) HeLa cells stained for Aurora A, astrin, NDC80, and DNA. Enlarged regions show the proximity of astrin-positive kinetochores to Aurora A spindle fibers in metaphase and anaphase. **(D and E)** Confocal images of metaphase (D) or anaphase (E) HeLa cells stained for Aurora B, Aurora A, astrin, and DNA. Enlarged regions show the proximity of astrin-positive kinetochores to Aurora A spindle fibers in both metaphase and anaphase. In metaphase, Aurora B flanks NDC80 and astrin-positive kinetochores. In anaphase, Aurora B relocates to the central spindle and spatially segregated from NDC80 and astrin-positive kinetochores. **(F)** Schematic showing the relationship of Aurora A bearing spindle fibers (green, green arrow), NDC80 phosphorylation (red), and spindle checkpoint signaling (yellow) at kinetochores during different stages of spindle formation. PP6 limits Aurora A activity toward NDC80. NDC80 phosphorylation alters the properties of kinetochores and feeds back on to the spindle microtubules (gray arrow).

relationship between the presence of the checkpoint protein MAD1 and NDC80 phosphorylation at pS55 was observed (Fig. 8 B). NDC80 pS55 was only detected at MAD1 negative kinetochores in both parental and PPP6C KO cells (Fig. 8, B and C). Consistent with the biochemical data already shown (Fig. 4 B), the NDC80 pS55 signal was significantly elevated at MAD1 negative kinetochores in PPP6C KO cells (Fig. 8 C); however the frequency of MAD1 positive kinetochores was not altered (Fig. 8 D).

NDC80 phosphorylation by Aurora A has also been associated with the congression of chromosomes to the metaphase plate during pole-based error correction (Ye et al., 2015). However, in that study, Aurora A inhibitor-sensitive phosphorylation of

aligned chromosomes at the metaphase plate was also observed, similar to our observations. To investigate this, we used CENP-E inhibition (Qian et al., 2010) to trap a subset of chromosomes at spindle poles (Fig. S5 A) and performed an analysis of NDC80 pS55 signal at kinetochores on chromosomes aligned to the forming metaphase plate and those trapped at spindle poles. This approach showed that the level of NDC80 pS55 was independent of chromosome position in both parental and PPP6C KO cells, but remained sensitive to Aurora A in all cases (Fig. S5 B). Therefore, although we can confirm that Aurora A phosphorylates NDC80 at uncongressed chromosomes during pole-based error correction (Ye et al., 2015), we find this is maintained on chromosomes aligned to the metaphase plate. This finding is

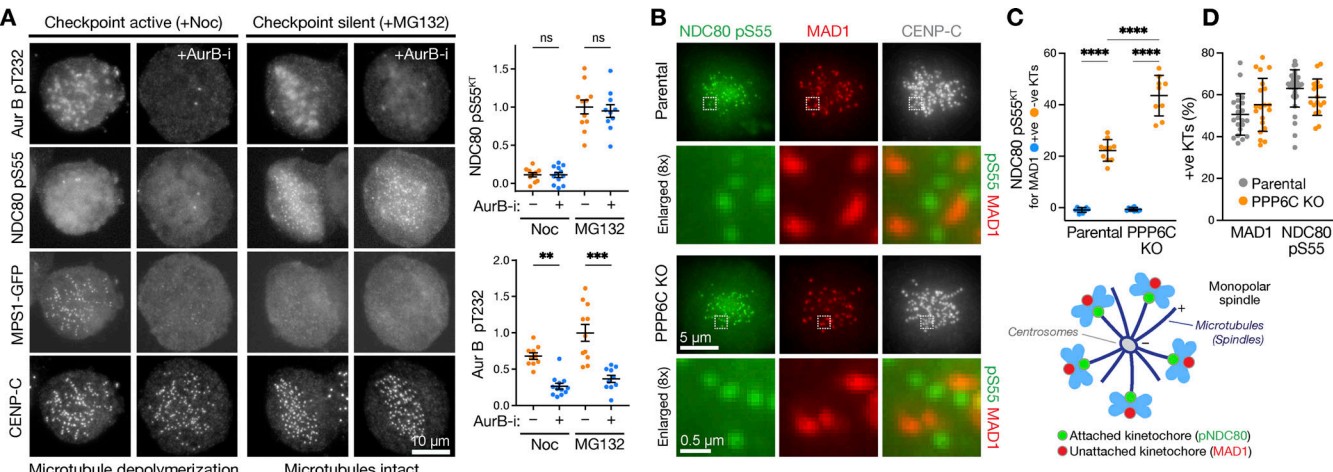

Figure 8. **NDC80 phosphorylation and spindle checkpoint signaling are mutually exclusive. (A)** HeLa MPS1-GFP[CRISPR] cells arrested in mitosis with either nocodazole or the proteasome inhibitor MG132 were left untreated or treated with Aurora B inhibitor (AurB-i). Cells were stained for active Aurora B pT232, phosphorylated NDC80 pS55, and the kinetochore marker CENP-C. The intensities of NDC80 pS55 signal at kinetochores and Aurora B pT232 signal are shown for the different conditions (mean ± SEM; n = 10–12; **, P < 0.01; ***, P < 0.001). **(B)** Parental and PPP6C KO HeLa cells arrested with STLC to create monoastral spindles with a mixture of checkpoint active and silent kinetochores states were stained for MAD1, NDC80 pS55, CENP-C, and DNA. **(C and D)** The intensity of NDC80 pS55 signal was measured at MAD1 positive (+ve) and negative (−ve) kinetochores (C; mean ± SD; n = 9–10), and the proportion of MAD1 or NDC80 pS55 positive kinetochores (D; mean ± SD; n = 17–34) was determined for parental and PPP6C KO HeLa cells. Statistical significance was analyzed using a Brown-Forsythe ANOVA (****, P < 0.0001).

consistent with data in the main figures or supplemental data in other studies that also identify NDC80 pS55 on the kinetochores of aligned metaphase chromosomes (Courtois et al., 2021; DeLuca et al., 2011; Posch et al., 2010; Schleicher et al., 2017; Suzuki et al., 2014). One study also reported NDC80 pS15, pS44, and pS55 staining in metaphase persisted at the same level into anaphase (DeLuca et al., 2011), suggesting this is true for multiple sites in the NDC80 N-terminus rather than representing unique behavior of an individual site.

Together, our data show that NDC80 phosphorylation is mediated by Aurora A at microtubule-attached kinetochores and is not associated with Aurora B activity. NDC80 phosphorylation although dependent on microtubule attachment is independent of chromosome position and inversely correlated with spindle checkpoint signaling. These data argue strongly against the possibility that amplified Aurora A activity in PPP6C KO cells takes over processes such as checkpoint signaling normally controlled by Aurora B. The Aurora–TPX2 dependent increase of spindle size in PPP6C KO cells suggested that NDC80 phosphorylation plays an important role during mitotic spindle size control and hence this was examined further.

**Multisite phosphorylation of NDC80 by Aurora A is counteracted by PP1/PP2A**

As a next step, we sought to understand the stoichiometry of NDC80 phosphorylation at kinetochores in parental and PPP6C KO cells. Since PP6 is not the NDC80 phosphatase (Fig. 4 C), we first asked what role the other major mitotic phosphatases PP1 and PP2A play in counteracting Aurora A. Both the number and intensity of NDC80 pS55 positive kinetochores increased following PP1/PP2A inhibition with the potent PP1 and PP2A inhibitor calyculin A (Fig. 9, A–C, Parental). In PPP6C KO cells, NDC80 pS55 was higher than in the parental control cells and

this increased further with PP1/PP2A inhibition (Fig. 9, A–C, PPP6C KO). These results show that under normal conditions NDC80 phosphorylation at kinetochores is substoichiometric, suggesting Aurora A activity being continuously counteracted by, and hence acts upstream of PP1/PP2A. PP1 and PP2A inhibition caused a complete upshift of NDC80 on Phos-tag gels and a large increase in NDC80 pS55 in both parental and PPP6C KO cells (Fig. 9, D and E). Both the upshift of NDC80 to a hyperphosphorylated state on Phos-tag gels and an increase in NDC80 pS55 were prevented by prior addition of Aurora A but not Aurora B inhibitors (Fig. 9 D). PP1/PP2A activity, therefore, counteracts Aurora A activity and maintains NDC80 in a predominantly hypophosphorylated or dephosphorylated state. Confirming this relationship, NDC80 remained phosphorylated when Aurora A inhibitor was added after the PP1/PP2A inhibitor (Fig. 9 E). Based on these observations, we conclude that NDC80 does not appear to undergo full stoichiometric phosphorylation at all Aurora sites under normal conditions, consistent with our own data (Fig. S3) and a recently published mass spectrometric analysis of NDC80 phosphorylation (Kucharski et al., 2022).

To test for roles of individual phosphorylation sites, we created phosphorylation-deficient point mutant forms of NDC80 for sites to which we had specific phospho-antibodies and a combined NDC80-9A mutant deficient for all phosphorylation. Endogenous NDC80 was replaced by expression of NDC80-GFP constructs in cells depleted with an siRNA targeting the 5′-UTR of the NDC80 messenger RNA (Fig. S4, A and B), an approach developed previously by others (Nijenhuis et al., 2013). The NDC80-GFP expressing cells were then arrested with STLC to trap them in mitosis with monopolar spindles, where we would expect to see phosphorylated kinetochores. To confirm expression of the mutant constructs, cells were then stained with a

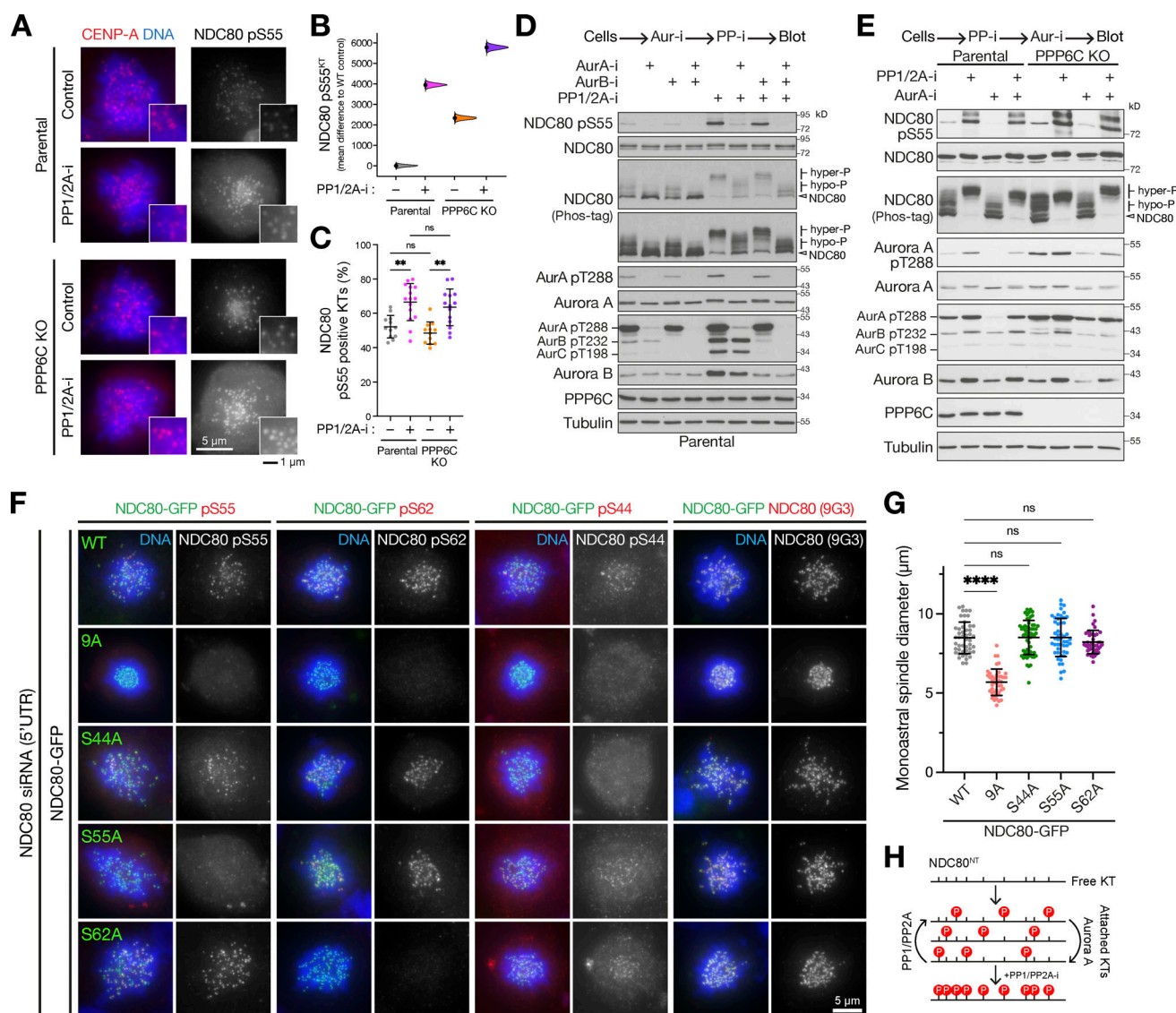

Figure 9. **Multisite phosphorylation of NDC80 by Aurora A is counteracted by PP1/PP2A. (A)** Parental and PPP6C KO cell lines were treated with STLC to arrest cells in mitosis in the absence (Control) and presence of PP1/2A-i and then stained for NDC80 pS55, CENP-A, and DNA. **(B)** Level of NDC80 pS55 signal at kinetochores (KTs) was measured for the different conditions (n = 701–951 KTs). Each mean difference is depicted as a dot. Each 95% confidence interval is indicated by the ends of the vertical error bars; the confidence interval is bias-corrected and accelerated. **(C)** The number of NDC80 pS55 positive kinetochores was measured for the different conditions (mean ± SD; n = 12–15). Statistical significance was analyzed using a Brown-Forsythe ANOVA (**, P < 0.01). **(D and E)** Mitotic lysates of parental and PPP6C KO HeLa cells treated with Aurora A (AurA-i), Aurora B, and phosphatase (calyculin, PP1/2A-i) inhibitors in the combinations and order shown were blotted for the proteins listed in the figure. Overall NDC80 phosphorylation was monitored using a Phos-tag gel. **(F)** HeLa cells depleted of endogenous NDC80 using a 5′-UTR siRNA were transfected with NDC80-GFP WT and phospho-deficient mutant constructs as shown in the figure. Cells were stained for DNA, NDC80, and specific NDC80 phospho-antibodies. **(G)** Spindle size is plotted for the NDC80 WT and point mutant in panel F (mean ± SD; n = 42–58). Statistical significance was analyzed using a Brown-Forsythe ANOVA (****, P < 0.0001). **(H)** A schematic showing the proposed dynamic substoichiometric phosphorylation of the NDC80 N-terminus, and roles of Aurora A and PP1/PP2A. Source data are available for this figure: SourceData F9.

panel of NDC80 phospho-antibodies. In all cases, NDC80-GFP was localized to the kinetochores. None of the antibodies detected the NDC80-9A mutant and the specific kinetochore signal was lost when the phospho-antibody used corresponded to the specific point mutant (Fig. 9 F). For example, the S55A construct did not react with pS55 antibodies but was detected by pS44 and pS62. Likewise, on Phos-tag gels, only the NDC80-9A mutant failed to show an upshift (Fig. S3 E), indicating that the phosphorylation events are unlikely to be interdependent. When

performing these experiments, we noticed that monoastral spindle size was reduced in cells expressing the NDC80-9A mutant and this was found to be significant when measured, P < 0.0001, whereas single point mutants showed no significant difference in spindle size (Fig. 9 G). Previous work has also observed similar clustering of kinetochores around monopolar spindle poles in cells expressing NDC80-9A mutants (Etemad et al., 2015), consistent with the idea that Aurora phosphorylation of NDC80 is crucial for spindle size control.

Based on these observations, we conclude that NDC80 does not appear to undergo stoichiometric phosphorylation at all Aurora sites under normal conditions and exists in a mixture of hypophosphorylated states (Fig. 9 H) as others have also recently proposed (Kucharski et al., 2022). Our data is most consistent with a model where NDC80 phosphorylation is highly dynamic and dependent on PP6-regulated Aurora A–TPX2 activity and actively limited by counteracting NDC80 phosphatases that are discrete from PP6, most likely PP1 and PP2A. This reaction favors phosphorylation on microtubule attachment to the kinetochore due to the proximity of Aurora A–TPX2 on spindle fibers, whereas dephosphorylation predominates upon microtubule release.

### NDC80 phosphorylation regulates mitotic spindle size and chromosome segregation

Finally, we asked if altered NDC80 phosphorylation can explain the differences in spindle size and chromosome segregation leading to micronucleation and nuclear morphology defects in PPP6C KO cells (Fig. 1; Hammond et al., 2013; Zeng et al., 2010). For this purpose, cell lines conditionally expressing either WT NDC80-GFP (NDC80 WT), phosphorylation deficient (NDC80-9A), or phospho-mimetic (NDC80-9D) variants in which nine Aurora kinase consensus sites had been mutated (DeLuca et al., 2011) were used. In a parental background with WT PPP6C, expression of NDC80-9A or NDC80-9D resulted in smaller or larger metaphase spindles, respectively, compared with the WT NDC80 protein (Fig. 10, A and B). Similarly, monoastral spindles were smaller or larger in parental cells expressing NDC80-9A or NDC80-9D (Fig. 10, C and D), respectively. NDC80-9A was also dominant over PPP6C depletion for monoastral spindle size (Fig. 10, C and D), and furthermore, these spindles showed robust astrin localization to kinetochores suggesting formation of stable microtubule attachments (Fig. 10, C and E). Spindle size was the same in NDC80-9A and NDC80 WT cells treated with Aurora A inhibitor, and Aurora A inhibition was not additive with NDC80-9A expression (Fig. 10, F and G). These data supported the conclusion that the elevated NDC80 phosphorylation by Aurora A in PPP6C-depleted cells is responsible for the increased spindle size and consequent chromosome missegregation.

To further test this idea, we tested if NDC80-9A can suppress nuclear morphology defects in cells lacking PP6. When compared with expression of the WT protein, NDC80-9A significantly reduced the frequency of nuclear shape defects in PPP6C-depleted cells (Fig. 10, H and I). In contrast, NDC80-9D strongly exacerbated the nuclear shape defects (Fig. 10, H and I). This latter observation is in agreement with the observation that PPP6C KO results in a partial increase in NDC80 phosphorylation, but does not create the fully stoichiometrically phosphorylated state equivalent to NDC80-9D seen with PP1/PP2A inhibition.

In summary, we propose a model for spindle size control whereby PP6 limits the activity of Aurora A–TPX2 complexes toward NDC80. Reduction in Aurora A activity using inhibitors or by removing its activator TPX2 reduces spindle size, is consistent with the effects of expressing a phosphorylation-deficient NDC80-9A mutant. Conversely, increased Aurora A activity in cells lacking its negative regulator PP6 increases spindle size, consistent with the effects of expressing a phospho-mimetic NDC80-9D mutant (Fig. 10 J).

## Discussion

### Mitotic spindle size control and genome instability

How the mitotic spindle ensures accurate chromosome segregation and the way this control is modified in aneuploid tumor cells are important questions. Here, we have used CRISPR genomics to identify the Aurora A activator TPX2 and kinetochore protein NDC80 as key components explaining the cancer-associated PPP6C loss-of-function phenotype. Aurora A and TPX2 are important factors for mitotic spindle formation and are limiting for spindle size. The availability of TPX2 is regulated by the Ran-Importin pathway (Carazo-Salas et al., 1999; Clarke and Zhang, 2008; Heald et al., 1996; Kalab et al., 1999). This pathway triggers release of TPX2 from importin in the vicinity of chromatin to nucleate microtubules and activate Aurora A (Gruss et al., 2001; Gruss et al., 2002; Kufer et al., 2002; Schatz et al., 2003). Our study and previous work show that cells expressing TPX2 mutants unable to bind Aurora A form short spindles, which are still capable of chromosome segregation (Bird and Hyman, 2008). Using a combination of functional genomics and biochemistry, we identified NDC80 as a crucial target for Aurora A during the spindle formation process. Cells expressing the NDC80-9A Aurora-phosphorylation deficient mutant form smaller spindles of similar morphology to cells expressing a TPX2 YYD/AAA mutant; compare Fig. 10 A and Fig. 3, B and F. Thus, chromosomes provide a signal to trigger microtubule nucleation and activate Aurora A–TPX2 complexes, which in turn phosphorylate NDC80 and promote spindle growth away from the minimum size supported by TPX2 mutants unable to bind Aurora A. The Aurora A phosphatase PP6 is crucial to limit the activity of Aurora A–TPX2 in this process, and hence plays an important role in spindle size control.

NDC80, the major microtubule-binding protein at the outer kinetochore, is the keystone protein for mitotic spindle formation (Cheeseman et al., 2006). When NDC80 is removed or inactivated, mitosis fails catastrophically (Chen et al., 1997; McCleland et al., 2003). This is due to its role in integration of chromosome attachment to microtubules and chromosome alignment with checkpoint signaling, and as we show here spindle size control. Our biochemical analysis provides evidence that NDC80 undergoes dynamic substochiometric phosphorylation due to the balance of Aurora A and PP1/PP2A activities at microtubule-attached and microtubule-free kinetochores. This is consistent with other reports showing a pattern of dynamic substoichiometric phosphorylation on NDC80 (Kucharski et al., 2022) and phosphorylation of NDC80 by Aurora A (DeLuca et al., 2018; Kettenbach et al., 2011). Removal of PP6 results in increased Aurora A activity at the spindle and an approximately two- to threefold increase in NDC80 phosphorylation. However, NDC80 phosphorylation remains substoichiometric in PPP6C KO. We could only observe stoichiometric or complete phosphorylation of NDC80 when PP1/PP2A was inhibited, suggesting this state is either highly transient or not achieved through normal regulation.

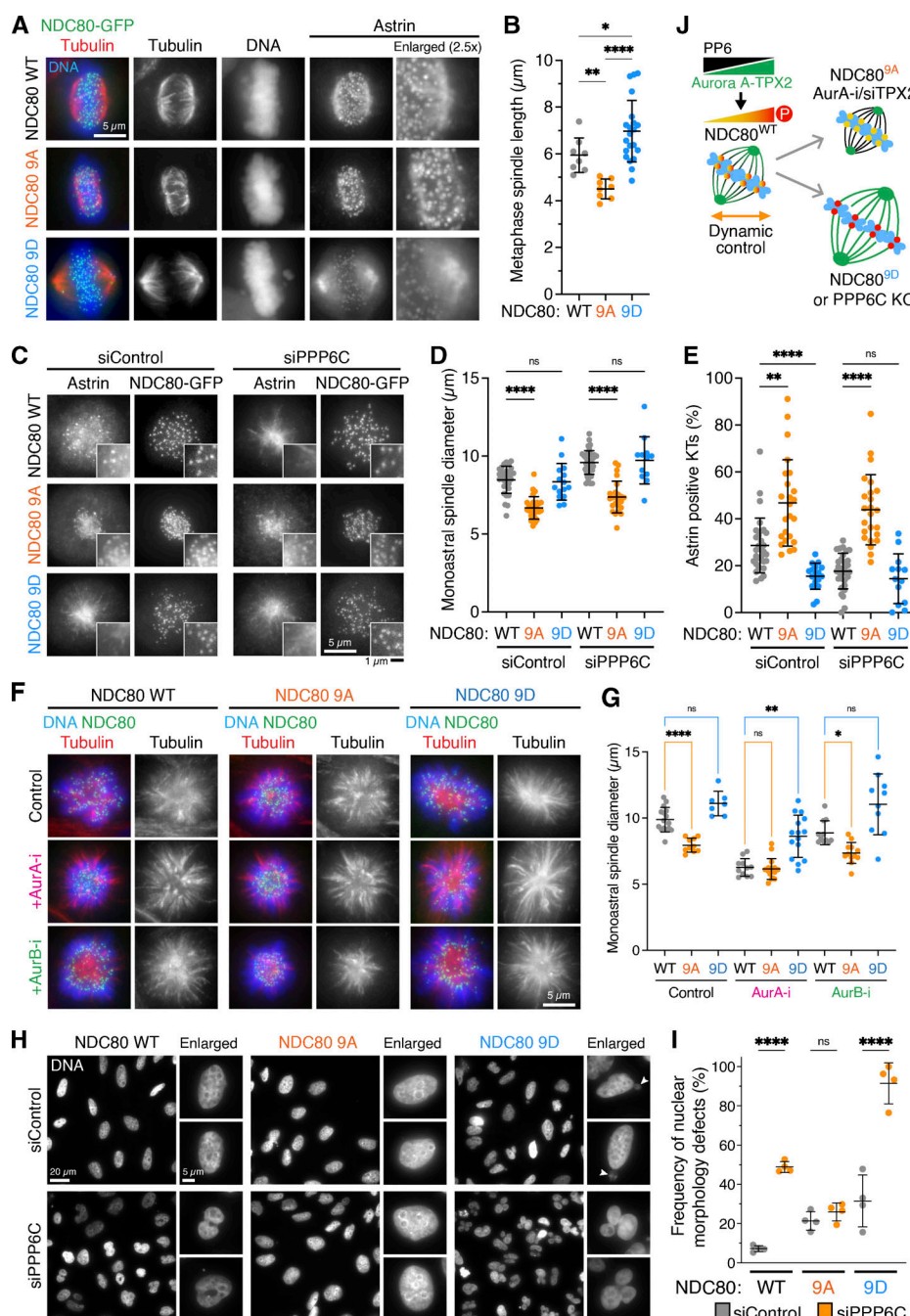

Figure 10. **NDC80 phospho-mutants rescue the spindle size and micronucleation defects seen in PP6-deficient cells. (A)** HeLa FlpIn T-REx cells expressing WT NDC80-GFP, phospho-deficient (9A), and phospho-mimetic (9D) were depleted of endogenous NDC80 for 72 h using a 5'-UTR siRNA and then stained for tubulin and DNA. NDC80-GFP was visualized directly. Enlarged regions show astrin staining at kinetochores in mitotic cells. **(B)** Metaphase spindle length (mean ± SD; n = 8–21) is plotted in the graph. Statistical significance was analyzed using a Brown-Forsythe ANOVA (*, P < 0.05; **, P < 0.01; ****, P < 0.0001). **(C)** HeLa FlpIn T-REx cells expressing NDC80-GFP WT, 9A, and 9D were transfected with control or PPP6C siRNA for 72 h, treated with STLC for 3 h to create monopolar spindles, and then stained for astrin. NDC80-GFP was visualized directly. Enlarged regions show the presence or loss of astrin staining at NDC80-GFP–labeled kinetochores. **(D and E)** Monoastral spindle diameter (D) and the number of astrin-positive kinetochores (E) were measured for the conditions shown in C (mean ± SD; n = 12–38). Statistical significance was analyzed using a Brown-Forsythe ANOVA (**, P < 0.01; ****, P < 0.0001). **(F)** NDC80-GFP (WT), 9A, and 9D expressing cells arrested with STLC for 3 h to create monopolar spindles were treated with Aurora A or Aurora B inhibitors for 30 min and then stained for tubulin and DNA. NDC80-GFP was visualized directly. **(G)** Monoastral spindle diameter was measured for the conditions shown in F (mean ± SD; n = 7–16). Statistical significance was analyzed using a Brown-Forsythe ANOVA (*, P < 0.05; **, P < 0.01; ****, P < 0.0001). **(H and I)** HeLa FlpIn T-REx NDC80-GFP WT, 9A, and 9D cells treated with control or PPP6C siRNA for 72 h were stained for DNA (H), and the frequency of cells with nuclear morphology defects scored (I; mean ± SD; n = 4). Statistical significance was analyzed using a one-way ANOVA (****, P < 0.0001). **(J)** A schematic depicting the effects of PP6-regulated Aurora A–TPX2 activity on NDC80 phosphorylation and spindle size is shown to the right. Aurora A inhibition (AurA-i), TPX2 depletion (siTPX2), and NDC80-9A expression result in reduced spindle size, whereas Aurora A activity amplification in PPP6C KO and NDC80-9D expression both increase spindle size.

By studying specific point mutants of NDC80, we find that it is a key regulatory target for the PP6-regulated pool of Aurora A–TPX2 at mitotic spindles. This Aurora A–dependent regulation serves to minimize spindle size and ensure a highly compacted metaphase plate and maintenance of a compressed body of segregating chromosomes in anaphase. This tight compaction is important for the formation of a single nucleus in G1. Accordingly, removal of PPP6C or expression of the NDC80-9D phospho-mimetic mutant results in micronucleation and other structural defects in the nuclei. Conversely, expression of the NDC80-9A phospho-deficient mutant results in smaller more compact spindles and suppresses nuclear morphology defects in PPP6C KO cells. This supports the view that NDC80 is a key target for Aurora A during spindle formation and provides an additional spindle scaling mechanism complementing the chromatin-dependent Ran–Importin regulation of TPX2. Although it is widely accepted that the mitotic spindle scales with the number of chromosomes, the existence and purpose of such a scaling mechanism in somatic cells are counterintuitive since it would seem to benefit aneuploid tumor cells with increased numbers of chromosomes rather than normal diploid cells. Spindle size control must therefore serve another purpose in somatic cells. We propose that one such purpose is to ensure that chromosomes remain as compact masses and are captured into a single nucleus in each newly forming cell during mitotic exit, and it is interesting to note that the depletion of factors such as the barrier-to-autointegration factor required to crossbridge chromosomes in mitotic exit results in nuclear morphology defects similar to those seen here following PPP6C KO or PPP6C depletion (Samwer et al., 2017). When such mechanisms fail, as in the case of PPP6C mutation, the resulting DNA damage provides a potential driver mechanism for tumor development (Hammond et al., 2013).

## Aurora A and Aurora B function in spindle formation and checkpoint signaling

NDC80 phosphorylation has mainly been ascribed to Aurora B–mediated error correction and spindle checkpoint signaling in the absence of tension (Lampson and Cheeseman, 2011; Liu et al., 2009; Musacchio, 2015). In vitro biochemistry has shown that stoichiometric phosphorylation of NDC80 by Aurora B reduces its affinity for microtubules and has been suggested to create a switch-like mechanism to control microtubule binding at incorrectly attached kinetochores in cells (Alushin et al., 2010; Cheeseman et al., 2006; Ciferri et al., 2008; DeLuca et al., 2006). In contrast to these ideas, we find that Aurora A is the major kinase phosphorylating NDC80 at microtubule-attached kinetochores during spindle formation. Both the temporal distribution of NDC80 phosphorylation from prometaphase into anaphase, and association with microtubule-attached, rather than checkpoint-active, kinetochores argue against functions in Aurora B–mediated error correction and spindle checkpoint signaling. Furthermore, we demonstrate that phosphoregulation of NDC80 at microtubule-attached kinetochores by the PP6-regulated pool of Aurora A is necessary to control metaphase spindle size rather than to sustain checkpoint signaling. Other researchers have also reported that NDC80 phosphorylation at

multiples sites can be observed in metaphase and persists at the same level into anaphase (DeLuca et al., 2011). This is consistent with our data showing that Aurora A remains associated with spindle fibers contacting the kinetochores in anaphase, whereas Aurora B relocates to the central spindle (Fig. 7, B–E), and supports the view NDC80 is an Aurora A–specific target. Further work will be needed to explain precisely how NDC80 phosphorylation modulates its function, but clues are already provided in other published work showing that phosphorylation of NDC80 modulates its interaction with microtubules and balance of forces within the spindle, promoting oscillatory chromosome movements linked to Aurora A activity (DeLuca et al., 2018; Iemura et al., 2021; Long et al., 2017; Umbreit et al., 2012). A recent report has suggested an important function of the astrin–kinastrin complex is enabling kinetochore mobility and increasing depolymerization velocity of kinetochore fibers (Rosas-Salvans et al., 2022). Our data suggest that the NDC80 phosphorylation status affects the level of astrin at kinetochores, with less astrin found at the hyperphosphorylated kinetochores of cells lacking PP6 (Fig. 10 C). This finding may explain the abnormal behavior of the microtubule–kinetochore interface in these cells, since reduced astrin levels would result in decreased kinetochore fiber depolymerization and hence longer kinetochore fibers, as we observe. Thus, we propose that PP6 limits Aurora A activity at microtubule-attached kinetochores to fine-tune their properties during spindle formation, possibly by regulating the levels of the astrin–kinastrin complex, and that it is this feedback from kinetochores to the microtubules that controls spindle size. This regulation ensures that chromosome segregation leads to the formation of equal nuclei lacking structural defects.

Aurora B activity is necessary for MPS1-recruitment and spindle checkpoint signaling, even in cells expressing NDC80 mutants lacking all the N-terminal Aurora sites (Hayward et al., 2022; Nijenhuis et al., 2013). This relationship between Aurora B and checkpoint signaling remains in force even in PPP6C KO cells with amplified Aurora A activity. As we show here, PPP6C KO cells still depend on Aurora B but not Aurora A for checkpoint signaling. How then do we explain some of the differences between our observations and other studies? Previous work has largely relied on the use of NDC80 phospho-mimetic and phospho-null mutants. However, due to the similar kinase consensus motifs for Aurora A and B, it is necessary to confirm which kinase acts at the phospho-sites in vivo under the specific conditions being studied. Additionally, common kinase inhibitors are often not completely specific, and this is especially problematic when dealing with closely related kinases such as Aurora A and Aurora B. It is therefore important that inhibitors to both kinases are carefully titrated and tested in parallel to confirm specific inhibition. In some studies, Förster resonance energy transfer sensors for Aurora B have been used to show that its activity at the centromere is sensitive to microtubule attachment at kinetochores and correlated with the effects of expressing mutant NDC80 proteins (Yoo et al., 2018). However, this does not prove NDC80 is the crucial target for Aurora B, and with such approaches, it is important to directly confirm when and where,

and to what level NDC80 is phosphorylated, as we have done here with biochemical- and microscopy-based approaches. We also find that Aurora B inhibition leads to more scattered chromosomes in monopolar spindle assays, whereas both Aurora A inhibition and expression of NDC80-9A result in compact monopoles. Some of the effects of Aurora B inhibition may be indirect through its role in MPS1 recruitment, which has been shown to contribute to the turnover of kinetochore–microtubule attachments possibly via NDC80 phosphorylation at other sites or phosphorylation of the Ska complex (Hayward et al., 2022; Maciejowski et al., 2017; Sarangapani et al., 2021).

To explain these effects of Aurora B, we conclude there must be other crucial targets for Aurora B that cannot be accessed by Aurora A. Aurora B is known to phosphorylate and regulate components important for assembly of the microtubule-binding components of the kinetochore with the inner centromere (Petrovic et al., 2016) and hence may indirectly affect the state of the outer kinetochore. An important conclusion of our work is therefore that Aurora A and Aurora B must play specific roles at the outer kinetochore and inner centromere, respectively. Exploring how microtubule binding and the generation of force regulate Aurora A and Aurora B activity differentially to result in defined and non-overlapping biological effects is therefore an important question for future work in this area.

## Materials and methods
### Reagents and antibodies
General laboratory chemicals were obtained from Merck and Thermo Fisher Scientific. Commercially available polyclonal antibodies (pAbs) or monoclonal antibodies (mAbs) were used for pT288 Aurora A (rabbit mAb; 3079S; Cell Signaling Technology), Aurora A (for immunoblotting: rabbit mAb; 4718S; Cell Signaling Technology), Phospho–Aurora A (Thr288)/Aurora B (Thr232)/Aurora C (Thr198; rabbit mAb, 2914S; Cell Signaling Technology), Aurora B (mouse mAb, 611083; BD Transduction labs), pT232 Aurora B (rabbit pAb, Rockland, 600-401-677), PPP6C (rabbit, mAb, ab131335; Abcam), NDC80 pS55 (for immunoblotting: rabbit pAb, GeneTex, GTX70017, batch no 42774), NDC80 (for immunoprecipitation and immunoblotting: mouse mAb, Santa Cruz, sc-515550; for immunoprecipitation only: rabbit pAb, ProteinTech, 18932-1-AP and mouse mAb, Abcam, ab3613; for immunofluorescence: mouse mAb [clone 9G3], ab3613, Abcam), MPS1 (rabbit pAb, 10381-1-AP; ProteinTech), Cyclin B1 (mouse mAb; 05-373; Millipore), pT320 PPP1CA (rabbit mAb, ab62334; Abcam), PPP1CA (rabbit pAb, A300-904A; Bethyl Laboratories), pT481 PRC1 (rabbit mAb, ab62366; Abcam), tubulin (mouse mAb; T6199; Merck), BUB1 (rabbit pAb, Bethyl Laboratories, A300-373A), BUBR1 (mouse mAb, Merck, MAB3612), CENP-A (mouse mAb, ab13939; Abcam; mouse mAb, GTX13939; GeneTex), CENP-C (guinea pig pAb, PD030; MBL life science), CREST (human pAb, 15-234; Antibodies Inc.), Mab414 (nuclear pore proteins [NUPs]; mouse mAb, ab24609; Abcam), NUP153 (rabbit pAb, A301-788A; Bethyl Laboratories), TPX2 (mouse mAb, ab32795; Abcam), MAD1 (rabbit pAb, GTX105079; GeneTex), pericentrin (rabbit pAb, ab4448; Abcam), pT210 PLK1 (mouse mAb, BD-558400; BD Biosciences, rabbit mAb, ab155095;

Abcam, rabbit pAb, 5472S; Cell Signaling Technology), Nuf2 (mouse mAb, sc-271251; Santa Cruz) and GFP (mouse mAbs, 11814460001; Roche). Antibodies against Aurora A (sheep pAb), Astrin (rabbit, pAb), Kinastrin, (sheep, pAb), mCherry (sheep, pAb), GFP (sheep, pAb), and PLK1 (goat, pAb) have been described previously (Bastos and Barr, 2010; Dunsch et al., 2011; Neef et al., 2005; Thein et al., 2007; Zeng et al., 2010).

NDC80 phosphoantibodies were raised to Aurora sites described in the literature (Kettenbach et al., 2011). Custom rabbit pAb anti-pS62 and anti-pT61-pS62 NDC80 were raised by Eurogentec using a 28-d protocol. For immunizations, peptides CLFGKRT-S(PO$_3$H$_2$)-GHGSRN and CSLFGKR-T(PO$_3$H$_2$)-S(PO$_3$H$_2$)-GHGSRN fused to KLH carrier protein were used, respectively. Total IgG was isolated from postimmune sera by affinity purification on Sepharose-protein G (17-0618-05; Merck, Cytiva). Custom sheep pAb anti-pS55 and anti-pS44 NDC80 were raised by the Scottish Blood Transfusion Service using a 4-mo-long program, and peptides CSERKVS(PO$_3$H$_2$)LFGKR and CFGKLS(PO$_3$H$_2$)INKP, respectively, were fused to KLH carrier protein for immunizations. Total IgG was isolated from postimmune sera by affinity purification on Sepharose-protein G. NDC80 pS55 and pS44 specific antibodies were further immunopurified by incubation with the phosphopeptide antigen immobilized on Sulfo-Link Coupling Gel (20401; Pierce). Secondary donkey antibodies against mouse, rabbit, or sheep, labeled with HRP were purchased from Jackson ImmunoResearch Laboratories (715-035-150, 711-035-152, 713-035-147, respectively). Secondary antibodies for immunofluorescence were purchased from Thermo Fisher Scientific and Jackson ImmunoResearch Laboratories.

### Expression plasmid and CRISPR gRNA construction
CRISPR-Cas9 plasmids were as follows: pSpCas9(BB)-2A-Puro (PX459) V2.0 (Addgene 62988; a gift from Feng Zhang; Ran et al., 2013) was digested with BbsI, and the gRNA sequence targeting human PPP6C (5′- GCTAGACCTGGACAAGTATG-3′) was inserted by annealed oligonucleotide cloning. LentiCRISPR v2-Blast (83480; Addgene; a gift from Mohan Babu) and lentiCRISPR v2 (52961; Addgene; a gift from Feng Zhang; Sanjana et al., 2014) were digested with BsmBI, and the gRNA sequences targeting human PPP6C (5′-ACAGTGTGTGGAGATATCCA-3′) and human NDC80 (guide #1, 5′-CTCACGTTTGAGGGGTATAG-3′; guide #2, 5′-GTGTATTCGACAACTCTGTG-3′; guide #3, 5′-CTGGGATCT TAACTCCTGCA-3′), respectively, was inserted by annealed oligonucleotide cloning.

Human NDC80, Aurora A, and TPX2 were amplified from human testis cDNA (Marathon cDNA, Takara Bio Inc.) using Pfu polymerase (Promega). Mammalian expression constructs were based on pcDNA5/FRT/TO vectors, modified to encode the NDC80-GFP, GFP-TPX2, and GFP-Aurora A reading frames, respectively. To generate single/double point alanine mutants (S44A, S55A, S62A, and T61A-S62A), the NDC80 sequence was mutagenized using Pfu polymerase, and the presence of point mutations was confirmed by Sanger sequencing (Source Bioscience service). To generate phospho-null (9A) or phosphomimetic (9D) NDC80 phosphorylation sites, the site encoding amino acids Ser4, Ser5, Ser8, Ser15, Ser44 Thr49, Ser55, Ser62, and Ser69 were changed to either alanine (9A) or aspartic acid

(9D) by mutagenesis. To generate Aurora A–binding TPX2 mutant (YYD-AAA), the site encoding amino acids Tyr8, Tyr10, and Asp11 were changed to alanine by mutagenesis.

## Cell lines and cell culture

HeLa, HEK293T, and HEK293FT cell lines were purchased from ATCC and cultured in DMEM, high glucose, GlutaMAX Supplement, pyruvate (#10569010; Gibco, Thermo Fisher Scientific) supplied with 10% FBS (Merck). eHAP cells were purchased from Horizon Discovery (C669) and cultured in Iscove's Modified Dulbecco's Medium (#12440061; Gibco, Thermo Fisher Scientific) supplied with 10% FBS. All cells were cultured at 37°C and 5% $CO_2$. CRISPR/Cas9-edited HeLa cells with an inserted GFP tag in the C-terminus of MPS1 gene (HeLa MPS1-GFP[CRISPR]) have been described previously (Alfonso-Pérez et al., 2019).

To generate eHAP PPP6C KO cells, parental haploid eHAP cells were transfected with pSpCas9(BB)-2A-Puro (PX459) V2.0 containing the gRNA sequence targeting human PPP6C. Non-transfected cells were killed with transient 2-d puromycin selection and then cell colonies expanded in non-selective medium to be analyzed by immunoblotting and sequencing. To generate HeLa PPP6C KO cells, lentiCRISPR v2-Blast containing the gRNA sequence targeting human PPP6C, pMD2.G, and psPAX2 (12259 and 12260; Addgene; gifts from Didier Trono) were used for lentiviral-based packaging from HEK293T cells, and the supernatant was used to infect HeLa cells. After infection, antibiotic-resistant clones were selected by blasticidin and expanded in a non-selective medium to be analyzed by immunoblotting. HeLa PPP6C KO clone#34 was used for all further experiments. HeLa cell lines with single integrated copies of the desired NDC80-GFP transgene in pcDNA5/FRT/TO were created using the doxycycline-inducible T-REx Flp-In system (Invitrogen). Transgene expression was induced by addition of 1 µg/ml doxycycline (D9891; Merck) to the culture medium for minimum of 24 h.

## Genome-wide CRISPR KO screen for synthetic growth defects with PPP6C

Human genome targeting CRISPR KO (GeCKO) v2 lentiviral pooled libraries (A and B libraries) were purchased from Addgene and amplified according to a published protocol (Sanjana et al., 2014). A genome-wide CRISPR screen was performed as described previously (Joung et al., 2017). Briefly, lentivirus produced by HEK293FT cells was tested to achieve an MOI of 0.3 in eHAP parental and PPP6C KO cells. A total of $2 \times 10^8$ cells were transduced with GeCKO v2 libraries A and B. After 24 h, the cells were treated with 1.5 µg/ml puromycin, which was replaced with a non-selective medium 5 d later. A minimum of $1.2 \times 10^8$ cells were maintained throughout the screen. Subsequently, $1 \times 10^7$ cells were used for genomic DNA extraction. Library preparation was done for Next Generation Sequencing by PCR amplification of sgRNA inserts from 108 µg of parental or PPP6C KO cell genomic DNA using NEBNext Ultra II Q5 Master Mix (M0544L). The PCR products were purified and sequenced on Illumina NextSeq with 80 cycles of read 1 (forward) and eight cycles of index 1 in the Next Generation facility at the Wellcome Trust Centre of Human Genetics, University of Oxford. Quality control, normalization, and identification of positively and negatively selected genes using robust rank aggregation were performed with the Model-based Analysis of Genome-wide CRISPR-Cas9 pipeline (MAGeCK; Wang et al., 2019). Data were combined for the two screens using Fisher's method to calculate a combined P value and test for significance (Table S1). For the volcano plot, mean $\log_2$ fold change (LFC) was plotted against negative $\log_{10}$ of Fisher's combined P value calculated. Genes labeled in blue and orange met the following criteria: (i) the LFC value showed either consistent negative (blue) or positive (orange) enrichment in both screens, respectively; (ii) mean LFC was either below –0.25 or above 0.25; (iii) P value was below 0.05 in both of the individual screens; (iv) genes had to pass the criteria of Fisher's test for the combined P value. The volcano plot was created in R, version 4.1.2 using tidyverse package version 1.3.1 (ggplot2 version 3.3.6, ggrepel 0.9.1).

## Clonogenic survival assays

Cells were seeded in 6-well plates (500-2,000 HeLa or 5,000-12,000 eHAP cells per well) and treated with indicated concentration of MPS1-i or AurA-i, or transduced with lentivirus derived from lentiCRISPR v2, which expressed sgRNAs targeting human NDC80. Puromycin-containing medium (1 µg/ml) was added 3 d after the lentiviral transduction. Medium containing the inhibitors or puromycin was refreshed every 2–3 d. After 7–11 d, colonies were fixed with 4% (wt/vol) PFA in PBS and stained with 5% (wt/vol) crystal violet in 25% (vol/vol) methanol for 10 min. The staining solution was aspirated and the plates were rinsed with water before air-drying at room temperature. After the plates were photographed, the retained dye was solubilized with 100% methanol to measure absorbance at 540 nm.

## Gene silencing (RNAi) and RNAi-rescue assays

RNA duplexes were purchased from Dharmacon and Qiagen (listed in Table S2). Cells were transfected with siRNA using Oligofectamine for the times described in the figure legends. For RNAi-rescue assays in Flp-In cells (Fig. 10), NDC80-GFP transgenes were induced by 1 µg/ml doxycycline, while endogenous NDC80 was depleted using oligonucleotides against the 5′-UTR. For RNAi-rescue assays with transient transfection of TPX2 (Fig. 3, B, C, and F–H) or NDC80 (Fig. 9, F and G), cells were transfected twice with GFP-TPX2 or NDC80-GFP expression plasmids at 72 and 24 h before fixation. Endogenous TPX2 or NDC80 were depleted using siRNA against the 3′-UTR for TPX2 or the 5′-UTR for NDC80. The siRNA transfection was performed 48 h before fixation.

## End-point analysis of NDC80 phosphorylation and dephosphorylation

HeLa cells were arrested in mitosis using either 100 ng/ml nocodazole (487928; Merck) or 10 µM STLC (164739-5G; Merck) for 2.5 h (immunofluorescence) or 18 h (immunoblotting).

For biochemical analysis, mitotic cells were harvested by shake-off, washed twice in PBS and once in Opti-MEM, both of which had been pre-equilibrated to 37°C, 5% $CO_2$. Cells were counted and resuspended in pre-equilibrated Opti-MEM to give $7.5 \times 10^6$ cells/ml. Subsequently, cells were either treated with 0.5 µM Aurora A inhibitor MLN8237 (A4110-APE-10 mM;

ApexBio), 10 µM Aurora B inhibitor ZM 447439 (A4113-APE-10 mM; ApexBio), 10 µM Aurora B inhibitor AZD1152 (A4112-APE-10 mM; ApexBio), 25–100 nM PP1/PP2A inhibitor calyculin A (1,336/100 U; TOCRIS), and 40 µM proteasome inhibitor MG-132 (474790-5MG; Merck) along with 1–10 µM MPS1 inhibitor (AZ3146; TOCRIS) or DMSO (D8418; Merck) for up to 25 min.

For immunofluorescence microscopy, nocodazole- or STLC-arrested cells were treated with 20 µM MG132 and the inhibitors described above for 30 min prior to fixation. For the CENP-E inhibitor experiment, asynchronous cultures were treated with MG132 for 3 h, or 300 nM CENP-E inhibitor (GSK923295; TOCRIS) for 3 h and MG-132 with or without Aurora A inhibitor for 30 min before fixation.

### Mitotic exit time courses for NDC80 phosphorylation state
For analysis of mitotic exit time, HeLa cells were arrested with nocodazole, collected by shake-off released into nocodazole-free Opti-MEM at 37°C, 5% $CO_2$ for 25 min to allow mitotic spindle formation. After 25 min, cell suspensions were treated with 20 µM CDK inhibitor flavopiridol (3094/10; TOCRIS) for up to 40 min. At each time point, cells were harvested for NDC80 immunoprecipitations or Western blot analysis.

### NDC80 immunoprecipitations and Western blot analysis
Pellets of $2.25 \times 10^6$ cells were lysed in 0.6 ml of lysis buffer (50 mM Hepes, pH 8, 100 mM KCl, 50 mM EDTA, 1% IGEPAL, 0.25% Triton X-100, 1 mM DTT, 1 mM PMSF [36978; Thermo Fisher Scientific], 1:250 Protease Inhibitor Cocktail [P8340-5Ml; Merck], 1:100 Phosphatase Inhibitor cocktail 3 [P0044-5Ml; Merck], 100 nM okadaic acid [ALX-350-011-M001; Enzo Life-Sciences], and 50 U micrococcal nuclease [M0247S; New England Biolabs]). Cells lysates were either mixed with 3 × Laemmli buffer for immunoblotting analysis or subject to preclearing, and NDC80 complexes were isolated from 1 mg of cell lysate by a 2-h incubation at 4°C with anti-NDC80 antibody cocktail (18932-1-AP; ProteinTech; ab3613;Abcam; sc-515550; SantaCruz) or, in the case of NDC80-GFP constructs, anti-GFP antibodies and 20 µl of protein G–sepharose (GE17-0618-01; Merck). The sepharose beads were washed twice with lysis buffer, once with PBS, and resuspended in 1 × Laemmli buffer. To control for antibody specificity, PBS-washed NDC80 immunoprecipitates were subject to two further washes in λ-phosphatase buffer, treated with 400 U of λ-phosphatase (New England Biolabs, P0753L) for 45 min at room temperature, and then mixed with 3 × Laemmli buffer. For immunoblotting, proteins were separated by SDS-PAGE and transferred onto nitrocellulose membrane using a Trans-blot Turbo system (1704159; Bio-Rad). All Western blots were revealed using ECL (RPN2106; GE Healthcare and GE28-9068-37; GE Healthcare). Protein concentrations were measured by Bradford assay using Protein Assay Dye Reagent Concentrate (5000006; Bio-Rad).

### NDC80 and dephosphorylation time course assays
Nocodazole-arrested HeLa cells were harvested by shake-off, washed twice in ice-cold PBS, and lysed on ice at $1 \times 10^6$ cells/ml ice-cold lysis buffer either deprived of phosphatase inhibitors or containing 100 nM okadaic acid. At the indicated timepoints,

okadaic acid was supplemented to a final concentration of 1.6 µM and lysates were mixed with 3 × Laemmli buffer for immunoblotting.

### Phos-tag gel analysis of NDC80 phosphorylation state
To reveal differently phosphorylated forms of NDC80, cell lysates were precipitated on magnetic carboxylate-modified beads (GE45152105050250; Merck) in the presence of 77% (vol/vol) acetonitrile (271004; Merck) for 30 min at room temperature, washed twice with 70% (vol/vol) ethanol, and resuspended in 1 × Laemmli buffer supplemented with 100 µM $MnCl_2$ (Hughes et al., 2014; Hughes et al., 2019). We found this approach removes components of the lysis buffer and cell lysates that interfere with the resolution of phosphorylated proteins in the next step. Subsequently, samples were resolved on a gel supplemented with 25 µM Phos-tag (Alpha Laboratories Ltd) and 100 µM $MnCl_2$ (Kinoshita et al., 2006). Prior to protein transfer onto the nitrocellulose membrane, gels were soaked for 30 min in SDS-PAGE running buffer supplemented with 10 mM EDTA.

### Analysis of endogenous NDC80 phosphorylation by mass spectrometry
NDC80 immunoprecipitates were resuspended in 1 × Laemmli buffer and subject to SP3 protein digests (Sielaff et al., 2017). Briefly, protein samples were reduced in 10 mM tris(2-carboxyethyl)phosphine (77720; Thermo Fisher Scientific) for 10 min at room temperature and alkylated for 30 min in dark at room temperature in 50 mM 2-Chloroacetamide (C0267; Sigma-Aldrich) dissolved in 50 mM ammonium bicarbonate (09830; Sigma-Aldrich). Subsequently, proteins were precipitated on magnetic carboxylate-modified beads (GE45152105050250; Merck) in the presence of 77% (vol/vol) acetonitrile (271004; Merck) for 30 min at room temperature. Afterward, samples were subject to five rounds of three step-long washes: two washes with 70% (vol/vol) ethanol followed by one 100% acetonitrile wash. Finally, samples were resuspended in 50 µl of 50 mM ammonium bicarbonate buffer and digested at 37°C for 16 h using 1 µg of trypsin (Trypsin Gold, Promega, V5280) and 1 µg of LysC (125-05061; Wako Pure Chemical Corporation). Peptide digests were acidified with formic acid (FA; 33015-M; Sigma-Aldrich) to a final concentration of 5% (vol/vol) and cleared out by centrifugation at $20,800 \times g$ for 30 min.

Peptide samples were desalted on homemade stageTips (Rappsilber et al., 2007). Briefly, two disks of C18 matrix (66883-U; Merck, EmporeTM SPE disks) were inserted into 200 µl TipOne Tip (StarLab, S1111-0206) and activated using 100% acetonitrile. Subsequently, C18 matrix was equilibrated with 0.1% (vol/vol) trifluoroacetic acid (91707; Sigma-Aldrich), after which peptides were bound to C18 matrix, washed with 0.1% (vol/vol) trifluoroacetic acid, and eluted using 80% acetonitrile with 0.6% acetic acid (695092; Sigma-Aldrich). 10% of eluate was dried and analyzed by mass spectrometry (immunoprecipitated proteome), while the remaining eluate was subject to phospho-peptide enrichment protocol (Cundell et al., 2013). Briefly, C18-purified peptide samples were mixed 1:1 (vol:vol) with loading buffer (5% [vol/vol] trifluoroacetic acid, 1 M glycolic acid, and 80% [vol/vol] acetonitrile). Titanium dioxide columns (TopTip; Glygen)

were washed with 65 μl elution buffer (5% [vol/vol] ammonia solution), then three times with loading buffer, and peptide samples were bound 65 μl at a time. After binding, columns were washed once each with loading buffer, then with 0.2% (vol/vol) trifluoroacetic acid in 80% (vol/vol) acetonitrile, followed by 20% (vol/vol) acetonitrile. Finally, phosphopeptides were eluted with three washes of 20 μl of elution buffer into 20 μl of 20% (vol/vol) FA. Before analysis, phosphopeptides were purified on graphite columns (88302; Thermo Fisher Scientific) according to the manufacturer's protocol.

Peptide separation was performed using a Thermo Fisher Scientific Ultimate RSLC 3000 in which peptides were initially trapped on a C18 PepMac100 precolumn (300 μm internal diameter, length 5 mm, 100 Å bead size; Thermo Fischer Scientific) in solvent A (0.1% [vol/vol] FA in HPLC grade water) using a constant pressure of 500 bar. Peptides were then separated on a 40°C heated Easy-Spray RSLC C18 column (75 μm internal diameter, 50 cm length, Thermo Fischer Scientific) using a 15-min long (phosphopeptide samples) or 30-min long (immunoprecipitated proteome samples) linear gradient (15–38% solvent B (0.1% [vol/vol] FA in acetonitrile) at a flow rate of 200 nl/min. Peptide ions were injected into Q Exactive mass spectrometer equipped with an Easy-Spray source (Thermo Fischer Scientific), and spectra were acquired with resolution 70,000; automatic gain control target of $3x10^6$ ions, maximum injection time 50 ms, m/z range 350–1,500. Either the 5 or 10 most intense peaks were selected for higher-energy collision dissociation fragmentation at 30% of normalized collision energy for 15- or 30-min-long gradients, respectively. Peptide identification and quantification were performed using MaxQuant, version 2.1.4.0. (Cox et al., 2011; Tyanova et al., 2016). The mass spectrometry proteomics data have been deposited to the ProteomeXchange Consortium via the PRIDE (Perez-Riverol et al., 2022) partner repository with the dataset identifier PXD038197.

### Immunofluorescence microscopy of mitotic spindles

Cells plated on glass coverslips were either fixed and permeabilized with ice-cold methanol for 5 min on ice or fixed with 3% (wt/vol) PFA in PBS at room temperature for 15–20 min, followed by permeabilization with 0.2% (vol/vol) Triton X-100 in PBS at room temperature for 5–10 min. Combined PTEMF (20 mM Pipes-KOH, pH 6.8, 0.2% [vol/vol] Triton X-100, 1 mM $MgCl_2$, 10 mM EGTA, and 4% [vol/vol] formaldehyde) fixation and permeabilization for 12 min was used for detection of NDC80 pS44, pS55, pS62, BUB1, BUBR1, MAD1, CENP-A, and CENP-C. Antibody dilutions were performed in PBS. Samples were imaged on a standard upright microscope system (BX61; Olympus) with filter sets for DAPI, GFP/Alexa Fluor 488, Cy3/Alexa Fluor 555, and Cy5/Alexa Fluor 647 (Chroma Technology Corp.), a 2,048 × 2,048-pixel complementary metal oxide semiconductor camera (PrimΣ, Photometrics), and MetaMorph 7.5 imaging software (Molecular Devices). Illumination was provided by an LED light source (pE300, CoolLED Illumination Systems). Image stacks with a spacing of 0.4 μm through the cell volume were maximum intensity projected and cropped in Image J (National Institutes of Health). For high-resolution imaging, cells were imaged under an FV3000 confocal microscope using FV31S-SW software (Olympus; 60× 1.40 NA oil-immersion objective; Galvano scanner type, 10 μs/px scan speed, 1,024 × 1,024 px scan size, 2× zoom, 0.42 μm step size) and the images were processed in Image J.

### Metaphase chromosome spreads

For chromosome spreads, 3 × 10 cm dishes per condition of 50% confluent HeLa cells were treated with 100 ng/ml nocodazole for 6 h. Following treatment, mitotic cells were isolated by shake-off and pelleted at 300 × g for 4 min at 37°C in a 15-ml Falcon tube. Cells were resuspended in 5 ml KCl (75 mM), which had been prewarmed to 37°C. Cells were incubated at 37°C for 30 min, with gentle inversion of the tube every 5 min to help prevent cell clumping. Cells were pelleted at 300 × g for 4 min at 37°C and the KCl solution was decanted. Cells were resuspended in the residual volume of KCl that remained in the tube. Cells were fixed in 1 ml cold fixative (3:1 ratio of MeOH and glacial acetic acid) whilst gently vortexing the cells. An additional 9 ml cold fixative was added to the tube and fixed cells were stored in the fridge at 4°C overnight. The next day, fixed cells were pelleted at 300 × g for 4 min at room temperature and the fixative decanted. Cells were resuspended in 1 ml cold fixative. Sterile glass microscopy slides were washed in distilled/deionized water and placed at 45° angles against a tube rack. A 100 μl aliquot of fixed cells was then dropped onto the slides from a height of around 50 cm. Slides were then washed by gently applying cold fixative over the slides with a Pasteur pipette. The slides were then placed on a wet paper towel above a heat block set to 80°C for 2 min to dry. Slides were allowed to dry at room temperature before application of Mowiol 4–88 mounting medium containing Hoechst DNA dye (1:3,000 dilution from a 1 mg/ml stock). A sterile coverslip was placed on top of the Mowiol drops and slides were placed in the fridge at 4°C overnight to dry. Slides were imaged using a 100× oil objective on an Olympus BX61 fixed cell microscope.

### Flow cytometry profiling of DNA content and cell cycle stage

HeLa cells grown in culture were trypsinized and pelleted at 500 × g for 3 min at 37°C in a 15 ml Falcon tube. The media was decanted and cells were gently resuspended in 1 ml PBS. Cells were pelleted at 500 × g for 3 min at 37°C and the supernatant was decanted. Cells were resuspended in 90% (vol/vol) ice-cold MeOH in deionized water while gently vortexing to help prevent cell clumping. Cells were fixed at –20°C overnight. The next day, fixed cells were adjusted to $5 × 10^5$ cells/ml and pelleted at 500 × g for 3 min at room temperature. Cells were resuspended in 300 μl staining buffer (50 μg/ml propidium iodide [P4864; Sigma-Aldrich] and 50 μg/ml RNase [11119915001; Roche]) in PBS. Stained cells were incubated at room temperature for 1 h, protected from light. Samples were then analyzed using a BD LSRFortessa X20 flow cytometer. Data were visualized using FlowJo 10.8.1 (TreeStar, BD).

### Time-lapse imaging of chromosome alignment and segregation in mitosis

Cells were plated in 35-mm dishes with a 14-mm 1.5-thickness coverglass window on the bottom (MatTek Corp). SiR-DNA

(Silicon–Rodamine Hoechst 33342, Spirochrome) was added at a concentration of 50 nM 2–3 h before imaging. For imaging, the dishes or plates were placed in a 37°C and 5% $CO_2$ environment chamber (Tokai Hit) on the microscope stage. Imaging was performed on an Ultraview Vox spinning-disk confocal system running Volocity 6 (PerkinElmer) using a standard inverted microscope (IX81; Olympus) equipped with an electron multiplying charge-coupled device camera (C9100-13; Hamamatsu Photonics). Image stacks with 15 planes and 0.6-µm apart were maximum intensity projected and cropped in ImageJ (National Institutes of Health).

### Measurement of mitotic spindle size and kinetochore intensities

All image processing and analysis were performed using ImageJ. To measure bipolar or monoastral spindle size, a circle was drawn around the spindle capturing the spindle poles and chromosomes. The diameter of this circle was taken as spindle length. Kinetochore intensities for each fluorescent channel were determined by placing 5–8-px-diameter circular regions of interest (ROIs) at the maxima of individual kinetochores, and the mean pixel intensity of each kinetochore ROI was measured and subtracted from a mean background. Possible 10–20 kinetochores were measured per cell and background measurements were derived by taking an equivalent number of pixels as in the ROI, which were as close as possible to the ROI without overlapping with kinetochores.

### Statistical analysis

Statistical analysis was performed using GraphPad Prism 9.3.1 (GraphPad Software). Statistical significance was analyzed using an unpaired two-tailed *t* test with Welch's correction, a one-way ANOVA, a Brown-Forsythe ANOVA, a Kruskal–Wallis test, or Dunn's multiple comparison test. Graphs display the SEM or mean ± SD as indicated in the figure legends. P values shown on graphs are as follows: $P \geq 0.05$ = not significant (n.s.), $P < 0.05$ = *, $P < 0.01$ = **, $P < 0.001$ = ***, $P < 0.0001$ = ****.

To estimate the effect sizes on kinetochore intensities when removing PPP6C or inhibiting other cellular phosphatases, pooled data were analyzed in R using the open-source package dabestr (Ho et al., 2019). Mean differences are plotted in Fig. 9 B as bootstrap sampling distributions for 5,000 bootstrap samples. Each mean difference is depicted as a dot. Each 95% confidence interval is indicated by the ends of the vertical error bars; the confidence interval is bias-corrected and accelerated.

### Online supplemental material

Fig. S1 shows the characterization of PPP6C KO HeLa cells used in Fig. 1 and throughout the study. Additional information on the design and cell lines used for the functional genomics screens are presented in Fig. S2 and extend Fig. 2. Summary data from the screen are presented in Table S1. Target sequences for RNA interference used throughout the study are provided in Table S2. Fig. S3 shows an extended analysis of NDC80 phosphorylation using antibodies and mass spectrometry in support of Fig. 4. Additional NDC80 phospho-antibody specificity controls for Fig. 4 are shown in Fig. S4. Additional analysis of the role of chromosome position in NDC80 phosphorylation is shown in Fig. S5 and extends Fig. 8.

## Acknowledgments

We thank our colleagues in the Barr and Gruneberg groups for much helpful discussion and support.

This work was supported by a Cancer Research UK program grant award C20079/A15940 (F.A. Barr, T. Sobajima, K.M. Kowalczyk, S. Skylakakis, and L.J. Fulcher), DRCNPG-Nov21\100004 (U. Gruneberg, D. Hayward), and UK Medical Research Council Senior Non-Clinical Research fellowship MR/K006703/1 (U. Gruneberg, D. Hayward).

Author contributions: Conceptualization: F.A. Barr; Methodology: S. Skylakakis, T. Sobajima, K.M. Kowalczyk, E. Roberts, and F.A. Barr; Investigation: T. Sobajima, K.M. Kowalczyk, S. Skylakakis, C. Batley, S. Kurlekar, C. Neary, D. Hayward, and L.J. Fulcher; Funding acquisition: F.A. Barr and U. Gruneberg; Supervision: F.A. Barr and U. Gruneberg; Writing—original draft: F.A. Barr; Writing—review and editing: F.A. Barr, T. Sobajima, K.M. Kowalczyk, L.J. Fulcher, and U. Gruneberg.

Disclosures: The authors declare no competing interests exist.

Submitted: 24 May 2022

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

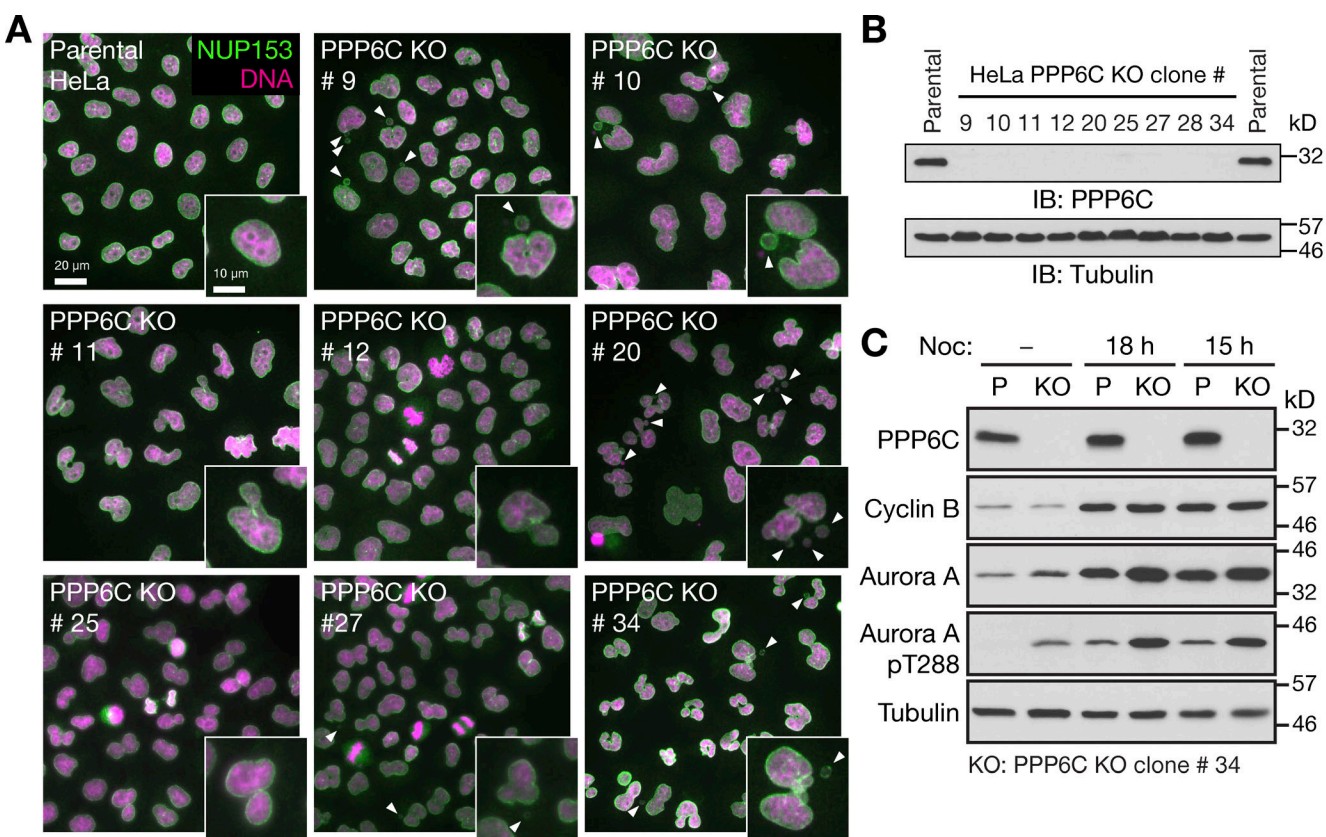

Figure S1.    **Characterization of PPP6C KO HeLa cell lines. (A)** Parental and candidate PPP6C KO clones were stained for DNA and NUP153. Enlarged insets show examples of nuclear morphology in the parental cells and defects in the different KO clones. Arrowheads indicate micronuclei or nuclear morphology defects. **(B)** To confirm PPP6C was deleted, parental HeLa and the candidate PPP6C KO clones were Western blotted for PPP6C and tubulin as a loading control. **(C)** Parental and PPP6C KO clone#34 in asynchronous culture or arrested in mitosis for 15 or 18 h with nocodazole were Western blotted for PPP6C, cyclin B, Aurora A, and the activating pT288 phosphorylation on Aurora A. Tubulin was used as a loading control. Source data are available for this figure: SourceData FS1.

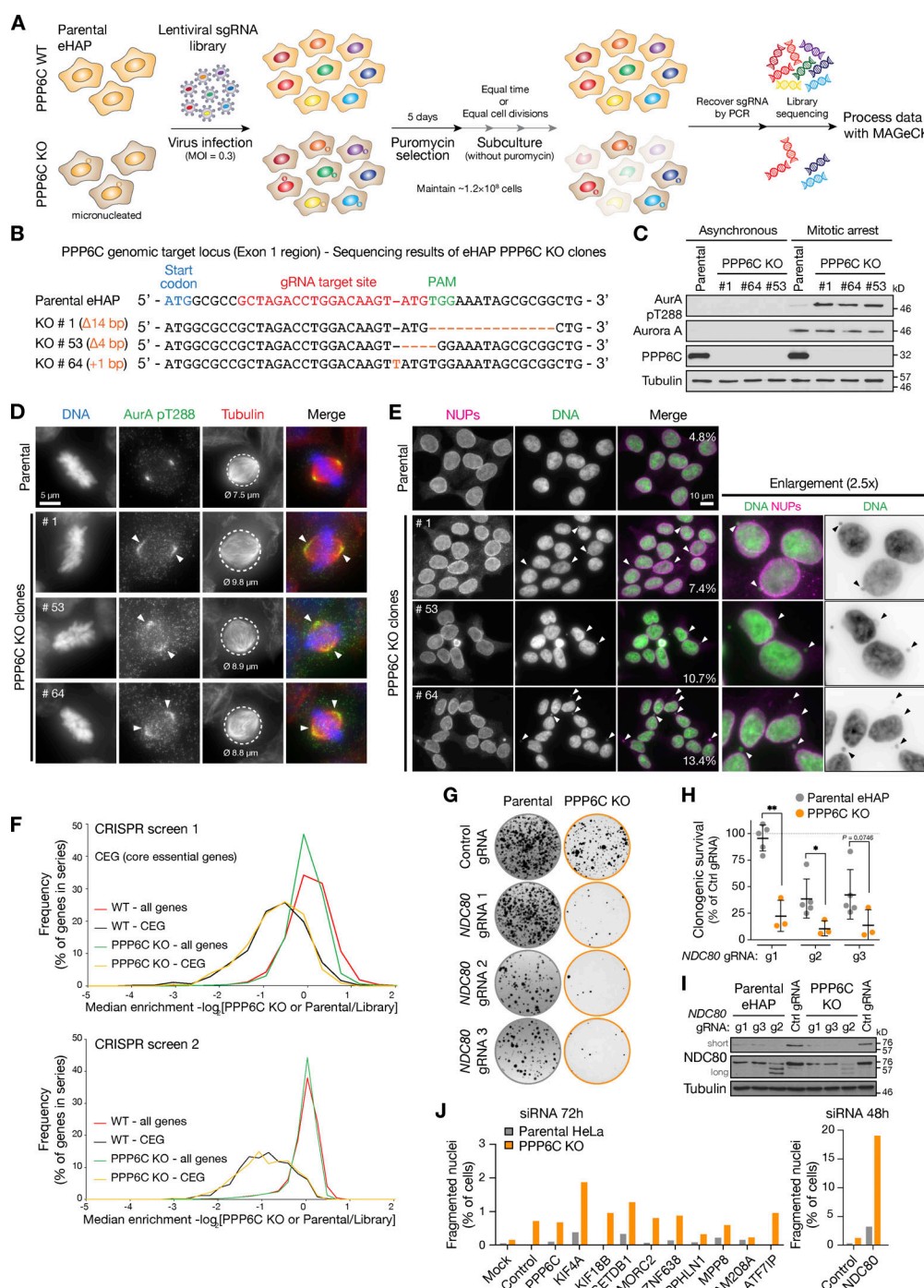

Figure S2. **Comparative functional genomics screening for synthetic growth defects in parental and PPP6C KO eHAP cells. (A)** Workflow for genome-wide CRISPR KO screens using the GeCKO V2 libraries, with data analysis in MaGeCK comparing gene selection in PPP6C KO to parental haploid eHAP cells. **(B)** DNA sequence of the PPP6C genomic locus showing the sequence of candidate PPP6C KO alleles in three candidate haploid eHAP cell clones. **(C)** Western blot of parental eHAP and candidate PPP6C KO alleles showing loss of PPP6C protein and elevation of active Aurora A pT288. **(D)** Parental eHAP and PPP6C KO clone #1 stained for Aurora A pT288, tubulin, and DNA. Representative cells in metaphase are shown, with arrowheads to mark the spread of active Aurora A on the mitotic spindle. Circled areas and numbers indicate the spindle diameter in μm (∅). **(E)** Parental eHAP and PPP6C KO clones cells stained for NUPs and DNA. Groups of interphase cells are shown, with arrowheads to indicate micronuclei. **(F)** Frequency plot of median enrichment of all genes and a selected set of core essential genes (CEG) in screens 1 and 2. **(G)** Clonogenic survival assays for three gRNA sequences targeting NDC80 in parental and PPP6C KO eHAP cell lines. Example images of survival assays are shown. **(H)** Clonogenic survival assays for NDC80 gRNAs g1–g3 relative to the control gRNA in eHAP cells (mean ± SD; n = 3–5). Statistical significance was analyzed using an unpaired two-tailed t test with Welch's correction (*, P < 0.05; **, P < 0.01). **(I)** Western blot validation of NDC80 depletion by the NDC80 g1–g3 gRNAs in eHAP cells. Note g2 results in reduced expression and a ladder of truncated NDC80 protein species. **(J)** Parental and PPP6C KO HeLa cells were treated for 48 or 72 h with siRNA for the indicated negatively selected genes identified by genome-wide screening as candidates for synthetic lethality with PPP6C KO. The proportion of morphologically abnormal nuclei is plotted in the graph. Source data are available for this figure: SourceData FS2.

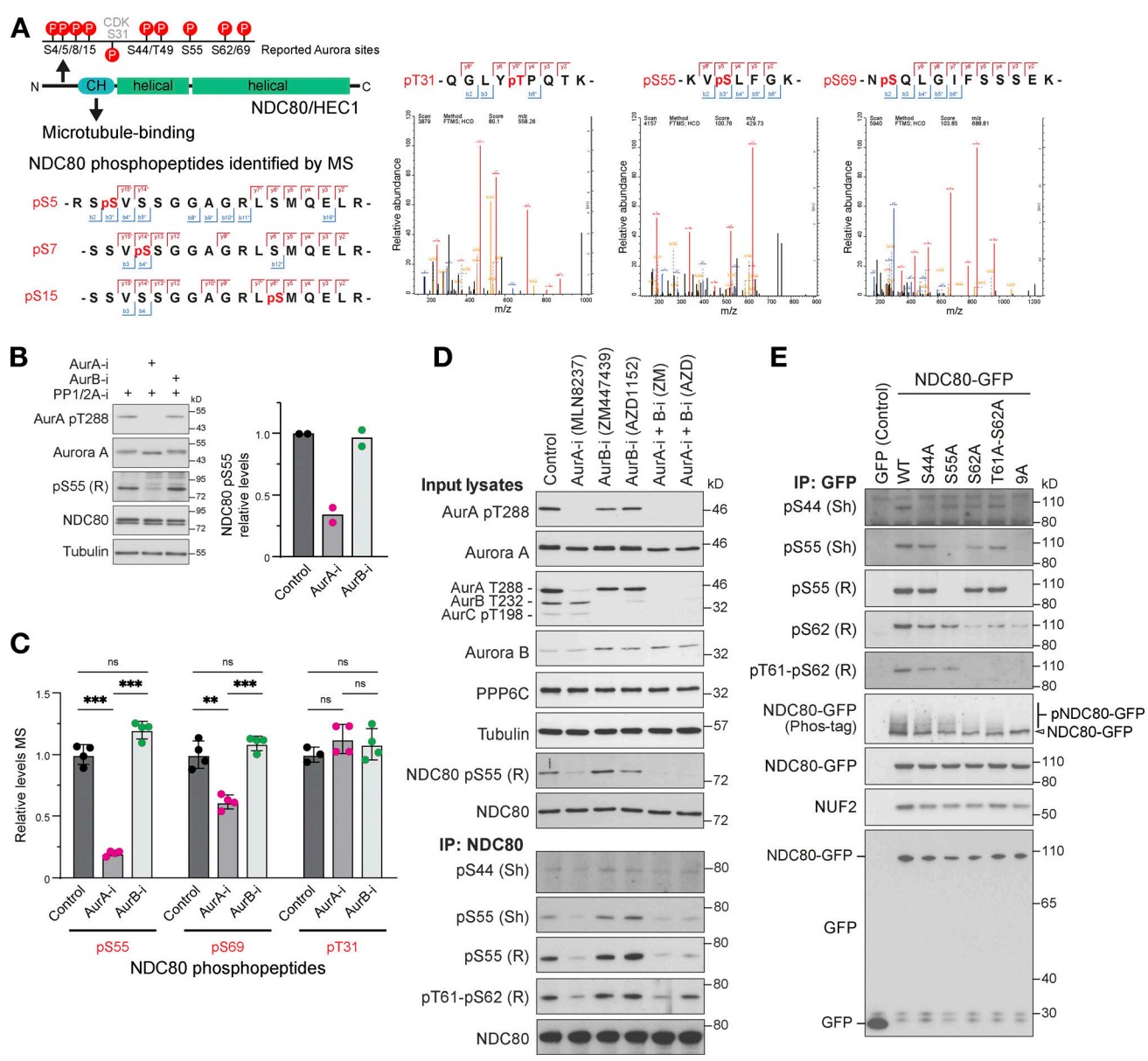

Figure S3.   **Analysis of NDC80 phosphorylation in mitosis using mass spectrometry. (A)** A schematic of NDC80 showing predicted Aurora and CDK1 phosphorylation consensus sites (P) in the N-terminal region adjacent to the calponin homology domain (CH). HeLa cells were arrested in mitosis with nocodazole and then released to allow mitotic spindle formation in the presence of Aurora A and Aurora B kinase inhibitors. Cell lysates were prepared and endogenous NDC80 was isolated by immunoprecipitation (NDC80 IP) and mass spectrometry (MS) as described in the methods. Mass spectrometry/mass spectrometry spectra for NDC80 phospho-peptides are shown for pT31, pS55, and pS69. Due to incomplete cleavage, the peptide containing pS5/7/15 was not quantified. **(B)** Western blot of NDC80 IPs showing effects of Aurora A and B inhibition on NDC80 pS55 (mean; $n$ = 2). **(C)** Mass spectrometry of NDC80 IPs showing effects of Aurora A and B inhibition on NDC80 pS55, pS69, and the CDK-consensus site at pT31 (mean ± SD; $n$ = 3–4). Statistical significance was analyzed using unpaired $t$ test (**, $P < 0.01$; ***, $P < 0.001$). **(D)** HeLa cells arrested in mitosis with nocodazole were released for 25 min to allow mitotic spindle formation in the presence of Aurora A (AurA-i) and Aurora B (AurB-i) kinase inhibitors as indicated. Cell lysates and NDC80 IPs were Western blotted with antibodies to Aurora A and B, activated Aurora A pT288, a pan-phospho-Aurora antibody detecting pT288 and pT232, NDC80, NDC80 pS55, pS44, and pT61-pS62 (rabbit, R; sheep, Sh as indicated in the figure). **(E)** To test for NDC80 phospho-antibody specificity, HeLa cells were transfected for 24 h with plasmids expressing GFP (control), NDC80-GFP (WT), S44A, S55A, S62A, T61-S62A point mutants, or a combined 9A mutant where all consensus Aurora sites described in A. HeLa cells arrested in mitosis for 18 h with nocodazole were released for 25 min to allow mitotic spindle formation in the presence of Aurora A and Aurora B kinase inhibitors as indicated. NDC80 IPs were Western blotted with antibodies to Aurora A and B, activated Aurora A pT288, a pan-phospho-Aurora antibody detecting pT288 and pT232, NDC80, NDC80 pS44, pS55, pS62, or pT61-pS62 (rabbit, R; sheep, Sh as indicated in the figure). Source data are available for this figure: SourceData FS3.

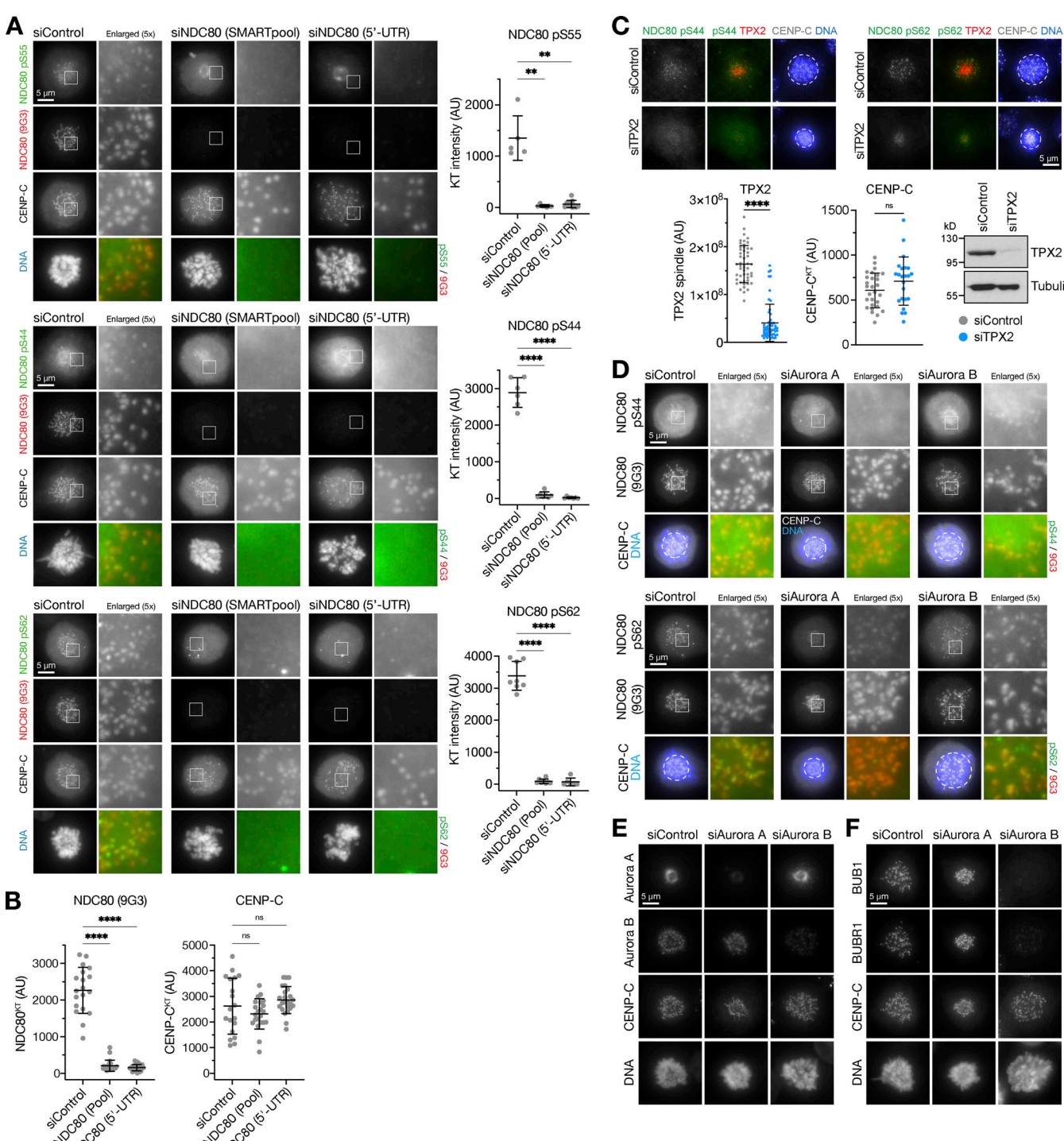

Figure S4. **Specificity of NDC80 antibodies and phospho-antibodies. (A)** HeLa cells were depleted of endogenous NDC80 for 48 h using either an NDC80 siRNA SMARTpool or a single siRNA to the 5′-UTR of NDC80, or treated with a non-targeting control siRNA (siControl). Cells were fixed and then stained for NDC80 (9G3 mouse monoclonal), CENP-C, DNA, or antibodies to specific NDC80 phosphorylation sites pS55, pS44, and pS62. The graphs in each panel show the NDC80 phospho-antibody signal is significantly reduced after NDC80 siRNA (mean ± SD; $n$ = 5–11). Statistical significance was analyzed using a Brown-Forsythe ANOVA (**, P < 0.01; ****, P < 0.0001). **(B)** NDC80 siRNAs resulted in loss of NDC80 but not the inner centromere protein CENP-C (mean ± SD; $n$ = 19–24). Statistical significance was analyzed using a Brown-Forsythe ANOVA (****, P < 0.0001). **(C)** HeLa cells were depleted of endogenous TPX2 for 72 h or treated with a non-targeting control siRNA (siControl). Cells were fixed and then stained for TPX2, CENP-C, DNA, or antibodies to specific NDC80 phosphorylation sites pS44 and pS62. TPX2 siRNA resulted in loss of TPX2 but not the inner centromere protein CENP-C (mean ± SD; $n$ = 23–57), confirmed by Western blot. Statistical significance was analyzed using an unpaired two-tailed $t$ test with Welch's correction (****, P < 0.0001). **(D)** HeLa cells were depleted of Aurora A or Aurora B for 48 h using siRNA, or treated with a non-targeting control siRNA (siControl). Cells were fixed and then stained with antibodies to specific NDC80 phosphorylation sites pS55, pS44, and pS62. **(E and F)** Cells treated as in D were stained for Aurora A, Aurora B, CENP-C, DNA (E), or the spindle checkpoint proteins BUB1 and BUBR1 (F). Source data are available for this figure: SourceData FS4.

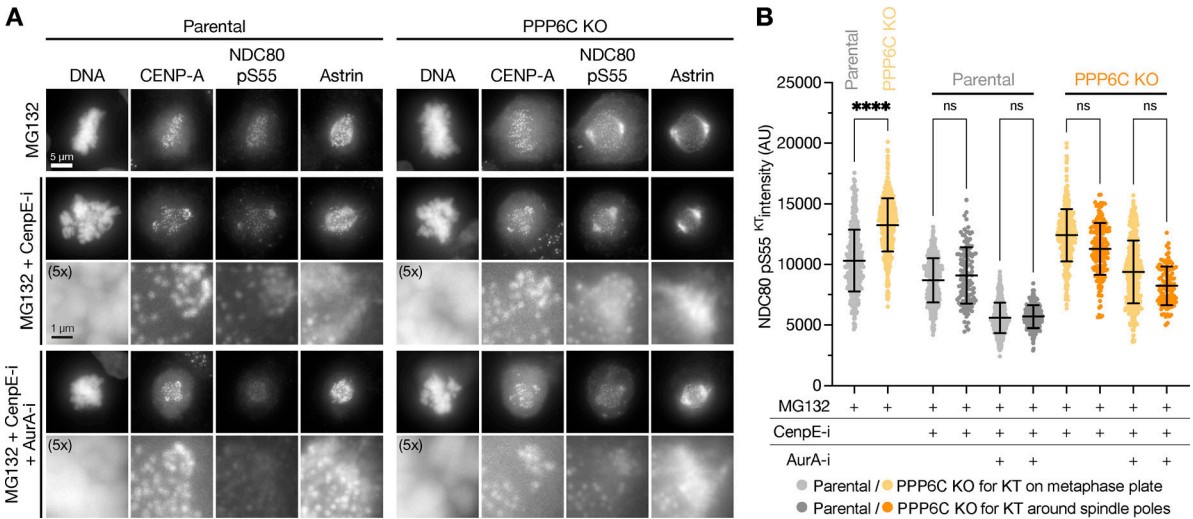

Figure S5.   **NDC80 phosphorylation is elevated in PPP6C KO cells, independent of the spatial position of chromosomes within the mitotic spindle. (A)** Parental and PPP6C KO HeLa cells were arrested in metaphase with MG132 with aligned chromosomes, with MG132 and CENP-E (CenpE-i) inhibitor to trap some chromosomes at the spindle poles, or with MG132, CENP-E, and Aurora A (AurA-i) inhibitors. Cells were stained for NDC80 pS55, astrin, CENP-A, and DNA. Enlarged panels for CENP-E inhibitor conditions show kinetochores (KTs) close to spindle poles. **(B)** NDC80 pS55 intensity (mean ± SD; *n* = 101–709 KTs) at metaphase-aligned chromosomes (MG132) and chromosomes trapped at spindle poles (MG132+CenpE-i). Aurora A inhibitor (AurA-i) was used to confirm NDC80 pS55 signal was dependent on Aurora A activity for aligned and trapped chromosomes. Statistical significance was analyzed using a Dunn's multiple comparison test (****, P < 0.0001).

**Provided online are two tables. Table S1 is a genome-wide CRISPR screening summary data table. Table S2 lists descriptions of the commercial and custom siRNA duplexes used in this study.**

