## [Peer Review File · The Journal of Cell Biology]

PP6 regulation of Aurora A-TPX2 limits NDC80 phosphorylation and mitotic spindle size

Tomoaki Sobajima, Katarzyna Kowalczyk, Stefanos Skylakakis, Daniel Hayward, Luke Fulcher, Colette Neary, Caleb Batley, Samvid Kurlekar, Emile Roberts, Ulrike Gruneberg, and Francis Barr

Corresponding Author(s): Francis Barr, University of Oxford

Review Timeline:

Submission Date:	2022-05-24
Editorial Decision:	2022-07-25
Revision Received:	2022-11-22
Editorial Decision:	2023-01-31
Revision Received:	2023-02-02

Monitoring Editor: Hironori Funabiki

Scientific Editor: Tim Fessenden

Transaction Report:

DOI: <https://doi.org/10.1083/jcb.202205117>

July 25, 2022

Re: JCB manuscript #202205117

Prof. Francis A Barr
University of Oxford
Department of Biochemistry
South Parks Road
Oxford OX1 3QU
United Kingdom

Dear Prof. Barr,

Thank you for submitting your manuscript entitled "PP6-regulated Aurora A activity towards NDC80 limits mitotic spindle size to prevent micronucleation" to Journal of Cell Biology. The manuscript has now been assessed by expert reviewers, whose reports are appended below. Unfortunately, after an assessment of the reviewer feedback, our editorial decision is against publication in JCB.

As you will see, the reviewers find that overall the experiments are nicely executed, but they all agreed that the level of the conceptual advancement is not substantial enough to meet JCB standards. Specifically, it has been established that Ndc80 is phosphorylated and regulated by Aurora A. This manuscript shows some differences from the previous studies, but the reviewers feel that the conclusion is not compelling enough to provide a clearer view on this process.

Unfortunately I do not have the level of reviewer support that I would need to proceed further with the paper. I do realize that significant further work and expansion might convincingly address some of these issues, but I am hesitant to encourage you to work towards the aim of further consideration at JCB. The level of reviewer criticism makes it impossible for me to guarantee that we will be able to invite resubmission, even after revision. Therefore, I regret to return this manuscript to you, hoping that reviewers' comments will be useful to make necessary revisions to submit elsewhere. Our journal office will transfer your reviewer comments to another journal upon request.

I am sorry our decision is not more positive, but hope that you find the reviews constructive. Of course, this decision does not imply any lack of interest in your work and we look forward to future submissions from your lab.

Thank you for your interest in Journal of Cell Biology.

Sincerely,

Hironori Funabiki
Monitoring Editor
Journal of Cell Biology

Tim Fessenden
Scientific Editor
Journal of Cell Biology

Reviewer #1 (Comments to the Authors (Required)):

This paper investigates the role of the PP6-AurA axis in controlling chromosome segregation. PP6 is known to restrain AurA activity and so in the absence of PP6 the kinase becomes hyperactive. AurA has previously been implicated in controlling chromosome segregation by phosphorylating specific sites on Ndc80 N-terminal tail. Here the authors undertook a genome wide CRISPR-Cas9 screen to look for synthetic lethals in cells where PP6 has been removed (PPP6C KO eHAP cells). They find that Ndc80 is synthetic lethal with the absence of PPP6C and investigate this further. They show that AurA targets specific sites in Ndc80 (S55 and S62) once microtubules bind kinetochores and through chemical inhibition studies (using AurA and AurB inhibitors) argue that it must be AurA that phosphorylates Ndc80 rather than previous literature arguing for AurB. They also show that Ndc80 phosphorylation status reversely correlates with recruitment of SAC proteins and so a model where Ndc80 phosphorylation by AurB generates unattached kinetochores to activate the SAC is not supported by these data. They show that AurA-PP6 regulates spindle size through phosphorylation of Ndc80 in that AurA inhibition causes reduced size while PP6 KO cells have larger spindles which is suppressed in Ndc80 9A cells.

Although the paper makes some interesting conclusions and overall the experiments are nicely set up I have a number of concerns that prevents me from supporting publication in its current form.

- 1) The authors make a major point out of the fact that AurA regulates Ndc80 rather than AurB. However, this point is not fully correct and discards previous observations in the field that AurA can phosphorylate specific sites in Ndc80 while AurB can phosphorylate others (see for instance DeLuca JCB 2018). Given that they only look at S55 and S62 I think they have insufficient data to make this point. It is more likely that some sites in Ndc80 are phosphorylated by AurA and others by AurB and they have different functions. A more nuanced view of the literature is needed and testing of existing Ndc80 abs that have been used in the field is needed if they solidly want to make these claims.
- 2) I have looked for careful validation of the Ndc80 phospho abs but cannot find it. The authors need to express Ndc80 point mutants (in Ndc80 depleted cells) and show by IF and WB +/- phosphatase that they are specific - this is a major tool for them and essential to validate carefully. From the methods it is unclear when they use the different S55 abs they list and also it is unclear if their T61/S62 ab is used to probe for double phosphorylated Ndc80.
- 3) It is unclear to the reader why the other components of the Ndc80 complex is not scoring in the screen - one would expect this as they act as a complex. Is PPP6C KO cells sensitive to Nuf2 depletion? If the PPP6C KO cells are dependent on the SAC for survival why is this not reflected in the CRISPR screen data?
- 4) What is the level of Plk1 T210 in PP6C KO cells? Is this also elevated in activity.
- 5) In figure 9 they use Ndc80 9A and 9D but this to me disrupts the arguments in the paper. Why not use Ndc80 S55A/S61A/S62A (and D version) otherwise it is difficult to connect this data to the previous data. They need to show Ndc80 depletion efficiency.
- 6) Ndc80 depleted guides: are they targeting specific regions of Ndc80 or covering the entire gene?

Reviewer #2 (Comments to the Authors (Required)):

In this manuscript the authors aim to uncover pathways and proteins important for survival of cells in which Aurora Kinase A or its regulating phosphatase PP6 are mis-regulated. Using a CRISPR screen they uncover a synthetic lethality between catalytic subunit of PP6 and kinetochore protein NDC80. They present data that argues NDC80 is phosphorylated at Ser55 by PP6 regulated and spindle localized pool of Aurora A. They show that this regulation is not important for checkpoint signaling, and only localizes to properly-attached and checkpoint silenced kinetochores. They go on to show that loss of PP6 activity causes an increase in overall spindle size and that Aurora A phosphorylation of NDC80 is important for regulating this function which helps to promote accurate segregation and preserve nuclear morphology.

This work is technically well executed and the findings have a clear relevance to treatment of human cancer. This work also begins to address out gaps in understanding of Aurora A biology and substrates necessary for successful mitosis and the PP6C KO cell lines are an interesting tool for studying Aurora A upregulation. While most of the data/experiments are well executed, one of their main conclusions isn't necessarily novel. In fact, a recent publication from JCB (Iemura et al 2021) also presents evidence that Aurora A is the major kinase phosphorylating NDC80 at Ser55. They also propose a similar mechanism in which spindle localized, TPX2 activated-AurA is responsible for this phosphorylation. Furthermore, this conclusion does contradict work from the DeLuca lab (2018, 2011), which argues that NDC80 Ser55 is co-regulated by Aurora A and B. While the authors do acknowledge the discrepancies and posit Aurora kinase inhibitor promiscuity could explain the difference in results, they do not attempt to validate their findings with genetically rigorous methods (which is done in Iemura et al 2021).

Major Points

1. As mentioned above, the conclusion that Aurora A is the major kinase phosphorylating NDC80 at Ser55 isn't necessarily novel. That being said, the methods used to come to this conclusion are generally well thought out and supportive of this conclusion. The western blotting to validating the on-target specificity of the Aurora A vs B inhibitors is a key experiment, due to their noted promiscuity, and helps support the validity of their claims. That being said, in the discussion the authors note published work from DeGroot et al. that investigates Aurora kinase inhibitor specificity. The authors use this work to cite potential promiscuity of Aurora B inhibitors as an explanation for the discrepancy from published work-we encourage the authors to re-examine this text as the data support the specificity of Aurora B inhibitors and not promiscuity of Aurora A inhibitors. We suggest that the authors use parallel genetic methods (RNAi or CRISPR) to inactivate Aurora A and B to confirm the conclusions about pSer55 phospho-regulation.
2. The hypothesis that TPX2/PP6 regulated and spindle localized Aurora A phosphorylates Ser55 via at attached kinetochores is an intriguing and logical conclusion based on that data presented. We have a few suggestions to strengthen these claims. Firstly, pSer55 localization to kinetochores in metaphase and anaphase is strongly contradicted by data from DeLuca et al. 2011 which shows this phosphorylation disappearing as cells progress to metaphase. Explanation for this discrepancy is use of different phospho-specific antibodies for pSer55. Thus we suggest confirming the specificity of the phospho-specific antibody using RNAi knockdown and rescue experiments. Further experiments can be done to more definitively claim TPX2 activated AurA on the spindle is regulating NDC80 phosphorylation. For example, using an AurA-TPX2 binding mutant will definitively prove that this pool is doing the phosphorylation. That being said, the experiments using CENP-E inhibitors to show Ser55 phosphorylation is upregulated regardless of chromosome position do successfully support these claims (Figure 5).
3. The observed increased spindle size phenotype observed in the PP6C mutants is one of the most compelling observations in the paper. It strongly suggests titration of Aurora A activity is essential for ensuring proper spindle size. The connection between

spindle size and chromosome segregation abnormalities and nuclear morphology defects is well demonstrated. While the spindle size increase in PP6C KO is well demonstrated, we suggest examining spindle size in PP6C WT cells overexpressing Aurora Kinase A. This will further validate that the spindle size increase is Aurora A regulated and not another target of PP6. These experiments will give stronger validity to claims than spindle size decrease after treatment with Aurora A inhibitors, as Aurora A's role in microtubule nucleation and centrosome separation can't be ruled out in this condition like they were in the PP6C KO cells. We note that throughout the paper the authors have been mostly focused on pSer55 but for rescue experiments they use phospho-mutant/mimetic alleles with mutations at all 9 Aurora kinase consensus sites in NDC80. While these rescue experiments support their hypothesis for the most part, it may be more powerful to demonstrate rescue with a pSer55 single phospho-mutant/mimetic. Furthermore, in the rescue experiments, measurement of spindle size changes from the measurements used in the previous figure to just spindle length. It appears that while the 9A mutant does have a smaller spindle length the spindle width isn't changed and thus the phenotype of an overall decrease in spindle size is consistent.

Minor Points

1. In Figure 1, 10X images used make it hard to see nuclear morphology. I think the point is illustrated successfully with just with 40X images.
2. Figure 4 is only mentioned as a whole in the text. We suggest a more detailed discussion of the experiments and their conclusions in the text. Furthermore, while kinetochore localization can be a useful readout of SAC activity, these conclusions would be much stronger if an actual functional assay was used to measure mitotic arrest in SAC active conditions.
3. Several sentences could be made much clearer with some edits to grammar.

Reviewer #3 (Comments to the Authors (Required)):

The manuscript from Sobajima et al. focuses on PP6-Aurora A control of phosphorylation of the kinetochore-microtubule connector NDC80. There is some nice data presented in this manuscript, starting with screens in PP6KO haploid cells that led them to focus on NDC80. However, the work is not placed into the context of prior studies on control of NDC80 by Aurora A, and the suggestion that Aurora B has little-to-no contribution to regulation of NDC80 during mitosis seems unwarranted. Overall, while providing further evidence for NDC80 being a target of Aurora A regulation, the study lacks a significant conceptual or mechanistic advance that would make it suitable for JCB.

General comments: Prior work from the Maresca and DeLuca labs has established Aurora A regulation of NDC80 (e.g. PMIDs: 26166783 & 29187526). Most notably, the DeLuca lab showed that a specific site in the NDC80 tail (S69) is primarily targeted by Aurora A, and that this site is phosphorylated at microtubule-attached kinetochores, where its phosphorylation is important for chromosome oscillations and accurate chromosome segregation. These studies have led to the general acceptance of NDC80 as an important target of Aurora A (e.g. see PMID: 29700233); in particular, the DeLuca work has shown that Aurora A targets NDC80 at microtubule-attached kinetochores, and that this targeting is functionally significant in controlling kinetochore-microtubule attachment dynamics. As PP6 negatively regulates Aurora A, the hyper-activation of Aurora A in PP6KO cells is expected to elevate phosphorylation of substrates such as NDC80, which is confirmed here. As NDC80 is a essential ("common essential" in DepMap) gene, it is not surprising that its inhibition shows a synthetic growth phenotype with PP6KO, which already compromises growth (many other essential genes also showed synthetic growth phenotypes with PP6KO - the authors use the criterion of "both screens" and a modest significance threshold to focus on NDC80, but a systematic follow up of multiple essential genes is lacking). In addition, the effects on spindle size of TPX2 engineered to not interact with and activate Aurora A are established (work in both *C. elegans* and human cells from the Hyman lab; the *C. elegans* work also showed that the effect of inhibiting TPX2-Aurora A on spindle size in that system is dependent on kinetochores; PMIDs: 18663142 & 16054030). Thus, the fact that elevated Aurora A activity due to PP6 KO causes more phosphorylation of NDC80, longer spindles, and associated phenotypes is anticipated from the literature. The authors need to do a significantly better job of placing their work in the context of analysis in the field.

Specific comments:

1. NDC80 pS55 analysis by immunofluorescence has been previously reported following Aurora A and Aurora B inhibition by the DeLuca group. The authors seem to suggest aspects of this prior work are incorrect, without directly comparing their results to that from the prior studies. For example, they show signals for pS55 through anaphase, while the DeLuca group reported significant decline in pS55 signal by metaphase (except at unattached kinetochores). Have the same antibodies been tested and what is the evidence that p55 phosphorylation is specifically being detected under the conditions used? (phosphatase-treatment and blotting does not establish specific detection in immunofluorescence; one would need to replace endogenous NDC80 with an S55A mutant and analyze the signal at kinetochores). The same general consideration applies to Fig. 5 - e.g. the Maresca group reported higher pS55 signal at polar chromosomes using CENPEi treatment, as is also done here but with a different outcome. On a related note, as pS69 was strongly implicated as being Aurora A-regulated and being present at microtubule-attached kinetochores (where it was implicated in controlling kinetochore-microtubule dynamics), it seems important to analyze here. Given all the prior work on NDC80 tail phosphorylation (by both Aurora A & B kinases), it also seems unwise based on primarily one site and antibody to conclude that Aurora A is the dominant NDC80 tail kinase throughout mitosis. There is general agreement that Aurora A is an important NDC80 kinase and the data here supports that view, but it does not exclude a

contribution from Aurora B.

2. In Fig. 1D, it is surprising that NDC80 gRNAs are showing significant clonogenic survival. Have the survivor colonies been genotyped to assess if NDC80 was indeed knocked out in them? Is the knockout efficiency identical between the two cell lines being compared? Also, as the PP6KO growth is compromised, it may be easier to cause loss of long-term survival. Will a similar pattern be seen with other common essential genes? Finally, it would be helpful to know why growth of the PP6KO is compromised. Are the mitotic delay / errors the primary cause, or do these cells also show defects in other cell cycle phases (e.g. time in G1, S-phase, G2, etc.).

3. The analysis in Fig. 4 is nice but it largely shows that PP6KO does not perturb checkpoint signaling, a result anticipated by viability and extended mitosis in this cell line, and the analysis in Fig. 6 seems not very informative as PP1/PP2A represent the major phosphatases during mitotic exit. These data could be supplemental as they do not contribute to the primary focus of the study.

4. The final aspect of the manuscript is focused on spindle size and nuclear defects (caused by chromosome missegregation). Specifically, the authors show suppression of spindle size and mild suppression of nuclear morphology defects in PP6KO by NDC80 9A and enhancement by NDC80 9D. The extent to which the suppression ameliorates phenotypes of a PP6KO (e.g. the slowed growth) is not assessed. These data broadly support NDC80 being a target of regulation by Aurora A but do not have any implications with respect to NDC80 regulation by Aurora B. They also do not shed any mechanistic insight into how Aurora A phosphorylation on NDC80 is acting during chromosome segregation.

Re: JCB manuscript #202205117 Sobajima et al. PP6 regulation of Aurora A-TPX2 limits NDC80 phosphorylation and mitotic spindle size

Response to reviewer comments

Reviewer #1 (Comments to the Authors (Required)):

This paper investigates the role of the PP6-AurA axis in controlling chromosome segregation. PP6 is known to restrain AurA activity and so in the absence of PP6 the kinase becomes hyperactive. AurA has previously been implicated in controlling chromosome segregation by phosphorylating specific sites on Ndc80 N-terminal tail.

Here the authors undertook a genome wide CRISPR-Cas9 screen to look for synthetic lethalties in cells where PP6 has been removed (PPP6C KO eHAP cells). They find that Ndc80 is synthetic lethal with the absence of PPP6C and investigates this further. They show that AurA targets specific sites in Ndc80 (S55 and S62) once microtubules bind kinetochores and through chemical inhibition studies (using AurA and AurB inhibitors) argue that it must be AurA that phosphorylates Ndc80 rather than previous literature arguing for AurB. They also show that Ndc80 phosphorylation status reversely correlates with recruitment of SAC proteins and so a model where Ndc80 phosphorylation by AurB generates unattached kinetochores to activate the SAC is not supported by these data. They show that AurA-PP6 regulates spindle size through phosphorylation of Ndc80 in that AurA inhibition causes reduced size while PP6 KO cells have larger spindles which is suppressed in Ndc80 9A cells.

Although the paper makes some interesting conclusions and overall the experiments are nicely set up I have a number of concerns that prevents me from supporting publication in its current form.

1) The authors make a major point out of the fact that AurA regulates Ndc80 rather than AurB. However, this point is not fully correct and discards previous observations in the field that AurA can phosphorylate specific sites in Ndc80 while AurB can phosphorylate others (see for instance DeLuca JCB 2018). Given that they only look at S55 and S62 I think they have insufficient data to make this point. It is more likely that some sites in Ndc80 are phosphorylated by AurA and others by AurB and they have different functions. A more nuanced view of the literature is needed and testing of existing Ndc80 abs that have been used in the field is needed if they solidly want to make these claims.

Our data analyse phosphorylation of NDC80 during spindle formation and support the view that during this process Aurora A is the main kinase phosphorylating NDC80 at multiple sites (S44, S55, T61/S62 and S69). A major difference to other work, are our observations that NDC80 phosphorylation occurs at checkpoint-silenced, microtubule-attached kinetochores, rather than checkpoint active kinetochores undergoing error correction. Using a combination of immunofluorescence analysis and biochemistry, we demonstrate that Aurora A is the major kinase generating this phosphorylation and is in the right place to do this (Figures 4 and 7).

There have been a number of published reports suggesting that Aurora A can phosphorylate NDC80 and our data confirm and, importantly, extend these findings to include a major biological function, namely spindle size control. Our data thus make an important contribution to the field. We would like to stress that independent reports using newly generated and well-controlled reagents, with some overlap to previous studies are critical for progress in cell biology and biomedical research more broadly.

2) I have looked for careful validation of the Ndc80 phospho abs but cannot find it. The authors need to express Ndc80 point mutants (in Ndc80 depleted cells) and show by IF and WB +/- phosphatase that they are specific - this is a major tool for them and essential to

validate carefully. From the methods it is unclear when they use the different S55 abs they list and also it is unclear if their T61/S62 ab is used to probe for double phosphorylated Ndc80.

Validation data was included in both the main and supplemental figures of the initial manuscript, and we have extended this data. Western blot specificity controls for the antibodies include phosphatase treatment experiments with the various antibodies (now Figure 4B), and also IP western blot controls from NDC80 mutant cell lines (Figure S3E). For both biochemical and microscopy experiments we show the signals are specific for Aurora A kinase activity, and increase on phosphatase inhibition (Figure 4 and 9A-9E). We have further extended this analysis to include further controls using specific point mutant constructs and depletion for NDC80 as suggested (Figure 9F and S4), as well as new data showing phosphorylation requires Aurora kinases and TPX2 (Figure 3 and 4).

3) It is unclear to the reader why the other components of the Ndc80 complex is not scoring in the screen - one would expect this as they act as a complex. Is PPP6C KO cells sensitive to Nuf2 depletion? If the PPP6C KO cells are dependent on the SAC for survival why is this not reflected in the CRISPR screen data?

The functional genomics screen was designed to identify genes showing differential contributions to growth/fitness in PPP6C KO versus the wild type control. Components that are not regulated by PP6 in some way would not be expected to show a differential effect in that case. This may explain why NUF2 was not identified. However, the penetrance of guides for different genes is not uniform so no screen can be viewed as exhaustive. Although we performed both equal time and equal cell division screens, we have not explored the full range of time points necessary to explore all genes. We have also carried out some additional detailed analysis of checkpoint sensitivity for the different parental and PPP6C KO cell lines and the window of sensitivity is significant but narrow, and that may explain why those components could not be identified in the screen. This data was removed during revision of the manuscript due to space considerations, and because it was not the main message of the work.

4) What is the level of Plk1 T210 in PP6C KO cells? Is this also elevated in activity.

This is an interesting question. The short answer is that PLK1 pT210 levels are not altered in PPP6C depleted cells. We have included these data for the referee, showing two replicate experiments and probing parental (WT) and PPP6C KO (KO) mitotic cell extracts with two different PLK1 pT210 antibodies.

Another project in the group by a former PhD student has examined the kinetics of PLK1 T210 phosphorylation during mitotic entry and explored the role of PP6 in relation to BORA. These unpublished data show that although BORA is crucial for PLK1 T210 phosphorylation, PP6 does not seem to play a role in regulation of Aurora A activity towards PLK1 during its activation at G2-M.

5) In figure 9 they use Ndc80 9A and 9D but this to me disrupts the arguments in the paper. Why not use Ndc80 S55A/S61A/S62A (and D version) otherwise it is difficult to connect this data to the previous data. They need to show Ndc80 depletion efficiency.

Previous work in the field has largely treated the cluster of Aurora-sites in the N-terminus of NDC80 as a functional block and used the mutants we have adopted here (DeLuca et al., 2011) Whether or not the sites have specific functions or act as a block remains unclear. Our data show that all of the NDC80 sites that we investigated were Aurora A targets, so to eliminate any phosphorylation for our functional experiments, the NDC80-9A mutant seemed the most sensible solution. Also, this mutant has been used by many other labs (DeLuca, Kops, Musacchio, Dumont) in their work, and hence appeared to be the standard mutant to use.

Importantly, the fully upshifted NDC80 protein on Phos-tag gels is fully downshifted by Aurora A inhibition and loses phosphorylation at multiple different sites in the N-terminus (Figure 4D; see also the full analysis of this experiment in Fig. S3D for different phosphorylation sites on NDC80). These observations, together with our analysis of NDC80 phosphorylation (Figure S3, S4, and Figure 9) suggested NDC80 phosphorylation is dynamic and sub-stoichiometric. These data thus prompted us to use the NDC80-9A/D mutants rather than specific point mutants in Figure 10.

NDC80 depletion is confirmed in Figure S4A-S4B under conditions where CENP-C is not altered.

6) Ndc80 depleted guides: are they targeting specific regions of Ndc80 or covering the entire gene?

The library guides cover the gene, and we have added a description of the specific sites to the methods text. Guide #1: Exon 2 (first coding exon) and intron; Guide #2 and #5: Exon 4; Guide #3: Exon 2; Guide #4 and #6: Exon 3. In line with other work, we find that CRISPR KO methods can generate hypomorphic states rather than being true knockouts. This is useful since it allows analysis of genes that might otherwise be considered "essential" in terms of cell fitness. This was confirmed for NDC80 by western blot (Figure S21).

Reviewer #2 (Comments to the Authors (Required)):

In this manuscript the authors aim to uncover pathways and proteins important for survival of cells in which Aurora Kinase A or its regulating phosphatase PP6 are mis-regulated. Using a CRISPR screen they uncover a synthetic lethality between catalytic subunit of PP6 and kinetochore protein NDC80. They present data that argues NDC80 is phosphorylated at Ser55 by PP6 regulated and spindle localized pool of Aurora A. They show that this regulation is not important for checkpoint signaling, and only localizes to properly-attached and checkpoint silenced kinetochores. They go on to show that loss of PP6 activity causes an increase in overall spindle size and that Aurora A phosphorylation of NDC80 is important for regulating this function which helps to promote accurate segregation and preserve nuclear morphology.

This work is technically well executed and the findings have a clear relevance to treatment of human cancer. This work also begins to address out gaps in understanding of Aurora A

biology and substrates necessary for successful mitosis and the PP6C KO cell lines are an interesting tool for studying Aurora A upregulation. While most of the data/experiments are well executed, one of their main conclusions isn't necessarily novel. In fact, a recent publication from JCB (Iemura et al 2021) also presents evidence that Aurora A is the major kinase phosphorylating NDC80 at Ser55. They also propose a similar mechanism in which spindle localized, TPX2 activated-AurA is responsible for this phosphorylation. Furthermore, this conclusion does contradict work from the DeLuca lab (2018, 2011), which argues that NDC80 Ser55 is co-regulated by Aurora A and B. While the authors do acknowledge the discrepancies and posit Aurora kinase inhibitor promiscuity could explain the difference in results, they do not attempt to validate their findings with genetically rigorous methods (which is done in Iemura et al 2021).

We have added controls using siAurora A and siAurora B as well as TPX2 point mutants unable to bind Aurora A to further confirm that during our experimental situations Aurora A is the major kinase phosphorylating NDC80 (Figure 3 and 4).

Major Points

1. As mentioned above, the conclusion that Aurora A is the major kinase phosphorylating NDC80 at Ser55 isn't necessarily novel. That being said, the methods used to come to this conclusion are generally well thought out and supportive of this conclusion. The western blotting to validating the on-target specificity of the Aurora A vs B inhibitors is a key experiment, due to their noted promiscuity, and helps support the validity of their claims. That being said, in the discussion the authors note published work from DeGroot et al. that investigates Aurora kinase inhibitor specificity. The authors use this work to cite potential promiscuity of Aurora B inhibitors as an explanation for the discrepancy from published work—we encourage the authors to re-examine this text as the data support the specificity of Aurora B inhibitors and not promiscuity of Aurora A inhibitors. We suggest that the authors use parallel genetic methods (RNAi or CRISPR) to inactivate Aurora A and B to confirm the conclusions about pSer55 phospho-regulation.

We agree that previous work has reported that both Aurora A and B act as NDC80 kinases, even if that is not what we find here. However, despite that work, the precise roles and functions of the two kinases in regulating NDC80 are not completely understood as we discuss in the text of the manuscript. That PP6 regulation of NDC80 phosphorylation by Aurora A is important for spindle size control is a major new finding from our work that starts to fill this gap. Our work shows that NDC80 phosphorylation by Aurora A occurs on at checkpoint-silenced microtubule-attached kinetochores in metaphase and persists into anaphase.

We have provided extra support for this idea with multiple lines of evidence: including elevated NDC80 phosphorylation in PPP6C KO (Figure 4B), dependence on Aurora A (Figure 4D-4E) and TPX2 (Figure 4F-4G), and the independence from Aurora B (Figure 4E, 4I-4K, and S3). This is clearly different from ideas in the literature relating to Aurora B function which predict the opposite behaviour.

2. The hypothesis that TPX2/PP6 regulated and spindle localized Aurora A phosphorylates Ser55 via at attached kinetochores is an intriguing and logical conclusion based on that data presented. We have a few suggestions to strengthen these claims. Firstly, pSer55 localization to kinetochores in metaphase and anaphase is strongly contradicted by data from DeLuca et al. 2011 which shows this phosphorylation disappearing as cells progress to metaphase. Explanation for this discrepancy is use of different phospho-specific antibodies for pSer55. Thus, we suggest confirming the specificity of the phospho-specific antibody using RNA knockdown and rescue experiments. Further experiments can be done to more definitively claim TPX2 activated AurA on the spindle is regulating NDC80 phosphorylation. For example, using an AurA-TPX2 binding mutant will definitively prove that this pool is

doing the phosphorylation. That being said, the experiments using CENP-E inhibitors to show Ser55 phosphorylation is upregulated regardless of chromosome position do successfully support these claims (Figure 5).

We agree that it is important to add data testing what happens in cells expressing Aurora A binding mutants of TPX2. This data has been added in the new Figure 3, addressing the role of TPX2 in Aurora A activation and localisation in PPP6C KO cells. Revised Figures 4F-4G and S4C show TPX2 is required for NDC80 phosphorylation by Aurora A during mitotic spindle formation.

Additional data using siRNA specific for Aurora A and B provides support for the kinase inhibitor experiments. Figure 4I-4J, and S4D-S4F.

3. The observed increased spindle size phenotype observed in the PP6C mutants is one of the most compelling observations in the paper. It strongly suggests titration of Aurora A activity is essential for ensuring proper spindle size. The connection between spindle size and chromosome segregation abnormalities and nuclear morphology defects is well demonstrated. While the spindle size increase in PP6C KOs is well demonstrated, we suggest examining spindle size in PP6C WT cells overexpressing Aurora Kinase A. This will further validate that the spindle size increase is Aurora A regulated and not another target of PP6. These experiments will give stronger validity to claims than spindle size decrease after treatment with Aurora A inhibitors, as Aurora A's role in microtubule nucleation and centrosome separation can't be ruled out in this condition like they were in the PP6C KO cells. We note that throughout the paper the authors have been mostly focused on pSer55 but for rescue experiments they use phospho-mutant/mimetic alleles with mutations at all 9 Aurora kinase consensus sites in NDC80. While these rescue experiments support their hypothesis for the most part, it may be more powerful to demonstrate rescue with a pSer55 single phospho-mutant/mimetic. Furthermore, in the rescue experiments, measurement of spindle size changes from the measurements used in the previous figure to just spindle length. It appears that while the 9A mutant does have a smaller spindle length the spindle width isn't changed and thus the phenotype of an overall decrease in spindle size is consistent.

We agree with the view that the data describing a role for PP6 in spindle size are compelling and propose this data should be moved to the first figure to make it clear it is the major focus of the work.

We have extended the data in the new Figure 3 to test for the effects of Aurora A and TPX2 overexpression as the referee suggested (Figure 3A), and also note that the TPX2 mutant experiments suggested by the reviewer are also important in this context (Figure 3F-3H). We also show NDC80 phosphorylation requires TPX2 (Figure 4F-4G and S4C). This is a major addition to the work and we thank the referee for these comments.

Minor Points

1. In Figure 1, 10X images used make it hard to see nuclear morphology. I think the point is illustrated successfully with just with 40X images.

The 10x images helped show reduced overall cell viability, whereas the 40x images showed the nuclear morphology defects more clearly. We have removed the 10x images and revised the supplemental material when carrying out the revision. The important 40x images for NDC80 depletion in PPP6C KO and parental cells are now in revised Figure 4A.

2. Figure 4 is only mentioned as a whole in the text. We suggest a more detailed discussion of the experiments and their conclusions in the text. Furthermore, while kinetochore localization can be a useful readout of SAC activity, these conclusions would be much

stronger if an actual functional assay was used to measure mitotic arrest in SAC active conditions.

We have added a far more detailed description of this figure is provided in the text of the revised manuscript (now Figure 8). Additional data on spindle assembly checkpoint signalling has been added in Figure 5B-5C and S4E-S4F.

3. Several sentences could be made much clearer with some edits to grammar.

We have carefully revised the text for clarity.

Reviewer #3 (Comments to the Authors (Required)):

The manuscript from Sobajima et al. focuses on PP6-Aurora A control of phosphorylation of the kinetochore-microtubule connector NDC80. There is some nice data presented in this manuscript, starting with screens in PP6KO haploid cells that led them to focus on NDC80. However, the work is not placed into the context of prior studies on control of NDC80 by Aurora A, and the suggestion that Aurora B has little-to-no contribution to regulation of NDC80 during mitosis seems unwarranted. Overall, while providing further evidence for NDC80 being a target of Aurora A regulation, the study lacks a significant conceptual or mechanistic advance that would make it suitable for JCB.

The reviewer writes "*the suggestion that Aurora B has little-to-no contribution to regulation of NDC80 during mitosis seems unwarranted*". Multiple figures in the manuscript using different approaches show Aurora B is not the major kinase explaining NDC80 phosphorylation at different stages of mitosis and mitotic exit. Figure 4 shows elevated NDC80 phosphorylation in PPP6C KO cells, and that Aurora A rather than Aurora B is the major NDC80 kinase in spindle formation. That is supported by data showing NDC80 phosphorylation at kinetochores depends on Aurora A and TPX2 and is independent of Aurora B (Figure 4). Crucially we show that NDC80 phosphorylation is mutually exclusive with checkpoint signalling and extends into anaphase (Figures 6 and 8).

While our work clearly builds onto published findings, it also provides major new insights into the role of Aurora phosphorylation of NDC80 in regulating spindle size control, and the regulation of Aurora A by the PP6 phosphatase in this process. These are very clearly mechanistic advances that change our view of regulation of the kinetochore during mitotic spindle formation. Whether or not that provides a conceptual advance, is something that and other researchers will decide in the course of time.

General comments: Prior work from the Maresca and DeLuca labs has established Aurora A regulation of NDC80 (e.g. PMIDs: 26166783 & 29187526). Most notably, the DeLuca lab showed that a specific site in the NDC80 tail (S69) is primarily targeted by Aurora A, and that this site is phosphorylated at microtubule-attached kinetochores, where its phosphorylation is important for chromosome oscillations and accurate chromosome segregation. These studies have led to the general acceptance of NDC80 as an important target of Aurora A (e.g. see PMID: 29700233); in particular, the DeLuca work has shown that Aurora A targets NDC80 at microtubule-attached kinetochores, and that this targeting is functionally significant in controlling kinetochore-microtubule attachment dynamics. As PP6 negatively regulates Aurora A, the hyper-activation of Aurora A in PP6KO cells is expected to elevate phosphorylation of substrates such as NDC80, which is confirmed here. As NDC80 is a essential ("common essential" in DepMap) gene, it is not surprising that its inhibition shows a synthetic growth phenotype with PP6KO, which already compromises growth (many other essential genes also showed synthetic growth phenotypes with PP6KO - the authors use the criterion of "both screens" and a modest significance threshold to focus on NDC80, but a systematic follow up of multiple essential genes is lacking). In addition, the effects on spindle

size of TPX2 engineered to not interact with and activate Aurora A are established (work in both *C. elegans* and human cells from the Hyman lab; the *C. elegans* work also showed that the effect of inhibiting TPX2-Aurora A on spindle size in that system is dependent on kinetochores; PMIDs: 18663142 & 16054030). Thus, the fact that elevated Aurora A activity due to PP6 KO causes more phosphorylation of NDC80, longer spindles, and associated phenotypes is anticipated from the literature. The authors need to do a significantly better job of placing their work in the context of analysis in the field.

We carefully describe all of the papers mentioned by the reviewer and previous work on PP6 in the introduction and discussion. None of those papers addressed a role for PP6 in spindle size or NDC80 phosphorylation, so the results are not “expected”. What that previous work clearly does not address is the function of PP6 in regulating Aurora A, or show that this happens during processes important for spindle size control. The specific target of Aurora A in size control was also not identified. We demonstrate here that this target is NDC80; an important advance for the cell cycle field.

To address the comments relating to DepMap and the outcome of the screens. The referee does not explain why the criteria we use for reproducibility (“both screens”) and significance ($p < 0.01$ in both screens) are flawed. Many data in the JCB and other journals are viewed as significant at the 95% level ($p < 0.05$). We also show analysis in the supplemental data that negative selection of most essential genes is not altered by PPP6C KO in either of the screens. The differential analysis presented in the main figures identifies a very small number of genes showing significant differential positive and negative selection, not as the reviewer writes “*many other essential genes*”. All this when combined supports the view that NDC80, although a core essential gene, shows significant additional negative selection in the PPP6C KO cells, and is thus an interesting candidate for the cell biological part of our study. We have fully revised the data presentation to simplify the findings of the screens (Fig 2C) and provide more complete data in Table S1.

DepMap provides a score for the fitness contribution for a given gene across a very broad panel of cell lines. Although “*essential*”, NDC80, like other core “*essential*” genes, shows a spread of scores across different cell lines. This pattern is the basis of the correlative analysis that many researchers find useful in identifying novel pathways and co-dependencies between genes. The key point we wish to make is that how “*essential*” a gene is depends on the genetic background. It is this aspect of the CRISPR KO methodology that we exploit in our functional genomics screen. Cells lacking PPP6C become much more sensitive to the level of NDC80 (see Figure 4A for an example).

Specific comments:

1. NDC80 pS55 analysis by immunofluorescence has been previously reported following Aurora A and Aurora B inhibition by the DeLuca group. The authors seem to suggest aspects of this prior work are incorrect, without directly comparing their results to that from the prior studies. For example, they show signals for pS55 through anaphase, while the DeLuca group reported significant decline in pS55 signal by metaphase (except at unattached kinetochores). Have the same antibodies been tested and what is the evidence that p55 phosphorylation is specifically being detected under the conditions used? (phosphatase-treatment and blotting does not establish specific detection in immunofluorescence; one would need to replace endogenous NDC80 with an S55A mutant and analyze the signal at kinetochores). The same general consideration applies to Fig. 5 - e.g. the Maresca group reported higher pS55 signal at polar chromosomes using CENPEi treatment, as is also done here but with a different outcome. On a related note, as pS69 was strongly implicated as being Aurora A-regulated and being present at microtubule-attached kinetochores (where it was implicated in controlling kinetochore-microtubule dynamics), it seems important to analyze here. Given all the prior work on NDC80 tail phosphorylation (by

both Aurora A & B kinases), it also seems unwise based on primarily one site and antibody to conclude that Aurora A is the dominant NDC80 tail kinase throughout mitosis. There is general agreement that Aurora A is an important NDC80 kinase and the data here supports that view, but it does not exclude a contribution from Aurora B.

Our work is focussed on PP6 function during mitosis, and in doing this work we have found that Aurora A rather than Aurora B phosphorylates NDC80 at microtubule-attached checkpoint-silenced kinetochores during spindle formation and anaphase. Importantly, we don't base this conclusion on the analysis of a single site. Western blotting and microscopy analysis with multiple phospho-specific antibodies, as well as Phos-tag gels and mass spectrometry support this conclusion (Figure 4, S3 and S4). We have analysed multiple sites on NDC80 (S44, S55, T61, S62, S69) using these different approaches. Our reagents and experiments have been carefully controlled using biochemical methods (Figure S4), and we added phospho-antibody staining of NDC80 point mutants in microscopy experiments (Figure 9F). We also reference and discuss relevant findings from previous work at multiple points in the manuscript text, including the results section.

Some points of difference to previous work do arise from our data and are described in the results and discussed already. We don't seek to justify publication of our work in terms of controversy or by undermining previous findings. The focus of our work in terms of the title, abstract and data presentation is very much on the role of PP6-regulated Aurora A and spindle size control. It is therefore premature to talk about previous work as "incorrect". Our discussion text has been revised to constructively extend the comparison of our results to previous studies as the reviewer suggested. As we discuss, some findings shown in our work are in close agreement with previous reports that Aurora A can act on NDC80, however there are clearly some areas of difference relating to the relative extent to which Aurora A and B act on NDC80 and the relationship to spindle checkpoint signalling.

2. In Fig. 1D, it is surprising that NDC80 gRNAs are showing significant clonogenic survival. Have the survivor colonies been genotyped to assess if NDC80 was indeed knocked out in them? Is the knockout efficiency identical between the two cell lines being compared? Also, as the PP6KO growth is compromised, it may be easier to cause loss of long-term survival. Will a similar pattern be seen with other common essential genes? Finally, it would be helpful to know why growth of the PP6KO is compromised. Are the mitotic delay / errors the primary cause, or do these cells also show defects in other cell cycle phases (e.g. time in G1, S-phase, G2, etc.).

NDC80 CRISPR reduces protein levels and generates multiple truncated bands in some instances (Figure S2I). Western blot also indicates the NDC80 CRISPRs have the same efficiency in parental and PPP6C KO cell lines (Figure S2I). In all cases where cell lines are then compared, growth assay data is standardised for each cell line and not to the parental cells. The more general point is addressed by the supplemental data analysis of the functional genomics screens. This was an important control where we did not find the same pattern of selection that we observed for NDC80 for other core essential genes (Figure S2F).

3. The analysis in Fig. 4 is nice but it largely shows that PP6KO does not perturb checkpoint signaling, a result anticipated by viability and extended mitosis in this cell line, and the analysis in Fig. 6 seems not very informative as PP1/PP2A represent the major phosphatases during mitotic exit. These data could be supplemental as they do not contribute to the primary focus of the study.

Figure 5 provides support for the view that PPP6C does not drive the spindle checkpoint independently from Aurora B in PPP6C KO cells. That was important for us to establish, since it was an obvious possibility that could explain some aspects of the phenotype. Figure

8 (formerly Figure 6) is important for our analysis of NDC80. It shows Aurora A acts upstream of PP1/PP2A, and NDC80 phosphorylation is dynamic and sub-stoichiometric. That is relevant for the choice of NDC80-9A mutants for experiments in Figure 10.

4. The final aspect of the manuscript is focused on spindle size and nuclear defects (caused by chromosome missegregation). Specifically, the authors show suppression of spindle size and mild suppression of nuclear morphology defects in PP6KO by NDC80 9A and enhancement by NDC80 9D. The extent to which the suppression ameliorates phenotypes of a PP6KO (e.g. the slowed growth) is not assessed.

We have restructured the figures to put the data on the role of PP6 in spindle size control in Figure 1, then follow this up with the functional genomics (Figure 2) and characterisation of NDC80 phosphorylation and function (Figure 4 onwards). As reviewer 2 proposed, we look at TPX2 mutants unable to bind Aurora A in the parental and PPP6C KO backgrounds (Figure 3). To provide extra support for the inhibitor specificity we have provided data on the effects of Aurora A/B knockdown and depletion of the Aurora A activator TPX2 (Figure 4 and S4).

Figure 9 now includes data showing single site mutants at key Aurora sites don't alter spindle size, whereas NDC8-9A does. Figure 10H-10I shows a robust suppression not "*mild suppression*" of the nuclear morphology defects in PPP6C depleted cells. In this work we focussed on functions in mitosis, and although other elements of the response to micronucleation and nuclear morphology are interesting they go beyond the scope of this already extensive study.

These data broadly support NDC80 being a target of regulation by Aurora A but do not have any implications with respect to NDC80 regulation by Aurora B. They also do not shed any mechanistic insight into how Aurora A phosphorylation on NDC80 is acting during chromosome segregation.

Our data are clearly relevant for thinking about Aurora B regulation of NDC80. The biochemical analysis shown in our work demonstrates that Aurora A is the major kinase for NDC80 during spindle formation (Figure 4D-4E). We also show that NDC80 phosphorylation by Aurora A occurs at checkpoint-silenced but not checkpoint active kinetochores, and persists into anaphase (Figures 6 and 8). We accept those observations don't prove Aurora B never acts on NDC80, however the findings do have implications for thinking about that question since current thinking suggests Aurora B would be expected to phosphorylate NDC80 at checkpoint active kinetochores during prophase and prometaphase, and not in anaphase. Further work is clearly needed to resolve some questions, and we raise some of this in the discussion.

January 31, 2023

RE: JCB Manuscript #202205117R-A

Prof. Francis A Barr
University of Oxford
Department of Biochemistry
South Parks Road
Oxford OX1 3QU
United Kingdom

Dear Prof. Barr:

Thank you very much for the revised version of your manuscript, "PP6 regulation of Aurora A-TPX2 limits NDC80 phosphorylation and mitotic spindle size". The manuscript has been evaluated by two of the original reviewers. While the reviewers' opinions remain split, after careful assessment of the manuscript, I determined that your data satisfactorily support the major conclusions of the manuscript. Particularly, your finding that Aurora A, not Aurora B, is the major kinase for Ndc80 phosphorylation is compelling, and will have a strong impact on the mitosis field. Therefore, we would be happy to publish your paper in JCB pending resolution of the following minor points. Please provide a point by point response to each of these.

1. In Figure S1, you described isolation of multiple PPP6C KO HeLa clones, but it is never clearly state is a specific clone was used for the rest of the paper. Please define this in the method sections. Also, in the first paragraph of the Results section (page 3), I assume that HeLa cells are used. Please define this in the Results section.
2. Page 4, Fig. S2F and Table S1. Please disclose the list of core essential genes. Table S1 is difficult to understand without guiding explanations. For example, I don't understand "num 1st screen" and other acronyms such as "fdr", "goodsgrna", "lfc", "chl2", etc.
3. Figure 3B and C. Did you do this experiment with transient transfection of GFP-TPX2? If so, please define the plasmid construct, a transfection method, and selection basis.
4. Figure 9F and 9G. According to the figure legends, I understand that constructs were transiently transfected. If so, define the plasmid construct, a transfection method, and selection basis.

In order to avoid delays in publication, please also attend to the following points necessary to meet our formatting guidelines.

A. MANUSCRIPT ORGANIZATION AND FORMATTING:

Full guidelines are available on our Instructions for Authors page, <http://jcb.rupress.org/submission-guidelines#revised>. Submission of a paper that does not conform to JCB guidelines will delay the acceptance of your manuscript.

1) Text limits: Character count for Articles is < 40,000, not including spaces. Count includes abstract, introduction, results, discussion, and acknowledgments. Count does not include title page, figure legends, materials and methods, references, tables, or supplemental legends.

2) Figures limits: Articles may have up to 10 main figures and 5 supplemental figures/tables.

3) Figure formatting: Scale bars must be present on all microscopy images, including inset magnifications. Molecular weight or nucleic acid size markers must be included on all gel electrophoresis.

** Please include scale bars on inset images in Fig 4I.

4) Statistical analysis: Error bars on graphic representations of numerical data must be clearly described in the figure legend. The number of independent data points (n) represented in a graph must be indicated in the legend. Statistical methods should be explained in full in the materials and methods. For figures presenting pooled data the statistical measure should be defined in the figure legends. Please also be sure to indicate the statistical tests used in each of your experiments (either in the figure legend itself or in a separate methods section) as well as the parameters of the test (for example, if you ran a t-test, please indicate if it was one- or two-sided, etc.). Also, if you used parametric tests, please indicate if the data distribution was tested for normality (and if so, how). If not, you must state something to the effect that "Data distribution was assumed to be normal but this was not formally tested."

** Please indicate n for each assay in the corresponding figure legend.

5) Abstract and title: The abstract should be no longer than 160 words and should communicate the significance of the paper for a general audience. The title should be less than 100 characters including spaces. Make the title concise but accessible to a general readership.

6) Materials and methods: Should be comprehensive and not simply reference a previous publication for details on how an experiment was performed. Please provide full descriptions in the text for readers who may not have access to referenced manuscripts.

** Please add details as noted above in our decision letter.

7) Please be sure to provide the sequences for all of your primers/oligos and RNAi constructs in the materials and methods. You must also indicate in the methods the source, species, and catalog numbers (where appropriate) for all of your antibodies. Please also indicate the acquisition and quantification methods for immunoblotting/western blots.

8) Microscope image acquisition: The following information must be provided about the acquisition and processing of images:

a. Make and model of microscope

b. Type, magnification, and numerical aperture of the objective lenses

c. Temperature

d. Imaging medium

e. Fluorochromes

f. Camera make and model

g. Acquisition software

h. Any software used for image processing subsequent to data acquisition. Please include details and types of operations involved (e.g., type of deconvolution, 3D reconstitutions, surface or volume rendering, gamma adjustments, etc.).

10) Supplemental materials: There are strict limits on the allowable amount of supplemental data. Articles may have up to 5 supplemental figures. Please also note that tables, like figures, should be provided as individual, editable files. A summary of all supplemental material should appear at the end of the Materials and methods section.

13) ORCID IDs: ORCID IDs are unique identifiers allowing researchers to create a record of their various scholarly contributions in a single place. At resubmission of your final files, please consider providing an ORCID ID for as many contributing authors as possible.

Please note that JCB now requires authors to submit Source Data used to generate figures containing gels and Western blots with all revised manuscripts. This Source Data consists of fully uncropped and unprocessed images for each gel/blot displayed in the main and supplemental figures. Since your paper includes cropped gel and/or blot images, please be sure to provide one Source Data file for each figure that contains gels and/or blots along with your revised manuscript files. File names for Source Data figures should be alphanumeric without any spaces or special characters (i.e., SourceDataF#, where F# refers to the associated main figure number or SourceDataFS# for those associated with Supplementary figures). The lanes of the gels/blots should be labeled as they are in the associated figure, the place where cropping was applied should be marked (with a box), and molecular weight/size standards should be labeled wherever possible.

WHEN APPROPRIATE: The source code for all custom computational methods published in JCB must be made freely available

as supplemental material hosted at www.jcb.org. Please contact the JCB Editorial Office to find out how to submit your custom macros, code for custom algorithms, etc. Generally, these are provided as raw code in a .txt file or as other file types in a .zip file. Please also include a one-sentence summary of each file in the Online Supplemental Material paragraph of your manuscript.

B. FINAL FILES:

Thank you for this interesting contribution, we look forward to publishing your paper in Journal of Cell Biology.

Sincerely,

Hironori Funabiki
Monitoring Editor
Journal of Cell Biology

Tim Fessenden
Scientific Editor
Journal of Cell Biology

Reviewer #1 (Comments to the Authors (Required)):

The authors have addressed my concerns by providing additional experimental data and validation of Ndc80 antibodies. The paper makes a strong point that AurA-TPX2 and not AurB is the major kinase targeting Ndc80 which has important implications for the field. I therefore support publication.

Reviewer #3 (Comments to the Authors (Required)):

The authors have added additional information and experimental data to address the critical feedback on their initial submission. However, the authors have also argued against many of the substantial comments from reviewers regarding the overall advance presented by the work. The PP6/AurA axis is already established to act reciprocally during spindle assembly, chromosome segregation, and for ensuring normal nuclear morphology. Furthermore, AurA has been known to phosphorylate Ndc80 and it is established that AurA/TPX2 are key factors controlling spindle assembly/size and chromosome segregation. The importance of phosphorylation of Ndc80 at specific sites for ensuring proper spindle length has also been reported. Thus, it is unsurprising that the previously reported hyperactivation of AurA by PP6KO leads to hyperphosphorylation of its substrate(s), including Ndc80, and consequent changes in spindle size, chromosome segregation, and nuclear morphology. Potentially the most significant contribution of this effort is to challenge the prior DeLuca conclusions on AurA vs B contributions to Ndc80 tail phosphorylation in human cells. The findings reported here are radically different and will be important in the future to understand their functional significance. While the work is well-executed and does convince that AurA is an important regulator of Ndc80, it fails to advance understanding beyond that.

Response to reviewers/editor

Reviewer #1 (Comments to the Authors (Required)):

The authors have addressed my concerns by providing additional experimental data and validation of Ndc80 antibodies. The paper makes a strong point that AurA-TPX2 and not AurB is the major kinase targeting Ndc80 which has important implications for the field. I therefore support publication.

We thank the referee for their helpful and constructive comments during the review process. The suggestion to examine TPX2 in PPP6C KO cells, and Aurora A binding mutants of TPX2 was extremely valuable and the addition of that data has greatly improved the manuscript.

Reviewer #3 (Comments to the Authors (Required)):

The authors have added additional information and experimental data to address the critical feedback on their initial submission. However, the authors have also argued against many of the substantial comments from reviewers regarding the overall advance presented by the work. The PP6/AurA axis is already established to act reciprocally during spindle assembly, chromosome segregation, and for ensuring normal nuclear morphology. Furthermore, AurA has been known to phosphorylate Ndc80 and it is established that AurA/TPX2 are key factors controlling spindle assembly/size and chromosome segregation. The importance of phosphorylation of Ndc80 at specific sites for ensuring proper spindle length has also been reported. Thus, it is unsurprising that the previously reported hyperactivation of AurA by PP6KO leads to hyperphosphorylation of its substrate(s), including Ndc80, and consequent changes in spindle size, chromosome segregation, and nuclear morphology. Potentially the most significant contribution of this effort is to challenge the prior DeLuca conclusions on AurA vs B contributions to Ndc80 tail phosphorylation in human cells. The findings reported here are radically different and will be important in the future to understand their functional significance. While the work is well-executed and does convince that AurA is an important regulator of Ndc80, it fails to advance understanding beyond that.

We were pleased that the reviewer acknowledged that our work is well-executed and provides convincing evidence for the conclusions. We accept, and have acknowledged in the manuscript, that some of our interpretations, especially in relation to the importance of Aurora B, are different to those in other studies. However, we have concerns that the reviewer narrows this down to specific named authors in their comments. There is a general view that NDC80 phosphorylation by Aurora B is the crucial regulatory event at the kinetochore, whereas we and others, including the DeLuca lab mentioned by the reviewer, have evidence for the importance of Aurora A. These points and their potential implications are carefully discussed in the manuscript.

1. In Figure S1, you described isolation of multiple PPP6C KO HeLa clones, but it is never clearly state is a specific clone was used for the rest of the paper. Please define this in the method sections.

Also, in the first paragraph of the Results section (page 3), I assume that HeLa cells are used. Please define this in the Results section.

Text revised to address these points:

Methods. Page 16, line 37. "HeLa PPP6C KO clone#34 was used for all further experiments."

Results Page 3, line 27. "To address the mechanistic consequences of Aurora A amplification, we constructed PPP6C knockout (KO) HeLa cell lines and confirmed they showed abnormal nuclear structure and an increase in the activating pT288 phosphorylation on Aurora A (Fig. S1A-S1C)."

Also added clarification for cells used on Pages 4, 5 and 6.

2. Page 4, Fig. S2F and Table S1. Please disclose the list of core essential genes. Table S1 is difficult to understand without guiding explanations. For example, I don't understand "num 1st screen" and other acronyms such as "fdr", "goodsgRNA", "lfc", "chl2", etc.

The Excel file for Table S1 has been updated to include an additional tab listing "core essential genes" and definitions for the column headings.

3. Figure 3B and C. Did you do this experiment with transient transfection of GFP-TPX2? If so, please define the plasmid construct, a transfection method, and selection basis.
4. Figure 9F and 9G. According to the figure legends, I understand that constructs were transiently transfected. If so, define the plasmid construct, a transfection method, and selection basis.

We failed to fully update this method to account for the revised figures, and have now corrected this as requested. Page 17, lines 31-40.

“Gene silencing (RNAi) and RNAi-rescue assays

RNA duplexes were purchased from Dharmacon and Qiagen (listed in Table S2). Cells were transfected with siRNA using Oligofectamine for the times described in the figure legends. For RNAi-rescue assays in Flp-In cells (Fig. 10), NDC80-GFP transgenes were induced by 1 µg/ml doxycycline, while endogenous NDC80 was depleted using oligonucleotides against the 5' -UTR. For RNAi-rescue assays with transient transfection of TPX2 (Fig. 3B, 3C and 3F-3H) or NDC80 (Fig. 9F and 9G), cells were transfected twice with GFP-TPX2 or NDC80-GFP expression plasmids at 72 and 24 hours before fixation. Endogenous TPX2 or NDC80 were depleted using siRNA against the 3' -UTR for TPX2 or the 5' -UTR for NDC80. The siRNA transfection was performed 48 hours before fixation.”

In order to avoid delays in publication, please also attend to the following points necessary to meet our formatting guidelines.

A. MANUSCRIPT ORGANIZATION AND FORMATTING:

Full guidelines are available on our Instructions for Authors page, <http://jcb.rupress.org/submission-guidelines#revised>. Submission of a paper that does not conform to JCB guidelines will delay the acceptance of your manuscript.

- 1) Text limits: Character count for Articles is < 40,000, not including spaces. Count includes abstract, introduction, results, discussion, and acknowledgments. Count does not include title page, figure legends, materials and methods, references, tables, or supplemental legends.
- 2) Figures limits: Articles may have up to 10 main figures and 5 supplemental figures/tables.
- 3) Figure formatting: Scale bars must be present on all microscopy images, including inset magnifications. Molecular weight or nucleic acid size markers must be included on all gel electrophoresis.
**** Please include scale bars on inset images in Fig 4I.**
The missing 1 µm scale bar for the enlarged panels has been added to the figure.
- 4) Statistical analysis: Error bars on graphic representations of numerical data must be clearly described in the figure legend. The number of independent data points (n) represented in a graph must be indicated in the legend. Statistical methods should be explained in full in the materials and methods. For figures presenting pooled data the statistical measure should be defined in the figure legends. Please also be sure to indicate the statistical tests used in each of your experiments (either in the figure legend itself or in a separate methods section) as well as the parameters of the test (for example, if you ran a t-test, please indicate if it was one- or two-sided, etc.). Also, if you used parametric tests, please indicate if the data distribution was tested for normality (and if so, how). If not, you must state something to the effect that "Data distribution was assumed to be normal but this was not formally tested."
**** Please indicate n for each assay in the corresponding figure legend.**
We have fully revised figure legends to include the requested details. Highlighted text marks the key changes.

Fig. 1. PP6 and Aurora A regulate the size of the mitotic spindle.

(A) Metaphase spindle size (mean ± SD; n= 12-13) in parental and PPP6C KO HeLa cell lines stained for active Aurora A pT288, tubulin and DNA. **Statistical significance was analysed using an unpaired two-tailed t-test with Welch's correction (***, p < 0.001).** (B) Metaphase spindle size (mean ± SD; n = 15-29) in parental

and PPP6C KO HeLa cell lines after 30 min treatment in the presence (+) or absence (-) of Aurora A inhibitor (AurA-i). **Statistical significance was analysed using a Brown-Forsythe ANOVA (**, $p < 0.01$; ***, $p < 0.001$; ****, $p < 0.0001$).** (C) Parental and PPP6C KO cell lines were treated with STLC for 3 h to arrest cells in mitosis with monopolar spindles, and then treated for 30 min in the absence (Control) and presence of Aurora A (AurA-i) or Aurora B (AurB-i) inhibitors. The cells were then stained for DNA and CENP-C. Monoastral spindle diameter (mean \pm SD; $n = 9-21$) is shown for the different conditions. **Statistical significance was analysed using a Brown-Forsythe ANOVA (****, $p < 0.0001$).** (D) Time-lapse imaging of DNA segregation in parental and PPP6C KO cells. Nuclear envelope breakdown (NEBD) was taken as the start of mitosis. Anaphase is shown with higher time resolution with arrows to mark anaphase spindle defects in PPP6C KO cells. Arrows indicate chromosomes escaping the anaphase spindle. (E) Mitotic progression from NEBD to anaphase onset (the line marks the median value) and (F) cumulative mitotic index in parental and PPP6C KO cells ($n = 26-28$). PPP6C KO cells show extended mitosis and delayed mitotic exit. (G) Mitotic chromosome spreads from parental and PPP6C KO HeLa cell lines. Arrows indicate broken or unpaired chromosomes. (H) Flow cytometry was used to measure cell cycle distribution (mean \pm SD; $n = 3$) and ploidy of parental and PPP6C KO HeLa cells. Plots show counts of DNA content with dotted lines to mark 2c and 4c in the parental control cell line.

Fig. 3. Aurora A-TPX2 drives enlarged spindle size in PPP6C KO cells.

(A) Metaphase spindle size (mean \pm SD; $n = 15-52$) was measured in HeLa cells overexpressing GFP-Aurora A, GFP-TPX2 compared to the untransfected control (Control). Pericentrin staining marks centrosomes at the spindle poles. **Statistical significance was analysed using a Brown-Forsythe ANOVA (****, $p < 0.0001$).** (B) Metaphase spindle size (mean \pm SD; $n = 13-31$) in HeLa cells depleted of endogenous TPX2 using a 3'-UTR siRNA and then transfected with GFP-TPX2 (WT) or a mutant unable to bind Aurora A (YYD/AAA) or left untransfected (control). **Statistical significance was analysed using a Brown-Forsythe ANOVA (****, $p < 0.0001$).** (C) Western blot of cells in B showing depletion of endogenous TPX2 and expression of GFP-TPX2 constructs. (D) Metaphase spindle size (mean \pm SD; $n = 6-9$) was measured in parental and PPP6C KO HeLa cells treated with control or TPX2 siRNA and stained for activated Aurora A pT288, tubulin and DNA. **Statistical significance was analysed using a Brown-Forsythe ANOVA (**, $p < 0.01$; ****, $p < 0.0001$).** (E) Parental and PPP6C KO cell lines treated with control or TPX2 siRNA were arrested in mitosis with monopolar spindles by STLC for 3 h and treated in the absence (Control) and presence of Aurora A (AurA-i) or Aurora B (AurB-i) inhibitors for 30 min, then stained for DNA and CENP-C. Monoastral spindle diameter (mean \pm SD; $n = 8-13$) is shown for the different conditions. **Statistical significance was analysed using a Brown-Forsythe ANOVA (**, $p < 0.01$; ***, $p < 0.001$; ****, $p < 0.0001$).** (F-H) Parental and PPP6C KO HeLa cells were treated with control or TPX2 3'-UTR siRNA and either mock transfected (Control (-)) or transfected with either GFP-TPX2 (WT) or the YYD/AAA mutant, and then stained for activated Aurora A pT288, tubulin and DNA. The intensity of Aurora A pT288 signal (G) and metaphase spindle size (H) are shown for the different conditions in parental and PPP6C KO cells (mean \pm SD; $n = 7-15$). **Statistical significance was analysed using a Brown-Forsythe ANOVA (**, $p < 0.01$; ***, $p < 0.001$; ****, $p < 0.0001$).**

Fig. 4. PPP6C KO cells show elevated Aurora A dependent phosphorylation of the kinetochore protein NDC80.

(A) Parental and PPP6C KO cells depleted of NDC80 were stained for DNA, tubulin and the nuclear pore marker NUP153 and imaged at 40x magnification to visualize nuclear morphology. Arrows mark micronuclei in NDC80 depleted parental cells. (B) Mitotic lysates of parental and PPP6C KO HeLa cells were blotted for the proteins listed in the figure. Overall NDC80 phosphorylation was monitored using a Phos-tag gel. Relative level of NDC80 S55 phosphorylation, active Aurora A (pT288) and active Aurora B (pT232) were measured (mean \pm SEM; $n = 4-8$). **Statistical significance was analysed using an unpaired two-tailed t-test with Welch's correction (***, $p < 0.001$; ****, $p < 0.0001$).** To control for antibody phospho-selectivity HeLa cell lysates were mock (-) or lambda-phosphatase treated (+ λ -PPase) and blotted for NDC80, NDC80 pS55 and pS62. (C) Dephosphorylation kinetics of Aurora A pT288 and NDC80 pS55 were followed in extracts of parental and PPP6C KO cells. Graphs show dephosphorylation kinetics (mean \pm SD; $n = 3$). (D) Mitotic lysates of parental and PPP6C KO HeLa cells treated for 10 min with Aurora A (AurA-i) and two different Aurora B (AurB-i) kinase inhibitors in the combinations shown were blotted for the proteins listed in the figure. To increase

sensitivity, NDC80 was isolated by IP for NDC80 pS55 blots. An extended analysis of this experiment, including blots for additional NDC80 phosphorylation sites, is shown in **Figure S3D**. **(E)** Parental and PPP6C KO HeLa cells in different kinase inhibited and control conditions from D stained for NDC80 pS55, NDC80, and DNA. **(F)** HeLa cells were depleted of endogenous TPX2 for 72 h or treated with a non-targeting control siRNA (siControl). Cells were fixed and then stained for TPX2, CENP-C, DNA or antibodies to specific NDC80 pS55. **(G)** NDC80 pS55, pS44 and pS62 signal at kinetochores and **(H)** monoastal spindle size in siControl and siTPX2 (mean±SD; n = 5-29). **Statistical significance was analysed using an unpaired two-tailed t-test with Welch's correction (**, p < 0.01; ****, p < 0.0001)**. **(I)** HeLa cells were depleted of Aurora A or Aurora B for 72 h using siRNA, or treated with a non-targeting control siRNA (siControl). Cells were fixed and then stained with antibodies to specific NDC80 phosphorylation sites. Images for NDC80 pS55 are shown in the figure, pS44 and pS62 are shown in **Fig. S4D**. **(J)** NDC80 pS55, pS44 and pS62 signal at kinetochores and **(K)** monoastal spindle size in siControl, siAurora A and siAurora B treated cells (mean±SD; n = 5-24). **Statistical significance was analysed using a Brown-Forsythe ANOVA (**, p < 0.01; ****, p < 0.0001)**.

Fig. 5. Spindle checkpoint signaling remains Aurora B dependent in PPP6C KO cells.

(A) HeLa cells were treated with Aurora B inhibitor and lysed at the times indicated from 0 to 25 min. Samples were blotted with the pan-Aurora T-loop antibody that detects active Aurora A/B/C. **(B)** Parental and PPP6C KO HeLa cells treated with Aurora A (AurA-i) or Aurora B inhibitors (AurB-i) for 30 min were stained for the spindle checkpoint proteins BUB1 and BUBR1. **(C)** Graphs show the number of BUB1- or BUBR1-positive checkpoint active kinetochores is significantly reduced after Aurora A but not Aurora B inhibition (mean ± SD; n = 19-40). **Statistical significance was analysed using a Dunn's multiple comparison test (****, p < 0.0001)**.

Fig. 8. NDC80 phosphorylation and spindle checkpoint signaling are mutually exclusive.

(A) HeLa MPS1-GFP^{CRISPR} cells arrested in mitosis with either nocodazole or the proteasome inhibitor MG132 were left untreated or treated with Aurora B inhibitor (AurB-i). Cells were stained for active Aurora B pT232, phosphorylated NDC80 pS55, and the kinetochore marker CENP-C. The intensities of NDC80 pS55 signal at kinetochores and Aurora B pT232 signal are shown for the different conditions (mean ± SEM; n = 10-12; **, p < 0.01; ***, p < 0.001). **(B)** Parental and PPP6C KO HeLa cells arrested with STLC to create monoastal spindles with a mixture checkpoint active and silent kinetochores states were stained for MAD1, NDC80 pS55, CENP-C and DNA. **(C)** The intensity of NDC80 pS55 signal was measured at MAD1 positive (+ve) and negative (-ve) kinetochores (mean ± SD; n = 9-10), and **(D)** the proportion of MAD1 or NDC80 pS55 positive kinetochores (mean ± SD; n = 17-34) was determined for parental and PPP6C KO HeLa cells. **Statistical significance was analysed using a Brown-Forsythe ANOVA (****, p < 0.0001)**.

Fig. 9. Multi-site phosphorylation of NDC80 by Aurora A is counteracted by PP1/PP2A.

(A) Parental and PPP6C KO cell lines were treated with STLC to arrest cells in mitosis in the absence (Control) and presence of PP1/2A-i, then stained for NDC80 pS55, CENP-A and DNA. **(B)** Level of NDC80 pS55 signal at kinetochores was measured for the different conditions (n = 701-951 KTs). Each mean difference is depicted as a dot. Each 95% confidence interval is indicated by the ends of the vertical error bars; the confidence interval is bias-corrected and accelerated. **(C)** The number of NDC80 pS55 positive kinetochores was measured for the different conditions (mean ± SD; n = 12-15). **Statistical significance was analysed using a Brown-Forsythe ANOVA (**, p < 0.01)**. **(D and E)** Mitotic lysates of parental and PPP6C KO HeLa cells treated with Aurora A (AurA-i), Aurora B and phosphatase (calyculin, PP1/2A-i) inhibitors in the combinations and order shown were blotted for the proteins listed in the figure. Overall NDC80 phosphorylation was monitored using a Phos-tag gel. **(F)** HeLa cells depleted of endogenous NDC80 using a 5'-UTR siRNA were transfected with NDC80-GFP wild type (WT) and phospho-deficient mutant constructs as shown in the figure. Cells were stained for DNA, NDC80, and with specific NDC80 phospho-antibodies. **(G)** Spindle size is plotted for the NDC80 wild type and point mutant in panel F (mean ± SD; n = 42-58). **Statistical significance was analysed using a Brown-Forsythe ANOVA (****, p < 0.0001)**. **(H)** A schematic showing the proposed dynamic sub-stoichiometric phosphorylation of the NDC80 N-terminus, and roles of Aurora A and PP1/PP2A.

Fig. 10. NDC80 phospho-mutants rescue the spindle size and micronucleation defects seen in PP6-deficient cells.

(A) HeLa FlpIn T-REx cells expressing wild-type NDC80-GFP (WT), phospho-deficient (9A) and phospho-mimetic (9D) were depleted of endogenous NDC80 for 72 h using a 5'-UTR siRNA and then stained for tubulin and DNA. NDC80-GFP was visualized directly. Enlarged regions show the astrin staining at kinetochores in mitotic cells. (B) Metaphase spindle length (mean \pm SD; n = 8-21) is plotted in the graph. Statistical significance was analysed using a Brown-Forsythe ANOVA (*, p < 0.05; **, p < 0.01; ****, p < 0.0001). (C) HeLa FlpIn T-REx cells expressing NDC80-GFP WT, 9A and 9D were transfected with control or PPP6C siRNA for 72 h, treated with STLC for 3 h to create monopolar spindles and then stained for astrin. NDC80-GFP was visualized directly. Enlarged regions show the presence or loss of astrin staining at NDC80-GFP labelled kinetochores. (D) Monoastral spindle diameter and (E) the number of astrin-positive kinetochores were measured for the conditions shown in C (mean \pm SD; n = 12-38). Statistical significance was analysed using a Brown-Forsythe ANOVA (**, p < 0.01; ****, p < 0.0001). (F) NDC80-GFP (WT), 9A and 9D expressing cells arrested with STLC for 3 h to create monopolar spindles were treated with Aurora A or Aurora B inhibitors for 30 min and then stained for tubulin and DNA. NDC80-GFP was visualized directly. (G). Monoastral spindle diameter was measured for the conditions shown in F (mean \pm SD; n = 7-16). Statistical significance was analysed using a Brown-Forsythe ANOVA (*, p < 0.05; **, p < 0.01; ****, p < 0.0001). (H) HeLa FlpIn T-REx NDC80-GFP WT, 9A and 9D cells treated with control or PPP6C siRNA for 72 h were stained for DNA, and (I) the frequency of cells with nuclear morphology defects scored (mean \pm SD; n = 4). Statistical significance was analysed using a one-way ANOVA (****, p < 0.0001). (J) A schematic depicting the effects of PP6-regulated Aurora A-TPX2 activity on NDC80 phosphorylation and spindle size is shown to the right. Aurora A inhibition (AurA-i), TPX2 depletion (siTPX2) and NDC80-9A expression result in reduced spindle size, whereas Aurora A activity amplification in PPP6C KO and NDC80-9D expression both increase spindle size.

Fig. S2. Comparative functional genomics screening for synthetic growth defects in parental and PPP6C KO eHAP cells.

(A) Workflow for genome-wide CRISPR knockout screens using the GeCKO V2 libraries, with data analysis in MaGeCK comparing gene selection in PPP6C KO to parental haploid eHAP cells. (B) DNA sequence of the PPP6C genomic locus showing the sequence of candidate PPP6C KO alleles in 3 candidate haploid eHAP cell clones. (C) Western blot of parental eHAP and candidate PPP6C KO alleles showing loss of PPP6C protein and elevation of active Aurora A pT288. (D) Parental eHAP and PPP6C KO clone #1 stained for Aurora A pT288, tubulin, and DNA. Representative cells in metaphase are shown, with arrowheads to mark the spread of active Aurora A on the mitotic spindle. Circled areas and numbers indicate the spindle diameter in μ m (\emptyset). (E) Parental eHAP and PPP6C KO clones cells stained for nuclear pore proteins (NUPs) and DNA. Groups of interphase cells are shown, with arrowheads to indicate micronuclei. (F) Frequency plot of median enrichment of all genes and a selected set of core essential genes in screens 1 and 2. (G) Clonogenic survival assays for 3 gRNA sequences targeting NDC80 in parental and PPP6C KO eHAP cell lines. Example images of survival assays are shown. (H) Clonogenic survival assays for NDC80 gRNAs g1-g3 relative to the control gRNA in eHAP cells (mean \pm SD; n = 3-5). Statistical significance was analysed using an unpaired two-tailed t-test with Welch's correction (*, p < 0.05; **, p < 0.01). (I) Western blot validation of NDC80 depletion by the NDC80 g1-g3 gRNAs in eHAP cells. Note g2 results in reduced expression and a ladder of truncated NDC80 protein species. (J) Parental and PPP6C KO HeLa cells were treated for 48 or 72 h with siRNA for the indicated negatively selected genes identified by genome wide screening as candidates for synthetic lethality with PPP6C KO. The proportion of morphologically abnormal nuclei is plotted in the graph.

Fig. S3. Analysis of NDC80 phosphorylation in mitosis using mass spectrometry.

(A). A schematic of NDC80 showing predicted Aurora and CDK1 consensus sites in the N-terminal region. HeLa cells were arrested in mitosis with nocodazole and then released to allow mitotic spindle formation in the presence of Aurora A and Aurora B kinase inhibitors. Cell lysates were prepared and endogenous NDC80 isolated by immune precipitation (NDC80 IP) and mass spectrometry as described in the methods. MS/MS spectra for NDC80 phosphopeptides are shown for pT31, pS55 and pS69. Due to incomplete cleavage the peptide containing pS57/15 was not quantified. (B) Western blot of NDC80 IPs showing effects of Aurora A

and B inhibition on NDC80 pS55 (mean; n = 2). (C) Mass spectrometry of NDC80 IPs showing effects of Aurora A and B inhibition on NDC80 pS55, pS69 and the CDK-consensus site at pT31 (mean ± SD; n = 3-4). Statistical significance was analysed using a Brown-Forsythe ANOVA (**, p < 0.01; ***, p < 0.001). (D) HeLa cells arrested in mitosis with nocodazole were released for 25 min to allow mitotic spindle formation in the presence of Aurora A and Aurora B kinase inhibitors as indicated. Cell lysates and NDC80 IPs were western blotted with antibodies to Aurora A and B, activated Aurora A pT288, a pan-phospho-Aurora antibody detecting pT288 and pT232, NDC80, NDC80 pS55, pS44, pT61-pS62 (rabbit, R; sheep, Sh as indicated in the figure). (E) To test for NDC80 phospho-antibody specificity, HeLa cells were transfected for 24 h with plasmids expressing GFP (control), NDC80-GFP (WT), S44A, S55A, S62A, T61-S62A point mutants or a combined 9A mutant where all consensus Aurora sites described in A. HeLa cells arrested in mitosis for 18h with nocodazole were released for 25 min to allow mitotic spindle formation in the presence of Aurora A and Aurora B kinase inhibitors as indicated. NDC80 IPs were western blotted with antibodies to Aurora A and B, activated Aurora A pT288, a pan-phospho-Aurora antibody detecting pT288 and pT232, NDC80, NDC80 pS44, pS55, pS62 or pT61-pS62 (rabbit, R; sheep, Sh as indicated in the figure).

Fig. S4. Specificity of NDC80 antibodies and phosphoantibodies.

(A) HeLa cells were depleted of endogenous NDC80 for 48 h using either an NDC80 siRNA SMARTpool or a single siRNA to the 5'-UTR of NDC80, or treated with a non-targeting control siRNA (siControl). Cells were fixed and then stained for NDC80 (9G3 mouse monoclonal), CENP-C, DNA or antibodies to specific NDC80 phosphorylation sites pS55, pS44 and pS62. The graphs in each panel show the NDC80 phospho-antibody signal is significantly reduced after NDC80 siRNA (mean ± SD; n = 5-11). Statistical significance was analysed using a Brown-Forsythe ANOVA (**, p < 0.01; ***, p < 0.0001). (B) NDC80 siRNAs resulted in loss of NDC80 but not the inner centromere protein CENP-C (mean ± SD; n = 19-24). Statistical significance was analysed using a Brown-Forsythe ANOVA (****, p < 0.0001). (C) HeLa cells were depleted of endogenous TPX2 for 72h or treated with a non-targeting control siRNA (siControl). Cells were fixed and then stained for TPX2, CENP-C, DNA or antibodies to specific NDC80 phosphorylation sites pS44 and pS62. TPX2 siRNA resulted in loss of TPX2 but not the inner centromere protein CENP-C (mean ± SD; n = 23-57), confirmed by western blot. Statistical significance was analysed using an unpaired two-tailed t-test with Welch's correction (****, p < 0.0001). (D) HeLa cells were depleted of Aurora A or Aurora B for 48 h using siRNA, or treated with a non-targeting control siRNA (siControl). Cells were fixed and then stained with antibodies to specific NDC80 phosphorylation sites pS55, pS44 and pS62. (E) Cells treated as in D were stained for Aurora A, Aurora B, CENP-C, DNA or (F) the spindle checkpoint proteins BUB1 and BUBR1.

Fig. S5. NDC80 phosphorylation is elevated in PPP6C KO cells, independent of the spatial position of chromosomes within the mitotic spindle.

(A) Parental and PPP6C KO HeLa cells were arrested in metaphase with MG132 with aligned chromosomes, with MG132 and CENP-E (CenpE-i) inhibitor to trap some chromosomes at the spindle poles, or with MG132, CENP-E and Aurora A (AurA-i) inhibitors. Cells were stained for NDC80 pS55, astrin, CENP-A, and DNA. Enlarged panels for CENP-E inhibitor conditions show kinetochores close to spindle poles. (B) NDC80 pS55 intensity (mean ± SD; n = 101-709 KTs) at metaphase aligned chromosomes (MG132) and chromosomes trapped at spindle poles (MG132+CenpE-i). Aurora A inhibitor (AurA-i) was used to confirm NDC80 pS55 signal was dependent on Aurora A activity for aligned and trapped chromosomes. Statistical significance was analysed using a Dunn's multiple comparison test (****, p < 0.0001).

5) Abstract and title: The abstract should be no longer than 160 words and should communicate the significance of the paper for a general audience. The title should be less than 100 characters including spaces. Make the title concise but accessible to a general readership.

6) Materials and methods: Should be comprehensive and not simply reference a previous publication for details on how an experiment was performed. Please provide full descriptions in the text for readers who may not have access to referenced manuscripts.

** Please add details as noted above in our decision letter.

We have updated the Results and Methods Sections to include the details requested. Highlighted text marks the key changes.

Page 17, lines 31-40.

Gene silencing (RNAi) and RNAi-rescue assays

RNA duplexes were purchased from Dharmacon and Qiagen (listed in Table S2). Cells were transfected with siRNA using Oligofectamine for the times described in the figure legends. For RNAi-rescue assays in Flp-In cells (Fig. 10), NDC80-GFP transgenes were induced by 1 µg/ml doxycycline, while endogenous NDC80 was depleted using oligonucleotides against the 5' -UTR. For RNAi-rescue assays with transient transfection of TPX2 (Fig. 3B, 3C and 3F-3H) or NDC80 (Fig. 9F and 9G), cells were transfected twice with GFP-TPX2 or NDC80-GFP expression plasmids at 72 and 24 hours before fixation. Endogenous TPX2 or NDC80 were depleted using siRNA against the 3'-UTR for TPX2 or the 5'-UTR for NDC80. The siRNA transfection was performed 48 hours before fixation."

7) Please be sure to provide the sequences for all of your primers/oligos and RNAi constructs in the materials and methods. You must also indicate in the methods the source, species, and catalog numbers (where appropriate) for all of your antibodies. Please also indicate the acquisition and quantification methods for immunoblotting/western blots.

8) Microscope image acquisition: The following information must be provided about the acquisition and processing of images:

- a. Make and model of microscope
- b. Type, magnification, and numerical aperture of the objective lenses
- c. Temperature
- d. Imaging medium
- e. Fluorochromes
- f. Camera make and model
- g. Acquisition software
- h. Any software used for image processing subsequent to data acquisition. Please include details and types of operations involved (e.g., type of deconvolution, 3D reconstitutions, surface or volume rendering, gamma adjustments, etc.).

10) Supplemental materials: There are strict limits on the allowable amount of supplemental data. Articles may have up to 5 supplemental figures. Please also note that tables, like figures, should be provided as individual, editable files. A summary of all supplemental material should appear at the end of the Materials and methods section.

A summary of supplemental material is provided at the end of the Materials and methods section.

An eTOC summary has been added to the manuscript file.

13) ORCID IDs: ORCID IDs are unique identifiers allowing researchers to create a record of their various scholarly contributions in a single place. At resubmission of your final files, please consider providing an ORCID ID for as many contributing authors as possible.

Please note that JCB now requires authors to submit Source Data used to generate figures containing gels and Western blots with all revised manuscripts. This Source Data consists of fully uncropped and unprocessed images for each gel/blot displayed in the main and supplemental figures. Since your paper includes cropped gel and/or blot images, please be sure to provide one Source Data file for each figure that contains gels and/or blots along with your revised manuscript files. File names for Source Data figures should be alphanumeric without any spaces or special characters (i.e., SourceDataF#, where F# refers to the associated main figure number or SourceDataFS# for those associated with Supplementary figures). The lanes of the gels/blots should be labeled as they are in the associated figure, the place where cropping was applied should be marked (with a box), and molecular weight/size standards should be labeled wherever possible.

WHEN APPROPRIATE: The source code for all custom computational methods published in JCB must be made freely available as supplemental material hosted at www.jcb.org. Please contact the JCB Editorial Office to find out how to submit your custom macros, code for custom algorithms, etc. Generally, these are provided as raw code in a .txt file or as other file types in a .zip file. Please also include a one-sentence summary of each file in the Online Supplemental Material paragraph of your manuscript.

B. FINAL FILES:

We have provided some images as suggestions for the cover.